# The Franz–Parisi Criterion and Computational Trade-offs in High Dimensional Statistics

**Afonso S. Bandeira**
Department of Mathematics
ETH Zürich
bandeira@math.ethz.ch

**Ahmed El Alaoui**
Department of Statistics and Data Science
Cornell University
ae333@cornell.edu

**Samuel B. Hopkins**
MIT EECS
Cambridge, MA
samhop@mit.edu

**Tselil Schramm**
Department of Statistics
Stanford University
tselil@stanford.edu

**Alexander S. Wein**
Department of Mathematics
University of California, Davis
aswein@ucdavis.edu

**Ilias Zadik**
Department of Mathematics
MIT
izadik@mit.edu

## Abstract

Many high-dimensional statistical inference problems are believed to possess inherent computational hardness. Various frameworks have been proposed to give rigorous evidence for such hardness, including lower bounds against restricted models of computation (such as low-degree functions), as well as methods rooted in statistical physics that are based on free energy landscapes. This paper aims to make a rigorous connection between the seemingly different low-degree and free-energy based approaches. We define a free-energy based criterion for hardness and formally connect it to the well-established notion of low-degree hardness for a broad class of statistical problems, namely all Gaussian additive models and certain models with a sparse planted signal. By leveraging these rigorous connections we are able to: establish that for Gaussian additive models the "algebraic" notion of low-degree hardness implies failure of "geometric" local MCMC algorithms, and provide new low-degree lower bounds for sparse linear regression which seem difficult to prove directly. These results provide both conceptual insights into the connections between different notions of hardness, as well as concrete technical tools such as new methods for proving low-degree lower bounds.

## 1   Introduction

Many inference problems in high dimensional statistics appear to exhibit an *information-computation gap*, wherein at some values of the signal-to-noise ratio, inference is information-theoretically possible, but no (time-)efficient algorithm is known. Well-known problems that exhibit such gaps include sparse linear regression, sparse principal component analysis (PCA), tensor PCA, planted clique, community detection, graph coloring, and many others (we point the reader to the survey references [ZK16a, BPW18, RSS19, KWB19, Gam21] and references therein for many examples).

36th Conference on Neural Information Processing Systems (NeurIPS 2022).

A priori, it is unclear whether such gaps are a symptom of the inherent computational intractability of these problems, or whether they instead reflect a limitation of our algorithmic ingenuity. One of the main goals in this field is to provide, and understand, *rigorous evidence* for the existence of an information-computation gap. Indeed, there are several mature tools to establish *statistical* or *information-theoretic lower bounds*, and these often sharply characterize the signal-to-noise ratio at which inference is possible. We have relatively fewer tools for establishing *computational* lower bounds in statistical settings, and the study of such tools is still in its early days. Broadly, there are three approaches: (i) establishing computational equivalence between suspected-to-be-hard problems via reductions, (ii) proving lower bounds within restricted models of computation, or in other words, ruling out families of known algorithms, and (iii) characterizing geometric properties of the problem that tend to correspond to computational hardness, often by studying a corresponding energy or free energy landscape of the posterior distribution of the signal given the data, and establishing the existence of 'large barriers' in this landscape. In some cases it can be rigorously shown that these properties impede the success of certain families of algorithms (notable examples include the work of Jerrum [Jer92] and Gamarnik and Sudan [GS17]; see also references therein for other instances).

These complementary approaches give us a richer understanding of the computational landscape of high-dimensional statistical inference. Reductions contribute to establishing equivalence classes of (conjectured hard) problems, and lower bounds against restricted models, or characterizations of the problem geometry, give concrete evidence for computational hardness within the current limits of known algorithms. There have been considerable recent advances in this context (see for example [BB20], the surveys [ZK16a, BPW18, RSS19, KWB19, Gam21], and references therein). One particularly exciting direction, which is the topic of this paper, is the pursuit of rigorous connections between different computational lower bound approaches. For instance, a recent result shows (under mild assumptions) that lower bounds against statistical query algorithms and low-degree polynomials are essentially equivalent [BBH+20]. Results of this type help to unify our understanding about what makes problems hard. Following the work of [BHK+19, HKP+17, HS17] in the context of the sum-of-squares hierarchy of algorithms, a conjecture was made that there is a large and easy-to-characterize *universality class* of intractable problems [Hop18]: those for which low-degree statistics cannot distinguish data with a planted signal from (suitably defined) random noise. These problems are "hard for the low-degree likelihood ratio" or "low-degree hard," which we will define precisely below. Many problems mentioned above fall into this class precisely in the regime of their information-computation gaps.

Another successful approach to understand computational hardness of statistical problems borrows tools from statistical physics: tools such as the *cavity method* and *replica method* can be used to make remarkably precise predictions of both statistical and computational thresholds, essentially by studying properties of *free energy potentials* associated to the problem in question, or by studying related iterative algorithms such as belief propagation or approximate message passing (see e.g. [DKMZ11, LKZ15a, LKZ15b, DMK+16]) – the type of "free energy barrier" encountered in these potentials that is (heuristically) suggested to lead to computational hardness also often has an appealing interpretation as a barrier to algorithms based on Markov chain Monte Carlo methods. However, the physics approaches are not without drawbacks, chief among them (for us) that it is notoriously difficult to pin down with mathematical rigor just what the free energy potentials (certain one-dimensional curves) in question are, or precisely what properties of them are meant to predict computational hardness. Furthermore, it is clear that some versions of the "physics recipe" can make erroneous predictions (for instance, in the case of the tensor PCA problem), and can require seemingly problem-specific modifications to be redeemed [RM14, LML+17, BGJ20, WEM19, BCRT20, GZ19]. This has made it challenging to push any mathematical account of the physics approach to predicting computational hardness beyond a one-problem-at-a-time theory.

**Main Contributions**  A tantalizing question and a step towards a rigorous free-energy based theory of statistical hardness is whether some free-energy based criterion is actually rigorously connected with low-degree hardness. This paper aims to achieve exactly this: to make a rigorous connection between the low-degree and free-energy based approaches in the setting of statistical inference. (We note that this setting differs from that of random optimization problems with no planted signal, where a connection of this nature has already been established [GJW20, Wei22, BH21].) Our first contribution is a formal definition of a free-energy based criterion for computational hardness, the *Franz–Parisi criterion* (Definition 3.2) inspired by the so-called Franz–Parisi potential [FP95]. We formally connect this criterion to low-degree hardness for a broad class of statistical problems,

namely all Gaussian additive models (Theorems 4.3 and 4.4) and certain sparse planted models (e.g. Theorem 6.2). By leveraging these rigorous connections we are able to (i) establish that in the context of Gaussian additive models, low-degree hardness implies failure of local MCMC algorithms (Corollary 5.4), and (ii) provide new low-degree lower bounds for sparse linear regression which seem difficult to prove directly (Theorem 6.2). In the supplementary material, we also investigate whether the Franz–Parisi criterion, as we defined it, is accurate beyond Gaussian Additive Models, and find, with counterexamples, that it can make erroneous predictions in simple discrete settings. We thus leave as an exciting future direction the problem of determining its precise domain of applicability, and investigating what other free-energy based criteria may be more suitable in other inference problems.

## 2 Preliminaries

We will focus on problems in which there is a signal vector of interest $u \in \mathbb{R}^n$, drawn from a prior distribution $\mu$ over such signals, and the data observed is a sample from a distribution $\mathbb{P}_u$ on $\mathbb{R}^N$ that depends on the signal $u$. We focus mainly on *hypothesis testing*: we are given a sample generated either from the "planted" distribution $\mathbb{P} = \mathbb{E}_{u \sim \mu} \mathbb{P}_u$ (a mixture model were the data is drawn from $\mathbb{P}_u$ for a random $u \sim \mu$) or from a "null" reference distribution $\mathbb{Q}$ representing pure noise, and the goal is to decide whether it is more likely that the sample came from $\mathbb{P}$ or $\mathbb{Q}$. While not our main focus, we note that in the section on MCMC hardness we do also discuss the related task of *estimation/recovery*: given a sample from $\mathbb{P}_u$ with the promise that $u \sim \mu$, the goal is to estimate $u$ (different estimation error targets correspond to different versions of this problem, often referred to *weak/approximate recovery* or *exact recovery*). We now begin with the formal definitions.

**Problem 2.1** (High Dimensional Inference: Hypothesis Testing)**.** Given positive integers $n, N$, and distributions $\mu$ on $\mathbb{R}^n$, and $\mathbb{P}_u$ on $\mathbb{R}^N$ for each $u \in \mathrm{supp}(\mu)$, the goal is to perform simple hypothesis testing between the Null model $\mathbf{H_0} : Y \sim \mathbb{Q}$ and the Planted model $\mathbf{H_1} : Y \sim \mathbb{P} = \mathbb{E}_{u \sim \mu} \mathbb{P}_u$.

We will be especially interested in asymptotic settings where $n \to \infty$ and the other parameters scale with $n$ in some prescribed way: $N = N_n$, $\mu = \mu_n$, $\mathbb{P} = \mathbb{P}_n$, $\mathbb{Q} = \mathbb{Q}_n$. In this setting, we focus on the following two objectives.

**Definition 2.2** (Strong/Weak Detection)**.** *Strong detection* is achieved if the sum of type I and type II errors[1] tends to 0 as $n \to \infty$. *Weak detection* is achieved if the sum of type I and type II errors is at most $1 - \varepsilon$ for some fixed $\varepsilon > 0$ (not depending on $n$).

Throughout, we will work in the Hilbert space $L^2(\mathbb{Q})$ of (square integrable) functions $\mathbb{R}^N \to \mathbb{R}$ with inner product $\langle f, g \rangle_{\mathbb{Q}} := \mathbb{E}_{Y \sim \mathbb{Q}}[f(Y)g(Y)]$ and corresponding norm $\|f\|_{\mathbb{Q}} := \langle f, f \rangle_{\mathbb{Q}}^{1/2}$. For a function $f : \mathbb{R}^N \to \mathbb{R}$ and integer $D \in \mathbb{N}$, we let $f^{\leq D}$ denote the orthogonal (w.r.t. $\langle \cdot, \cdot \rangle_{\mathbb{Q}}$) projection of $f$ onto the subspace of polynomials of degree $\leq D$. We assume that $\mathbb{P}_u$ is absolutely continuous w.r.t. $\mathbb{Q}$ for all $u \in \mathrm{supp}(\mu)$, use $L_u := \frac{d\mathbb{P}_u}{d\mathbb{Q}}$ to denote the likelihood ratio, and assume that $L_u \in L^2(\mathbb{Q})$ for all $u \in \mathrm{supp}(\mu)$. $L := \frac{d\mathbb{P}}{d\mathbb{Q}} = \mathbb{E}_{u \sim \mu} L_u$ is the likelihood ratio of $\mathbb{P}$ and $\mathbb{Q}$.

### 2.1 (Low-degree) Likelihood Ratio and Computational Complexity of Inference

A key quantity of interest is the (squared) norm of the likelihood ratio, which is related to the *chi-squared divergence* $\chi^2(\mathbb{P} \,\|\, \mathbb{Q})$ as $\|L\|_{\mathbb{Q}}^2 = \chi^2(\mathbb{P} \,\|\, \mathbb{Q}) + 1$. If this quantity is asymptotically bounded (by $O(1)$ or $1 + o(1)$), there are well-known consequences for *information-theoretic* impossibility of testing (impossibility of strong and weak detection, respectively); see [MRZ15, Lemma 2]. The *low-degree likelihood ratio*, $L^{\leq D}$, which recall means the projection of the likelihood ratio onto the subspace of degree-$D$ polynomials, has been studied intensively in recent years as a means to "predict" computational complexity of high-dimensional testing problems (see [Hop18, KWB19]).

**Definition 2.3** (Low-Degree Likelihood Ratio)**.** Define the squared norm of the degree-$D$ likelihood ratio (also called the "low-degree likelihood ratio") to be the quantity

$$\mathrm{LD}(D) := \|L^{\leq D}\|_{\mathbb{Q}}^2 = \left\| \left( \underset{u \sim \mu}{\mathbb{E}} L_u \right)^{\leq D} \right\|_{\mathbb{Q}}^2 = \underset{u, v \sim \mu}{\mathbb{E}} \left[ \langle L_u^{\leq D}, L_v^{\leq D} \rangle_{\mathbb{Q}} \right], \tag{1}$$

---

[1]Type I error is the probability of outputting "$\mathbb{P}$" when given a sample from $\mathbb{Q}$. Type II error is the probability of outputting "$\mathbb{Q}$" when given a sample from $\mathbb{P}$.

where the last equality follows from linearity of the projection operator, and where $u, v$ are drawn independently from $\mu$. For some increasing sequence $D = D_n$, we say that the hypothesis testing problem above is *hard for the degree-D likelihood* or simply *low-degree hard* if $\mathrm{LD}(D) = O(1)$.

Heuristically speaking, the interpretation of $\mathrm{LD}(D)$ should be thought of as analogous to that of $\|L\|_{\mathbb{Q}}^2$ but for computationally-bounded tests: if $\mathrm{LD}(D) = O(1)$ (or $1 + o(1)$) this suggests computational hardness of strong (or weak, respectively) detection. The parameter $D = D_n$ should be loosely thought of as a proxy for the runtime allowed for our testing algorithm, where $D = O(\log n)$ corresponds to polynomial time and more generally, larger values of $D$ correspond to runtime $\exp(\tilde{\Theta}(D))$ where $\tilde{\Theta}$ hides factors of $\log n$. See supplementary material for further discussion of the relationship between low-degree hardness and hardness against other classes of algorithms.

## 3 The Franz Parisi criterion

We define a predictor of computational hardness, which we call the *Franz–Parisi criterion*. It is inspired by well-established ideas rooted in statistical physics, which we discuss further below and in the supplement. However, the precise definition we use here has not appeared before (to our knowledge). Throughout this paper we will argue for the significance of this criterion in a number of ways: its conceptual link to physics, its provable equivalence to the low-degree criterion for Gaussian additive models, its formal connection to MCMC methods for Gaussian additive models (Section 5), and its usefulness as a tool for proving low-degree lower bounds (Section 6).

**Definition 3.1** (Low-Overlap Likelihood Norm)**.** We define the *low-overlap likelihood norm* at overlap $\delta \geq 0$ as

$$\mathrm{LO}(\delta) := \mathop{\mathbb{E}}_{u,v \sim \mu} \left[ \mathbb{1}_{|\langle u,v \rangle| \leq \delta} \cdot \langle L_u, L_v \rangle_{\mathbb{Q}} \right], \tag{2}$$

where $u, v$ are drawn independently from $\mu$.

**Definition 3.2** (Franz–Parisi Criterion)**.** We define the *Franz–Parisi Criterion at D deviations* to be the quantity

$$\mathrm{FP}(D) := \mathrm{LO}(\delta), \quad \text{for } \delta = \delta(D) := \sup \left\{ \varepsilon \geq 0 \text{ s.t. } \mathop{\mathrm{Pr}}_{u,v \sim \mu} (|\langle u,v \rangle| \geq \varepsilon) \geq e^{-D} \right\}. \tag{3}$$

For some increasing sequence $D = D_n$, a problem is *FP-hard at D deviations* if $\mathrm{FP}(D) = O(1)$.

Heuristically speaking, $\mathrm{FP}(D)$ should be thought of as having a similar interpretation as $\mathrm{LD}(D)$: if $\mathrm{FP}(D) = O(1)$ this suggests hardness of strong detection, and if $\mathrm{FP}(D) = 1 + o(1)$ this suggests hardness of weak detection. The parameter $D$ is a proxy for runtime and corresponds to the parameter $D$ in $\mathrm{LD}(D)$, as we justify below.

We remark that $\mathrm{LD}(D)$ and $\mathrm{FP}(D)$ can be thought of as different ways of "restricting" the quantity

$$\|L\|_{\mathbb{Q}}^2 = \mathop{\mathbb{E}}_{u,v \sim \mu} \left[ \langle L_u, L_v \rangle_{\mathbb{Q}} \right], \tag{4}$$

which, recall, is related to information-theoretic *impossibility* of testing. For LD, the restriction is low-degree projection on each $L_u$, while for FP it excludes pairs $(u, v)$ of high overlap. Our results will show that (in some settings) these two restrictions are nearly equivalent.

**Relation to statistical physics** We defer to supplementary material an extended discussion of the relationship between the Franz–Parisi criterion and ideas from statistical physics. See also [MM09, ZK16b] for exposition on the well-explored connections between statistical physics and Bayesian inference. Roughly speaking, the logarithm of $\mathrm{LO}(\delta)$ is the 'annealed' approximation near $\delta = 0$ of the *Franz–Parisi potential* [FP95]; a tool used to study the free energy landscape of a disordered system locally around a reference configuration at equilibrium. In our statistical context this potential takes the form

$$f(\delta) = \mathbb{E}_{u \sim \mu} \mathbb{E}_{Y \sim \mathbb{P}_u} \log \mathbb{E}_{v \sim \mu} \left[ \mathbb{1}_{\langle u,v \rangle = \delta} L_v(Y) \right].$$

Roughly speaking, $f(\delta)$ measures the amount of mass that the posterior distribution places on vectors of overlap $\delta$ with the true signal. Jensen's inequality leads to the upper bound $f(\delta) \leq f^{\mathrm{ann}}(\delta) := \log \mathbb{E}_{u,v \sim \mu} \left[ \mathbb{1}_{\langle u,v \rangle = \delta} \langle L_u, L_v \rangle_{\mathbb{Q}} \right]$, which we call the *annealed FP potential*, c.f. $\mathrm{LO}(\delta)$, Eq. (2).

For inference problems which are information-theoretically solvable, most of the mass of the posterior distribution of $v \mid Y$, when $Y \sim \mathbb{P}_u$ is concentrated on those $v$ with large $\langle u, v \rangle$. But to predict computational complexity, the Franz–Parisi criterion stipulates that it is the contributions of $v$ with $|\langle u, v \rangle| \leq \delta$ for small values of $\delta$ which matter: when such $v$ contribute noticeably, $\mathrm{LO}(\delta)$ is large and hence so is $\mathrm{FP}(D)$.

The FP criterion, Definition 3.2, is whether $\mathrm{FP}(D)$ stays bounded or diverges as $n \to \infty$ for some choice of an increasing sequence $D = D_n$. A heuristic justification of this criterion is as follows: We should first note that since $L_u \geq 0$, we have $\mathrm{FP}(D) \leq \|L\|_{\mathbb{Q}}^2$. Thus if $\|L\|_{\mathbb{Q}} \to \infty$ but $\mathrm{FP}(D) = O(1)$, then the divergence of $\|L\|_{\mathbb{Q}}$ must be due to contributions to the sum Eq. (4) with high overlap values: $|\langle u, v \rangle| \gg \delta(D)$. Suppose now there is a free energy barrier separating small overlaps $|\langle u, v \rangle| \leq \delta(D)$ from larger ones $|\langle u, v \rangle| \gg \delta(D)$. For instance, suppose $\langle u, v \rangle = 0$ is a local maximum of the potential, separated by a barrier from a global maximum located at $\langle u, v \rangle \gg \delta(D)$ (see Fig. 1, Panel $(b)$), then one needs much more than $e^{O(D)}$ samples to guess an overlap value on the other side of the barrier and land in the basin of attraction of the global maximum. This suggests that tests distinguishing $\mathbb{P}$ and $\mathbb{Q}$ cannot be constructed in time $e^{O(D)}$. One of our main results (see Section 4) is an equivalence relation between the $\mathrm{FP}(D)$ criterion and the $\mathrm{LD}(D')$ criterion for Gaussian models, where $D' = \tilde{\Theta}(D)$, therefore grounding this heuristic in a rigorous statement.

**Example: The spiked Wigner model** As a concrete example, let us consider the spiked Wigner model with sparse Rademacher prior: The signal vector $u$ has i.i.d. entries drawn from a three-point prior $\mu_0 = \frac{\rho}{2}\delta_{+1/\sqrt{\rho}} + (1-\rho)\delta_0 + \frac{\rho}{2}\delta_{-1/\sqrt{\rho}}$, and for $1 \leq i \leq j \leq n$ we let $Y_{ij} = \frac{\lambda}{\sqrt{n}}u_i u_j + Z_{ij}$, where $Z$ is drawn from the Gaussian Orthogonal Ensemble: $Z_{ij} \sim N(0,1)$ for $i < j$ and $Z_{ii} \sim N(0,2)$. The null distribution is pure Gaussian noise: $Y = Z$. In this case, known efficient algorithms succeed at detecting/estimating the signal $u$ if and only if $\lambda > 1$ [BGN11, LM19, CL19]. Furthermore, the threshold $\lambda_{\mathrm{ALG}} = 1$ is information-theoretically tight when $\rho = 1$ (or more generally, if $\rho$ is larger than a known absolute constant). On the other hand if $\rho$ is small enough, then detection becomes information-theoretically possible for some $\lambda < 1$ but no known polynomial-time algorithm succeeds in the regime [BMV$^+$17, PWBM18, AKJ20]. Let us check that the behavior of the annealed potential is qualitatively consistent with these facts. A small computation (see Section 4) leads to the expression

$$\langle L_u, L_v \rangle_{\mathbb{Q}} = \exp\left(\frac{\lambda^2}{2n}\langle u, v \rangle^2\right),$$

and the annealed FP potential is

$$f^{\mathrm{ann}}(k/\rho) = \log \Pr\left(\langle u, v \rangle = k/\rho\right) + \frac{\lambda^2 k^2}{2n\rho^2}.$$

Letting $k = \lfloor nx \rfloor$, and using Stirling's formula, we obtain a variational formula for the annealed FP potential: $f^{\mathrm{ann}}(\lfloor nx \rfloor/\rho) = n\phi(x) + o(n)$ where

$$\phi(x) = \max_p \left\{ h(p) + (1-p_0)\log(\rho^2/2) + p_0\log(1-\rho^2) \right\} + \frac{\lambda^2 x^2}{2\rho^2}, \quad x \in [-1, 1]. \quad (5)$$

The max is over probability vectors $p = (p_{-1}, p_0, p_1)$ satisfying $p_1 - p_{-1} = x$, and $h(p) = -p_{-1}\log p_{-1} - p_0 \log p_0 - p_1 \log p_1$. It is not difficult to check that $\phi(0) = \phi'(0) = 0$, and $\phi''(0) = (\lambda^2 - 1)/\rho^2$. Hence when $\lambda < \lambda_{\mathrm{ALG}} = 1$, $f^{\mathrm{ann}}$ is negative for all $|x| \leq \varepsilon$ for some $\varepsilon = \varepsilon(\lambda, \rho)$ (Fig 1, Panels $(a), (b)$). This indicates that $\mathrm{FP}(D)$ is bounded for $D \leq c(\varepsilon)n$. On the other hand, if $\lambda > 1$, $f^{\mathrm{ann}} > 0$ over $[-\varepsilon, \varepsilon]$ for some $\varepsilon > 0$, which indicates that $\mathrm{FP}(D) \to \infty$ for $D = D_n \to \infty$ slowly (Panel $(c)$). One can also look at the global behavior of $f^{\mathrm{ann}}$, which we plot in Figure 1. Panel $(b)$ represents a scenario where $x = 0$ is a local maximum separated from the two symmetric global maxima by a barrier, while in panels $(a)$ and $(c)$, no such barrier exists.

**Differences to prior physical hardness criteria** Our FP criterion is conceptually similar to ideas that have appeared before in statistical physics, but there are a few key differences. While free energy barriers are typically thought of as an obstruction to algorithmic *recovery*, our criterion—due to its connection with $\|L\|_{\mathbb{Q}}^2$—is instead designed for the *detection* problem. As such, we do not expect the annealed FP to make sharp predictions about estimation error (e.g. MMSE) like approaches based on approximate message passing (AMP), e.g. [DMM09, LKZ15a, LKZ15b, DMK$^+$16]. (For instance,

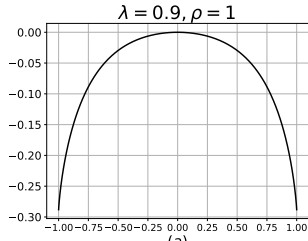 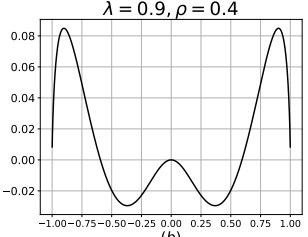 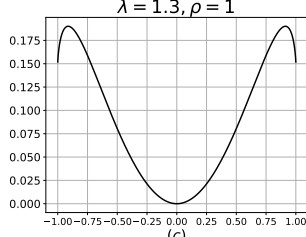

Figure 1: The annealed FP potential, Eq. (5), for various value of $\lambda$ and $\rho$. Panel $(a)$: A global maximum at $x = 0$. Panel $(b)$: Local maximum at $x = 0$ separated from two global maxima by a 'barrier'. Panel $(c)$: A local minimum at $x = 0$.

Figure 1, Panel $(b)$ *wrongly* predicts that estimation is 'possible but hard' for $\lambda = 0.9$, $\rho = 0.4$, when it is in fact information-theoretically impossible [LM19].) On the other hand, one advantage of our criterion is that, by virtue of its connection to LD in Section 4, it predicts the correct computational threshold[2] for tensor PCA (matching the best known poly-time algorithms), whereas existing methods based on AMP or free energy barriers capture a different threshold [RM14, LML+17, BGJ20] (see also [WEM19, BCRT20] for other ways to "redeem" the physics approach).

**One-sidedness of hardness criteria**    Finally, we note that the quantities $\|L\|_{\mathbb{Q}}^2$, $\mathrm{LD}(D)$, $\mathrm{FP}(D)$ should be thought of primarily as *lower bounds* that imply/suggest impossibility or hardness. If one of these quantities does *not* remain bounded as $n \to \infty$, it does not necessarily mean the problem is possible/tractable. We will revisit this issue again in Section 6, where a *conditional* low-degree calculation will be used to prove hardness even though the standard LD blows up (akin to the conditional versions of $\|L\|_{\mathbb{Q}}^2$ that are commonly used to prove information-theoretic lower bounds, e.g. [BMNN16, BMV+17, PWB16, PWBM18]).

## 4   The Gaussian Additive Model

We will for now focus on a particular class of estimation models, the so called *Gaussian additive models*. The distribution $\mathbb{P}_u$ describes an observation of the form

$$Y = \lambda u + Z \qquad (6)$$

where $\lambda \geq 0$ is the signal-to-noise ratio, $u \sim \mu$ is the signal of interest drawn from some distribution $\mu$ on $\mathbb{R}^N$, and $Z \sim \mathcal{N}(0, I_N)$ is standard Gaussian noise (independent from $u$). In recovery, the goal is to recover $u$, or more precisely to compute an estimator $\hat{u}(Y)$ that correlates with $u$. We note that in principle, $\lambda$ could be absorbed into the norm of $u$, but it will be convenient for us to keep $\lambda$ explicit because some of our results will involve perturbing $\lambda$ slightly. We focus on the hypothesis testing version of this question, where the goal is to distinguish a sample (6) from a standard Gaussian vector.

**Definition 4.1** (Gaussian Additive Model: Hypothesis Testing). Given $N$ a positive integer, $\lambda \geq 0$, and $\mu$ a distribution on $\mathbb{R}^N$ with all moments finite, let $Z \sim \mathcal{N}(0, I)$ and $u \sim \mu$; then hypothesis testing in the Gaussian additive model consists of performing a simple hypothesis test between

$$\mathbf{H_0} \qquad \mathbb{Q}: Y = Z \qquad \text{and} \qquad \mathbf{H_1} \qquad \mathbb{P}: Y = \lambda u + Z.$$

We are interested in understanding, as $N = N_n \to \infty$, for which prior distributions $\mu = \mu_n$ and SNR levels $\lambda = \lambda_n$ it is possible to computationally efficiently distinguish a sample from $\mathbb{P}$ from a sample from $\mathbb{Q}$ (in the sense of strong or weak detection; see Definition 2.2). As in Section 2 we use $\mathbb{P}_u$ to denote the distribution $\mathcal{N}(\lambda u, I)$, and write $L_u = \frac{d\mathbb{P}_u}{d\mathbb{Q}}$.

A number of classical inference tasks are captured by the Gaussian additive model, and bounds on $\mathrm{LD}(D)$ are known for many of them [HKP+17, KWB19, DKWB19, BBK+21], including matrix and tensor PCA.

**Example 4.2** (Matrix and Tensor PCA). In the matrix PCA case, we take $N = n^2$ and $u = x^{\otimes 2}$, where $x$ is for instance drawn with i.i.d. coordinates from some prior $\mu_0$. In the tensor case, we take $N = n^p$ and $u = x^{\otimes p}$, with $x$ drawn uniformly from the sphere $\mathbb{S}^{n-1}$.

---

[2]The detection and recovery thresholds are the same in this case.

**FP-LD Equivalence**    The following two theorems show that in the Gaussian additive model, FP and LD are equivalent up to logarithmic factors in $D$ and $1 + \varepsilon$ factors in $\lambda$. Recall the notation $\mathrm{LD}(D)$ and $\mathrm{FP}(D)$ from (1) and (3), here also with a dependency on the SNR $\lambda$.

**Theorem 4.3** (FP-hard implies LD-hard). *Assume the Gaussian additive model (Definition 4.1) and suppose $\|u\|^2 \leq M$ for all $u \in \mathrm{supp}(\mu)$, for some $M > 0$. Then for any $\lambda \geq 0$ and any odd integer $D \geq 1$, $\mathrm{LD}(D, \lambda) \leq \mathrm{FP}(\tilde{D}, \lambda) + e^{-D}$, where $\tilde{D} := D \cdot (2 + \log(1 + \lambda^2 M))$.*

We give the proof in the supplement. We are primarily interested in the regime $D = \omega(1)$, where we have shown that LD can only exceed FP by an additive $o(1)$ term. We have lost logarithmic factors in passing from $D$ to $\tilde{D}$. For many applications, these log factors are not an issue because (in the "hard" regime) FP is bounded for some $D = N^{\Omega(1)}$ while $\lambda, M$ are polynomial in $N$.

**Theorem 4.4** (LD-hard implies FP-hard). *Assume the Gaussian additive model (Definition 4.1). For all $\varepsilon \in (0, 1)$ there exists $D_0 = D_0(\varepsilon) > 0$ such that for any $\lambda \geq 0$ and any even integer $D \geq D_0$, if*

$$\mathrm{LD}(D, (1+\varepsilon)\lambda) \leq (1+\varepsilon)^D/(eD) \qquad then \qquad \mathrm{FP}(D, \lambda) \leq \mathrm{LD}(D, (1+\varepsilon)\lambda) + \varepsilon. \quad (7)$$

The proof, together with proofs for various corollaries, can be found in the supplementary material. In the asymptotic regime of primary interest, we have the following consequence.

**Corollary 4.5.** *Fix any constant $\varepsilon' > 0$ and suppose $D = D_n$, $\lambda = \lambda_n$, $N = N_n$, and $\mu = \mu_n$ are such that $D$ is an even integer, $D = \omega(1)$, and $\mathrm{LD}(D, (1+\varepsilon')\lambda) = O(1)$. Then $\mathrm{FP}(D, \lambda) \leq \mathrm{LD}(D, (1+\varepsilon')\lambda) + o(1)$.*

**Remark 4.6.** Above, we have taken the liberty to assume $D$ has a particular parity for convenience. Since FP and LD are both monotone in $D$, one can readily deduce similar results for all integers $D$.

## 5    FP-Hard Implies MCMC-Hard

In this section we show that in the Gaussian additive model, if FP is uniformly bounded then a natural class of local Markov chain Monte Carlo (MCMC) methods fail to recover the planted signal. Combining this with the results of the previous section, we also find that low-degree hardness, an impossibility result for polynomial ("algebraic") methods, implies MCMC-hardness, which is the failure of "geometric" methods. To the best of our knowledge this is the first time such a direct implication is established for any planted model. Note also that such a result is interesting as low-degree hardness is in principle an impossibility result for detection, while MCMC-hardness is a failure of some MCMC methods to perform estimation. We remark that the result of this section is similar in spirit to [BAJ18], which relates a version of annealed FP to the spectral gap for Langevin dynamics (albeit in a setting with no planted solution).

We again restrict our attention to the additive Gaussian model, now with the additional assumption that the prior $\mu$ is uniform on some finite set $S \subseteq \mathbb{R}^N$ with *transitive symmetry*, defined as follows.

**Definition 5.1.** We say $S \subseteq \mathbb{R}^N$ has *transitive symmetry* if for any $u, v \in S$ there exists an orthogonal matrix $R \in \mathrm{O}(N)$ such that $Ru = v$ and $RS = S$.

The assumption that $S$ is finite is not too restrictive because one can approximate any continuous prior to arbitrary accuracy by a discrete set (see the supplementary material for more details on that). Note that transitive symmetry implies that every $u \in S$ has the same 2-norm. Without loss of generality we will assume this 2-norm is 1, i.e., $S$ is a subset of the unit sphere $\mathbb{S}^{N-1}$.

Given an observation $Y = \lambda u + Z$ from the Gaussian additive model (with $\mu$ uniform on a finite, transitive-symmetric set $S \subseteq \mathbb{S}^{N-1}$), consider the associated *Gibbs measure* $\nu_\beta$ on $S$ defined by

$$\nu_\beta(v) = \frac{1}{\mathcal{Z}_\beta} \exp(-\beta H(v)) \quad (8)$$

where $\beta \geq 0$ is an inverse-temperature parameter (i.e., $1/\beta$ is the *temperature*), $H(v) = -\langle v, Y \rangle$ is the *Hamiltonian*, and $\mathcal{Z}_\beta = \sum_{v \in S} \exp(-\beta H(v))$ is the *partition function*. When $\beta = \lambda$ (the "Bayesian temperature"), the Gibbs measure $\nu_\beta$ is precisely the posterior distribution for the signal $u$ given the observation $Y = \lambda u + Z$.

We consider an arbitrary Markov chain $X_0, X_1, X_2, \ldots$ on state space $S$ with stationary distribution $\nu_\beta$ (for some $\beta$), i.e., if $X_t \sim \nu_\beta$ then $X_{t+1} \sim \nu_\beta$. We will assume a worst-case initial state, which

may depend adversarially on $Y$. We are interested in *hitting time lower bounds*, showing that such a Markov chain will take many steps before finding a "good" state that is close to the true signal $u$.

The core idea of our argument is to establish a *free energy barrier*, that is, a subset $B \subseteq S$ of small Gibbs mass that separates the initial state from the "good" states. Such a barrier is well-known to imply a lower bound for the hitting time of the "good" states using conductance; see e.g. [LP17, Theorem 7.4]. This is formalized in the proposition below.

**Proposition 5.2** (Free Energy Barrier Implies Hitting Time Lower Bound). *Suppose $X_0, X_1, X_2, \ldots$ is a Markov chain on a finite state space $S$, with some stationary distribution $\nu$. Let $A$ and $B$ be two disjoint subsets of $S$ and define the hitting time $\tau_B := \inf\{t \in \mathbb{N} : X_t \in B\}$. If the initial state $X_0$ is drawn from the conditional distribution $\nu|A$, then for any $t \in \mathbb{N}$, $\Pr(\tau_B \leq t) \leq t \cdot \frac{\nu(B)}{\nu(A)}$. In particular, for any $t \in \mathbb{N}$ there exists a $v \in A$ s.t. if $X_0 = v$ deterministically, then $\Pr(\tau_B \leq t) \leq t \cdot \frac{\nu(B)}{\nu(A)}$.*

We will need to impose some "locality" on our Markov chain so that it cannot jump from $A$ to a good state without first visiting $B$. We say a Markov chain on a finite state space $S \subseteq \mathbb{S}^{N-1}$ is $\Delta$-*local* if for every possible transition $v \to v'$ we have $\|v - v'\|_2 \leq \Delta$. We note that the use of local Markov chains is well motivated and preferred in theory and practice for the, in principle, low computation time for implementing a single step. For this reason, in most (discrete-state) cases the locality parameter $\Delta > 0$ is tuned so that the $\Delta$-neighborhood of each point is at most of polynomial size; see e.g. [Jer92, BWZ20, GZ19, CMZ22].

We now state the core result of this section, followed by various corollaries. (See the supplementary material for a slightly more general result and for the proof). The proof is based on an free energy barrier argument used in [BGJ20] which relies on rotational invariance of the Gaussian distribution.

**Theorem 5.3** (FP-Hard Implies Free Energy Barrier). *Let $\mu$ be the uniform measure on $S$, where $S \subseteq \mathbb{S}^{N-1}$ is a finite, transitive-symmetric set. The following holds for any $\varepsilon \in (0, 1/2)$, $D \geq 2$, $\lambda \geq 0$, and $\beta \geq 0$. Fix a ground-truth signal $u \in S$ and let $Y = \lambda u + Z$ with $Z \sim \mathcal{N}(0, I_N)$. Define $\delta = \delta(D)$ as in (3). Let $A = \{v \in S : |\langle u, v \rangle| \leq \delta\}$ and $B = \{v \in S : \langle u, v \rangle \in (\delta, (1 + \varepsilon)\delta]\}$. With probability at least $1 - e^{-\varepsilon D}$ over $Z$, the Gibbs measure (8) associated to $Y$ satisfies that $\frac{\nu_\beta(B)}{\nu_\beta(A)} \leq 2 \left( 2 \cdot \mathrm{FP}(D + \log 2, \tilde{\lambda}) \right)^{1-2\varepsilon} e^{-\varepsilon D}$ where $\tilde{\lambda} := \sqrt{\beta \lambda \cdot \frac{2+\varepsilon}{1-2\varepsilon}}$.*

As mentioned above, one particularly natural choice of $\beta$ is the Bayesian temperature $\lambda$ (which corresponds to sampling from the posterior distribution). By combining Proposition 5.2 and Theorem 5.3 we conclude that FP-hard implies that the hitting time grows exponential with $D$, which we simply refer to as "MCMC-hard". Combining now this result with the FP and LD equivalence part described in Corollary 4.5, we conclude also that LD-hard implies MCMC-hard.

**Corollary 5.4** (FP-Hard (and LD-hard) Implies Hitting Time Lower Bound). *In the setting of Theorem 5.3, suppose $X_0, X_1, X_2, \ldots$ is a $\Delta$-local Markov chain with state space $S$ and stationary distribution $\nu_\beta$, for some $\Delta \leq \varepsilon\delta$. Define the hitting time $\tau := \inf\{t \in \mathbb{N} : \langle u, X_t \rangle > \delta\}$. With probability at least $1 - e^{-\varepsilon D}$ over $Z$, there exists a state $v \in A$ such that for the initialization $X_0 = v$, with probability at least $1 - e^{-\varepsilon D/2}$ over the Markov chain,*

$$\tau \geq \frac{e^{\varepsilon D/2}}{2 \left( 2 \cdot \mathrm{FP}(D + \log 2, \tilde{\lambda}) \right)^{1-2\varepsilon}}.$$

*If furthermore $D = D_n$ is a sequence with $D = \omega(1)$ such that $D + \log 2$ is an even integer $\mathrm{LD}(D + \log 2, (1 + \varepsilon)\tilde{\lambda}) \leq B$ for a constant $B > 0$, then there is a constant $C = C(B, \epsilon) > 0$ for which $\tau \geq C(B, \varepsilon)e^{\varepsilon D/2}$.*

We note that Corollary 5.4 applies for all $\Delta \leq \varepsilon\delta(D)$. For various models of interest, we note that the range $\Delta \leq \varepsilon\delta(D)$ for the locality parameter $\Delta$ contains the "reasonable" range of values where the $\Delta$-neighborhoods are of polynomial size. For example, let us focus on the well-studied tensor PCA setting with a Rademacher signal and even tensor power, that is $S = \{u = x^{\otimes 2p} : x \in \{-n^{-p}, n^{-p}\}^n\}$. Then for any $D > 0$ and any $\varepsilon > 0$, we have $\epsilon\delta(D) = \Omega(n^{-p})$ and therefore the $\varepsilon\delta(D)$-neighborhood of any point $u = x^{\otimes 2p} \in S$, contains all $n^{\omega(1)}$ points $u = y^{\otimes 2p} \in S$ where $y$ has any Hamming distance $o(\sqrt{n})$ from $x$.

**Remark 5.5.** We note that the original work of [BGJ20], on which the proof of Theorem 5.3 is based, showed failure of MCMC methods in a strictly *larger* (by a power of $n$) range of $\lambda$ than the

low-degree-hard regime for the tensor PCA problem. In contrast, our MCMC lower bound uses the same argument but matches the low-degree threshold (at the Bayesian temperature). This is because [BGJ20] only considers temperatures that are well above the Bayesian one: in their notation, their result is for constant $\beta$ whereas the Bayesian $\beta$ grows with $n$.

## 6 Sparse Linear Regression

In this section we prove sharp low-degree lower bounds for the hypothesis testing version of a classical inference problem: sparse linear regression. This is significant for a number of reasons: (i) we uncover a new computational phase transition in sparse regression, (ii) our proof involves upper-bounding LD in terms of FP, illustrating that an FP-to-LD connection can be achieved outside the Gaussian additive model, and (iii) the proof requires a first-of-its-kind *conditional* low-degree calculation which seems difficult to carry out directly without using FP, illustrating that FP can be a powerful tool for proving low-degree lower bounds. We consider the following detection task.

**Definition 6.1** (Sparse Linear Regression: Hypothesis Testing). Given a sample size $m \in \mathbb{N}$, feature size $n \in \mathbb{N}$, sparsity level $k \in \mathbb{N}$ with $k \leq n$ and noise level $\sigma > 0$, we consider hypothesis testing between the following two distributions over $(X, Y) \in \mathbb{R}^{m \times n} \times \mathbb{R}^m$.

- $\mathbb{Q}$ generates a pair $(X, Y)$ where both $X$ and $Y$ have i.i.d. $\mathcal{N}(0, 1)$ entries.

- $\mathbb{P}$ generates a pair $(X, Y)$ as follows. First $X$ is drawn with i.i.d. $\mathcal{N}(0, 1)$ entries. Then, for a planted signal $u \in \{0, 1\}^n$ drawn uniformly from all binary vectors of sparsity exactly $k$, and independent noise $W \sim \mathcal{N}(0, \sigma^2 I_m)$, we set $Y = (k + \sigma^2)^{-1/2}(Xu + W)$. This normalization ensures $Y \sim \mathcal{N}(0, I_m)$.

In the following result, we provide rigorous "low-degree evidence" (and a matching upper bound) for the precise threshold separating the "easy" and "hard" regimes.

**Theorem 6.2.** *For $\theta \in (0, 1)$, define*

$$R_{\mathrm{LD}}(\theta) = \frac{2(1-\sqrt{\theta})}{1+\sqrt{\theta}} \cdot \mathbb{1}[0 < \theta < \tfrac{1}{4}] + \frac{1-2\theta}{1-\theta} \cdot \mathbb{1}[\tfrac{1}{4} \leq \theta < \tfrac{1}{2}]. \tag{9}$$

*Consider sparse linear regression (Definition 6.1) in the following scaling regime: for fixed constants $\theta \in (0, 1)$ and $R > 0$, take $n \to \infty$ with $k = n^{\theta + o(1)}$, $\sigma^2 = o(k)$, and $m = (1 + o(1))Rk \log(n/k)$.*

- *(a) (Hard regime) If $R < R_{\mathrm{LD}}(\theta)$ then no degree-$o(k)$ polynomial $f : \mathbb{R}^{m(n+1)} \to \mathbb{R}$ achieves the following notion of "strong separation":*

$$\sqrt{\max\{\mathrm{Var}_{\mathbb{Q}}[f], \mathrm{Var}_{\mathbb{P}}[f]\}} = o\left(|\mathbb{E}_{\mathbb{P}}[f] - \mathbb{E}_{\mathbb{Q}}[f]|\right). \tag{10}$$

- *(b) (Easy regime) If $R > R_{\mathrm{LD}}(\theta)$ then there is a polynomial-time algorithm for strong detection between $\mathbb{P}$ and $\mathbb{Q}$ (see Definition 2.2).*

In line with existing low-degree lower bounds in the literature, the notion of strong separation ruled out in part (a) is a natural notion of success for polynomial-based tests: by Chebyshev's inequality, (10) implies that strong detection is possible by thresholding $f$. Part (a) is interpreted as evidence that when $R < R_{\mathrm{LD}}(\theta)$, strong detection requires runtime $\exp(\tilde{\Omega}(k))$. This is tight, matching the runtime of a brute-force search algorithm, which succeeds for any constants $\theta \in (0, 1)$ and $R > 0$ (i.e., the problem is always information-theoretically possible in our regime) [RXZ21].

The algorithm that gives the matching upper bound in part (b) is fairly simple but somewhat subtle: letting $X_j$ denote the $j$th column of $X$, the idea is to count the number of indices $j \in [n]$ for which $\langle X_j, Y \rangle / \|Y\|_2$ exceeds a particular (carefully chosen) threshold, and then threshold this count.

We note that prior work has also considered the related task of recovering $u$, given $(X, Y) \sim \mathbb{P}$ (e.g. [Wai09, DT10, GZ22, RXZ19, RXZ21]). Our result gives evidence that the existing algorithms for recovery cannot be improved in the regime $\theta \downarrow 0$. See the supplement for further discussion.

**Proof techniques and beyond sparse regression** Proving the sharp low-degree lower bound in the regime $\theta < 1/4$ requires a *conditional* low-degree calculation: instead of bounding LD for testing $\mathbb{P}$ versus $\mathbb{Q}$, we bound LD for testing the conditional distribution $\mathbb{P}|A$ versus $\mathbb{Q}$, for some

high-probability event $A$. This is necessary because the standard LD blows up at a sub-optimal threshold due to a rare "bad" event under $\mathbb{P}$. Conditioning arguments of this type are common for information-theoretic lower bounds (see e.g. [BMNN16, BMV$^+$17, PWB16, PWBM18]), but this is (to our knowledge) the first instance where conditioning has been needed for a low-degree lower bound. We note that [Arp21] gave low-degree lower bounds for sparse regression by analyzing the standard (non-conditioned) LD, and our result improves the threshold when $\theta < 1/4$ via conditioning.

The standard approach to bounding LD involves direct moment calculations (see e.g. Section 2.4 of [Hop18]), but it seems difficult to carry out our conditional low-degree calculation by this approach. Instead, we prove that the conditional LD can be upper-bounded by the corresponding conditional FP, which is more tractable to bound directly. This illustrates that FP can be a powerful tool for proving low-degree lower bounds that may otherwise be out of reach. As we discuss in the supplement, the FP-to-LD connection we establish here holds for a broad class of problems with planted sparse signals, including planted clique, planted dense subgraph, and any sparse generalized linear model.

## 7   Conclusion

We have introduced the *Franz-Parisi criterion* and have shown that it implies lower bounds against local MCMC algorithms and is equivalent to low-degree hardness in Gaussian additive models. We have also used the FP criterion as a tool to prove new low-degree lower bounds for sparse regression. This is a first step in relating notions of computational intractability based on free energy potentials to lower bounds within restricted models of computation, such as the sum-of-squares hierarchy. A clear question in the wake of our work is whether similar connections can be drawn for a broader class of planted models. In the supplementary material, we demonstrate that this cannot be done in a straightforward way, by building counterexamples (namely, certain simple mixtures over Boolean product measures) where the Franz-Parisi criterion is not consistent with computational intractability. Nonetheless, given the impressive track-record of accuracy of free-energy based computational threshold predictions, we believe this to be a shortcoming of our specific criterion, and the development of one for a more general class of problems is a fascinating direction for future research.

## Acknowledgments and Disclosure of Funding

AEA: Part of this work was done while this author was supported by the Richard M. Karp Fellowship at the Simons Institute for the Theory of Computing (Program on Probability, Geometry and Computation in High Dimensions). This author is grateful to Florent Krzakala for introducing him to the work of Franz and Parisi.

SBH: Parts of this work were done while this author was supported by a Microsoft Research PhD Fellowship, by a Miller Postdoctoral Fellowship, and by the Microsoft Fellowship at the Simons Institute for the Theory of Computing.

ASW: Part of this work was done at Georgia Tech, supported by NSF grants CCF-2007443 and CCF-2106444. Part of this work was done while visiting the Simons Institute for the Theory of Computing. Part of this work was done while with the Courant Institute at NYU, partially supported by NSF grant DMS-1712730 and by the Simons Collaboration on Algorithms and Geometry.

IZ: Supported by the Simons-NSF grant DMS-2031883 on the Theoretical Foundations of Deep Learning and the Vannevar Bush Faculty Fellowship ONR-N00014-20-1-2826. Part of this work was done while visiting the Simons Institute for the Theory of Computing. Part of this work was done while with the Center for Data Science at NYU, supported by a Moore-Sloan CDS postdoctoral fellowship.

The authors thank Cris Moore and the Santa Fe Institute for hosting the 2018 "Santa Fe Workshop on Limits to Inference in Networks and Noisy Data," where the initial ideas in this paper were formulated. The authors thank Aukosh Jagannath and anonymous reviewers for helpful comments on an earlier version of this work.

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
