# THE FRANZ–PARISI CRITERION AND COMPUTATIONAL TRADE-OFFS IN HIGH DIMENSIONAL STATISTICS

AFONSO S. BANDEIRA, AHMED EL ALAOUI, SAMUEL B. HOPKINS, TSELIL SCHRAMM, ALEXANDER S. WEIN, AND ILIAS ZADIK

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

likelihood ratio" or "low-degree hard," a term which we will define precisely below. Many of the problems mentioned above fall into this universality class precisely in the regime of their information-computation gaps.

Another successful approach to understand computational hardness of statistical problems borrows tools from statistical physics: tools such as the *cavity method* and *replica method* can be used to make remarkably precise predictions of both statistical and computational thresholds, essentially by studying properties of free energy potentials associated to the problem in question, or by studying related iterative algorithms such as belief propagation or approximate message passing (see e.g. [DMM09, DKMZ11, LKZ15a, LKZ15b, DMK+16]). We will discuss some of these ideas further in Section 1.3.

**Main Contributions.** This paper aims to make a rigorous connection between the low-degree and free-energy based approaches in the setting of statistical inference. (We note that this setting differs from that of random optimization problems with no planted signal, where a connection of this nature has already been established [GJW20, Wei22, BH21].) We start by defining a free-energy based criterion, the *Franz–Parisi criterion* (Definition 1.5) inspired by the so-called Franz–Parisi potential [FP95] (see Section 1.3 for more on the connection with statistical physics). We formally connect this criterion to low-degree hardness for a broad class of statistical problems, namely all Gaussian additive models (Theorems 2.4 and 2.5) and certain sparse planted models (Theorem 3.7). By leveraging these rigorous connections we are able to (i) establish that in the context of Gaussian additive models, low-degree hardness implies failure of local MCMC algorithms (Corollary 2.18), and (ii) provide new low-degree lower bounds for sparse linear regression which seem difficult to prove directly (Theorem 3.10). We also include some examples that illustrate that this equivalence between different forms of hardness does not hold for all inference problems (see Section 4), leaving as an exciting future direction the problem of determining under which conditions it does hold, and investigating what other free-energy based criteria may be more suitable in other inference problems.

1.1. **Setting and Definitions.** We will focus on problems in which there is a signal vector of interest $u \in \mathbb{R}^n$, drawn from a prior distribution $\mu$ over such signals, and the data observed is a sample from a distribution $\mathbb{P}_u$ on $\mathbb{R}^N$ that depends on the signal $u$. One natural problem in this setting is *estimation*: given a sample from $\mathbb{P}_u$ with the promise that $u \sim \mu$, the goal is to estimate $u$ (different estimation error targets correspond to different versions of this problem, often referred to *weak/approximate recovery* or *exact recovery*). This roughly corresponds to the "search" version of the problem, but just as in classical complexity theory, it is productive to instead study a "decision" version of the problem, *hypothesis testing*: we are given a sample generated either from the "planted" distribution $\mathbb{P} = \mathbb{E}_{u \sim \mu} \mathbb{P}_u$ (a mixture model were the data is drawn from $\mathbb{P}_u$ for a random $u \sim \mu$) or from a "null" reference distribution $\mathbb{Q}$ representing pure noise, and the goal is to decide whether it is more likely that the sample came from $\mathbb{P}$ or $\mathbb{Q}$.

**Problem 1.1** (High Dimensional Inference: Hypothesis Testing). Given positive integers $n, N$, a distribution $\mu$ on $\mathbb{R}^n$, and a distribution $\mathbb{P}_u$ on $\mathbb{R}^N$ for each $u \in \text{supp}(\mu)$, the goal is to perform simple hypothesis testing between

$$\mathbf{H_0}: \quad Y \sim \mathbb{Q} \qquad\qquad\qquad\qquad \text{(Null model)},$$
$$\mathbf{H_1}: \quad Y \sim \mathbb{P} = \mathop{\mathbb{E}}_{u \sim \mu} \mathbb{P}_u \qquad\qquad\qquad \text{(Planted model)}.$$

We will be especially interested in asymptotic settings where $n \to \infty$ and the other parameters scale with $n$ in some prescribed way: $N = N_n$, $\mu = \mu_n$, $\mathbb{P} = \mathbb{P}_n$, $\mathbb{Q} = \mathbb{Q}_n$. In this setting, we focus on the following two objectives.

**Definition 1.2** (Strong/Weak Detection).

- **Strong detection**: we say *strong detection* is achieved if the sum of type I and type II errors[1] tends to 0 as $n \to \infty$.
- **Weak detection**: we say *weak detection* is achieved if the sum of type I and type II errors is at most $1 - \varepsilon$ for some fixed $\varepsilon > 0$ (not depending on $n$).

---

[1]Type I error is the probability of outputting "$\mathbb{P}$" when given a sample from $\mathbb{Q}$. Type II error is the probability of outputting "$\mathbb{Q}$" when given a sample from $\mathbb{P}$.

In other words, strong detection means the test succeeds with high probability, while weak detection means the test has some non-trivial advantage over random guessing.

While our main focus will be on the testing problem, we remark that computational hardness of strong detection often implies that estimating $u$ is hard as well.[2]

Throughout, we will work in the Hilbert space $L^2(\mathbb{Q})$ of (square integrable) functions $\mathbb{R}^N \to \mathbb{R}$ with inner product $\langle f, g \rangle_{\mathbb{Q}} := \mathbb{E}_{Y \sim \mathbb{Q}}[f(Y)g(Y)]$ and corresponding norm $\|f\|_{\mathbb{Q}} := \langle f, f \rangle_{\mathbb{Q}}^{1/2}$. For a function $f : \mathbb{R}^N \to \mathbb{R}$ and integer $D \in \mathbb{N}$, we let $f^{\leq D}$ denote the orthogonal (w.r.t. $\langle \cdot, \cdot \rangle_{\mathbb{Q}}$) projection of $f$ onto the subspace of polynomials of degree at most $D$. We will assume that $\mathbb{P}_u$ is absolutely continuous with respect to $\mathbb{Q}$ for all $u \in \text{supp}(\mu)$, use $L_u := \frac{\mathrm{d}\mathbb{P}_u}{\mathrm{d}\mathbb{Q}}$ to denote the likelihood ratio, and assume that $L_u \in L^2(\mathbb{Q})$ for all $u \in \text{supp}(\mu)$. The likelihood ratio between $\mathbb{P}$ and $\mathbb{Q}$ is denoted by $L := \frac{\mathrm{d}\mathbb{P}}{\mathrm{d}\mathbb{Q}} = \mathbb{E}_{u \sim \mu} L_u$.

A key quantity of interest is the (squared) norm of the likelihood ratio, which is related to the *chi-squared divergence* $\chi^2(\mathbb{P} \,\|\, \mathbb{Q})$ as

$$\|L\|_{\mathbb{Q}}^2 = \left\| \mathbb{E}_{u \sim \mu} L_u \right\|_{\mathbb{Q}}^2 = \chi^2(\mathbb{P} \,\|\, \mathbb{Q}) + 1 \,.$$

This quantity has the following standard implications for *information-theoretic* impossibility of testing, in the asymptotic regime $n \to \infty$. The proofs can be found in e.g. [MRZ15, Lemma 2].

- If $\|L\|_{\mathbb{Q}}^2 = O(1)$ (equivalently, $\limsup_{n \to \infty} \|L\|_{\mathbb{Q}}^2 < \infty$) then strong detection is impossible. (This is a classical second moment method associated with Le Cam's notion of *contiguity* [Le 60].)
- If $\|L\|_{\mathbb{Q}}^2 = 1 + o(1)$ (equivalently, $\lim_{n \to \infty} \|L\|_{\mathbb{Q}}^2 = 1$) then weak detection is impossible. (It is always true that $\|L\|_{\mathbb{Q}}^2 \geq 1$, by Jensen's inequality and the fact $\mathbb{E}_{Y \sim \mathbb{Q}} L(Y) = 1$.)

We will study two different "predictors" of computational complexity of hypothesis testing, both of which can be seen as different "restrictions" of $\|L\|_{\mathbb{Q}}^2$. Our first predictor is based on the *low-degree likelihood ratio* $L^{\leq D}$, which recall means the projection of the likelihood ratio onto the subspace of degree-at-most-$D$ polynomials. This is already a well-established framework for computational lower bounds [HS17, HKP$^+$17, Hop18] (we point the reader to the thesis [Hop18] or the survey [KWB19] for a pedagogical exposition).

**Definition 1.3** (Low-Degree Likelihood Ratio). Define the squared norm of the degree-$D$ likelihood ratio (also called the "low-degree likelihood ratio") to be the quantity

$$(1) \qquad \text{LD}(D) := \|L^{\leq D}\|_{\mathbb{Q}}^2 = \left\| \left( \mathbb{E}_{u \sim \mu} L_u \right)^{\leq D} \right\|_{\mathbb{Q}}^2 = \mathbb{E}_{u, v \sim \mu} \left[ \langle L_u^{\leq D}, L_v^{\leq D} \rangle_{\mathbb{Q}} \right] \,,$$

where the last equality follows from linearity of the projection operator, and where $u, v$ are drawn independently from $\mu$. For some increasing sequence $D = D_n$, we say that the hypothesis testing problem above is *hard for the degree-$D$ likelihood* or simply *low-degree hard* if $\text{LD}(D) = O(1)$.

Heuristically speaking, the interpretation of $\text{LD}(D)$ should be thought of as analogous to that of $\|L\|_{\mathbb{Q}}^2$ but for computationally-bounded tests: if $\text{LD}(D) = O(1)$ this suggests computational hardness of strong detection, and if $\text{LD}(D) = 1 + o(1)$ this suggests computational hardness of weak detection (it is always the case that $\text{LD}(D) \geq 1$; see (31).). The parameter $D = D_n$ should be loosely thought of as a proxy for the runtime allowed for our testing algorithm, where $D = O(\log n)$ corresponds to polynomial time and more generally, larger values of $D$ correspond to runtime $\exp(\tilde{\Theta}(D))$ where $\tilde{\Theta}$ hides factors of $\log n$ (or equivalently, $\log N$, since we will always take $N$ and $n$ to be polynomially-related). In Section 1.2 we further discuss the significance of low-degree hardness, including its formal implications for failure of certain tests based on degree-$D$ polynomials, as well as the more conjectural connection to the sum-of-squares hierarchy.

We now introduce our second predictor, which we call the *Franz–Parisi criterion*. On a conceptual level, it is inspired by well-established ideas rooted in statistical physics, which we discuss further in Section 1.3.

---

[2]There is no formal reduction from hypothesis testing to estimation at this level of generality (see Section 3.4 of [BMV$^+$17] for a pathological counterexample) but it is typically straightforward to give such a reduction for the types of testing problems we will consider in this paper (see e.g. Section 5.1 of [MW15]). On the other hand, estimation can sometimes be strictly harder than the associated testing problem (see e.g. [SW20]).

However, the precise definition we use here has not appeared before (to our knowledge). Throughout this paper we will argue for the significance of this definition in a number of ways: its conceptual link to physics (Section 1.3), its provable equivalence to the low-degree criterion for Gaussian additive models (Section 2.1), its formal connection to MCMC methods for Gaussian additive models (Section 2.2), and its usefulness as a tool for proving low-degree lower bounds (Section 3.2).

**Definition 1.4** (Low-Overlap Likelihood Norm). We define the *low-overlap likelihood norm* at overlap $\delta \geq 0$ as

$$\text{LO}(\delta) := \mathop{\mathbb{E}}_{u,v \sim \mu} \left[ \mathbb{1}_{|\langle u,v \rangle| \leq \delta} \cdot \langle L_u, L_v \rangle_{\mathbb{Q}} \right], \tag{2}$$

where $u, v$ are drawn independently from $\mu$.

**Definition 1.5** (Franz–Parisi Criterion). We define the *Franz–Parisi Criterion at D deviations* to be the quantity

$$\text{FP}(D) := \text{LO}(\delta), \quad \text{for } \delta = \delta(D) := \sup \{ \varepsilon \geq 0 \text{ s.t. } \mathop{\Pr}_{u,v \sim \mu} (|\langle u,v \rangle| \geq \varepsilon) \geq e^{-D} \}. \tag{3}$$

For some increasing sequence $D = D_n$, we say a problem is *FP-hard at D deviations* if $\text{FP}(D) = O(1)$.

**Remark 1.6.** Two basic properties of the quantity $\delta$ defined in (3) are $\Pr(|\langle u,v \rangle| \geq \delta) \geq e^{-D}$ (in particular, the supremum in (3) is attained) and $\Pr(|\langle u,v \rangle| > \delta) \leq e^{-D}$. These follow from continuity of measure and are proved in Section 5.1.

**Remark 1.7.** To obtain a sense of the order of magnitudes, let us assume that the product $\langle u,v \rangle$ is centered and sub-Gaussian with parameter $\sigma^2 n$. This is for instance the case if the prior distribution is a product measure: $\mu = \mu_0^n$, and the distribution of the product $xx'$ of two independent samples $x, x' \sim \mu_0$ is sub-Gaussian with parameter $\sigma^2$. Then $\Pr(|\langle u,v \rangle| \geq \delta) \leq 2e^{-\delta^2/(2n\sigma^2)}$ for all $\delta$, so $\delta(D)^2 \leq 2n\sigma^2(D + \log 2)$.

Heuristically speaking, $\text{FP}(D)$ should be thought of as having a similar interpretation as $\text{LD}(D)$: if $\text{FP}(D) = O(1)$ this suggests hardness of strong detection, and if $\text{FP}(D) = 1 + o(1)$ this suggests hardness of weak detection. The parameter $D$ is a proxy for runtime and corresponds to the parameter $D$ in $\text{LD}(D)$, as we justify in Section 1.3.

We remark that $\text{LD}(D)$ and $\text{FP}(D)$ can be thought of as different ways of "restricting" the quantity

$$\|L\|_{\mathbb{Q}}^2 = \mathop{\mathbb{E}}_{u,v \sim \mu} [\langle L_u, L_v \rangle_{\mathbb{Q}}], \tag{4}$$

which recall is related to information-theoretic *impossibility* of testing. For LD, the restriction takes the form of low-degree projection on each $L_u$, while for FP it takes the form of excluding pairs $(u,v)$ of high overlap. Our results will show that (in some settings) these two types of restriction are nearly equivalent.

Finally, we note that the quantities $\|L\|_{\mathbb{Q}}^2$, $\text{LD}(D)$, $\text{FP}(D)$ should be thought of primarily as *lower bounds* that imply/suggest impossibility or hardness. If one of these quantities does *not* remain bounded as $n \to \infty$, it does not necessarily mean the problem is possible/tractable. We will revisit this issue again in Section 3.2, where a *conditional* low-degree calculation will be used to prove hardness even though the standard LD blows up (akin to the conditional versions of $\|L\|_{\mathbb{Q}}^2$ that are commonly used to prove information-theoretic lower bounds, e.g. [BMNN16, BMV+17, PWB16, PWBM18]).

1.2. **Relation of LD to Low-Degree Algorithms.** We now give a brief overview of why the low-degree likelihood ratio is meaningful as a predictor of computational hardness, referring the reader to [Hop18, KWB19] for further discussion. Notably, bounds on $\text{LD}(D)$ imply failure of tests based on degree-$D$ polynomials in the following specific sense.

**Definition 1.8** (Strong/Weak Separation). For a polynomial $f : \mathbb{R}^N \to \mathbb{R}$ and two distributions $\mathbb{P}, \mathbb{Q}$ on $\mathbb{R}^N$ (where $N, f, \mathbb{P}, \mathbb{Q}$ may all depend on $n$),

- we say $f$ *strongly separates* $\mathbb{P}$ and $\mathbb{Q}$ if, as $n \to \infty$,

$$\sqrt{\max \{ \text{Var}_{\mathbb{P}}[f], \text{Var}_{\mathbb{Q}}[f] \}} = o \left( \left| \mathop{\mathbb{E}}_{\mathbb{P}}[f] - \mathop{\mathbb{E}}_{\mathbb{Q}}[f] \right| \right),$$

- we say $f$ *weakly separates* $\mathbb{P}$ and $\mathbb{Q}$ if, as $n \to \infty$,

$$\sqrt{\max\left\{\mathrm{Var}_{\mathbb{P}}[f], \mathrm{Var}_{\mathbb{Q}}[f]\right\}} = O\left(\left|\mathop{\mathbb{E}}_{\mathbb{P}}[f] - \mathop{\mathbb{E}}_{\mathbb{Q}}[f]\right|\right).$$

These are natural *sufficient* conditions for strong and weak detection, respectively: strong separation implies (by Chebyshev's inequality) that strong detection is achievable by thresholding $f$, and weak separation implies that weak detection is possible using the value of $f$ (see Proposition 6.1). The quantity $\mathrm{LD}(D)$ can be used to formally rule out such low-degree tests. Namely, Proposition 6.2 implies that, for any $D = D_n$,

- if $\mathrm{LD}(D) = O(1)$ then no degree-$D$ polynomial strongly separates $\mathbb{P}$ and $\mathbb{Q}$;
- if $\mathrm{LD}(D) = 1 + o(1)$ then no degree-$D$ polynomial weakly separates $\mathbb{P}$ and $\mathbb{Q}$.

While one can think of $\mathrm{LD}(D)$ as simply a tool for rigorously ruling out certain polynomial-based tests as above, it is productive to consider the following heuristic correspondence between polynomial degree and runtime.

- We expect the class of degree-$D$ polynomials to be as powerful as all $\exp(\tilde{\Theta}(D))$-time tests (which is the runtime needed to naively evaluate the polynomial term-by-term). Thus, if $\mathrm{LD}(D) = O(1)$ (or $1 + o(1)$), we take this as evidence that strong (or weak, respectively) detection requires runtime $\exp(\tilde{\Omega}(D))$; see Hypothesis 2.1.5 of [Hop18].
- On a finer scale, we expect the class of degree-$O(\log n)$ polynomials to be at least as powerful as all polynomial-time tests. This is because it is typical for the best known efficient test to be implementable as a spectral method and computed as an $O(\log n)$-degree polynomial using power iteration on some matrix; see e.g. Section 4.2.3 of [KWB19]. Thus, if $\mathrm{LD}(D) = O(1)$ (or $1 + o(1)$) for some $D = \omega(\log n)$, we take this as evidence that strong (or weak, respectively) detection cannot be achieved in polynomial time; see Conjecture 2.2.4 of [Hop18].

We emphasize that the above statements are not true in general (see for instance [HW21, Kun21, KM21, ZSWB21, DK21] for some discussion of counterexamples) and depend on the choice of $\mathbb{P}$ and $\mathbb{Q}$, yet remarkably often appear to hold up for a broad class of distributions arising in high-dimensional statistics.

*Low-degree hardness and sum-of-squares algorithms.* An intriguing conjecture posits that $\mathrm{LD}(D)$ characterizes the limitations of algorithms for hypothesis testing in the powerful sum-of-squares (SoS) hierarchy. In particular, when $\mathrm{LD}(D) = 1 + o(1)$, this implies a canonical construction of a candidate sum-of-squares lower bound via a process called pseudo-calibration. We refer the reader to [Hop18] for a thorough discussion on connections between LD and SoS via pseudo-calibration, and [HKP$^+$17, KWB19] for connections through spectral algorithms.

## 1.3. **Relation of FP to Statistical Physics.**

1.3.1. *The Franz–Parisi potential.* Our definition of the low-overlap likelihood norm has a close connection to the Franz–Parisi potential in the statistical physics of glasses [FP95]. We explain here the connection, together with a simple derivation of the formula in Eq. (2). For additional background, we refer the reader to [MM09, ZK16b] for exposition on the well-explored connections between statistical physics and Bayesian inference.

Given a system with (random) Hamiltonian $H : \mathbb{R}^n \to \mathbb{R}$, whose configurations are denoted by vectors $u \in \mathbb{R}^n$, we let $u_0$ be a reference configuration, and consider the free energy of a new configuration $u$ drawn from the Gibbs measure $\mathrm{d}\nu_\beta(u) = Z^{-1} \exp(-\beta H(u)) \mathrm{d}\mu(u)$ constrained to be at a fixed distance $r$ from $u_0$. This free energy is

$$(5) \qquad\qquad F_\beta(u_0, r) = \log \int \mathbb{1}_{d(u, u_0) = r}\, e^{-\beta H(u)} \mathrm{d}\mu(u).$$

(For simplicity, we proceed with equalities inside the indicator in this discussion; but for the above to be meaningful, unless the prior is supported on a discrete set, one has to consider events of the form

$d(u, u_0) \in (r - \delta, r + \delta)$.) The Franz–Parisi potential is the average of the above free energy when $u_0$ is drawn from the Gibbs measure $\nu_{\beta'}$, at a possibly different temperature $\beta'$:

$$(6) \qquad f_{\beta, \beta'}(r) := \mathbb{E}\Big[\mathbb{E}_{u_0 \sim \nu_{\beta'}}\big[F_\beta(u_0, r)\big]\Big],$$

where the outer expectation is with respect to the randomness (or disorder) of the Hamiltonian $H$. This potential contains information about the free energy landscape of $\nu_\beta$ seen locally from a reference configuration $u_0$ 'equilibrated' at temperature $\beta'$, and allows to probe the large deviation properties of $\nu_\beta$ as one changes $\beta$. The appearance of local maxima separated by 'free energy barriers' in this potential is interpreted as a sign of appearance of 'metastable states' trapping the Langevin or Glauber dynamics, when initialized from a configuration at equilibrium at temperature $\beta'$, for long periods of time. This observation, which is reminiscent of standard 'bottleneck' arguments for Markov chains [LP17], has been made rigorous in some cases; see for instance [BGJ20, BWZ20].

In a statistical context, the Gibbs measure $\nu_\beta$ corresponds to the posterior measure of the signal vector $u$ given the observations $Y$, and $\beta$ plays the role of the signal-to-noise ratio. Using Bayes' rule we can write

$$\mathrm{d}\nu_\beta(u) = \frac{\mathrm{d}\mathbb{P}_u}{\mathrm{d}\mathbb{P}}(Y)\,\mathrm{d}\mu(u) = \frac{L_u(Y)}{L(Y)}\,\mathrm{d}\mu(u)\,.$$

From the above formula we make the correspondence $L(Y) = Z$, $L_u(Y) = e^{-\beta H(u)}$, and $Y$ is the source of randomness of $H$. Letting $\beta' = \beta$, and then omitting the temperatures from our notation, the FP potential (6) becomes

$$f(r) = \mathbb{E}_{Y \sim \mathbb{P}}\mathbb{E}_{u_0 \sim \mathbb{P}(\cdot|Y)} \log \mathbb{E}_{u \sim \mu}\big[\mathbb{1}_{d(u, u_0) = r}\, L_u(Y)\big]$$
$$(7) \qquad = \mathbb{E}_{u_0 \sim \mu}\mathbb{E}_{Y \sim \mathbb{P}_{u_0}} \log \mathbb{E}_{u \sim \mu}\big[\mathbb{1}_{d(u, u_0) = r}\, L_u(Y)\big]$$

where the second line follows from Bayes' rule. It is in general extremely difficult to compute the exact asymptotics of the FP potential $f$, save for the simplest models[3]. Physicists have used the replica method together with structural assumptions about the Gibbs measure to produce approximations of this potential, which are then used in lieu of the true potential [FP95, FP98]. One such approximation is given by the so-called *replica-symmetric* potential which describes the behavior of *Approximate Message Passing (AMP)* algorithms; see for instance [LKZ15a, LKZ15b, DMK+16, BPW18, AK18] (for an illustration, see Figure 1 of [DMK+16] or Figure 1 of [BPW18]). But the simplest approximation is the *annealed approximation* which can be obtained by Jensen's inequality: $f(r) \leq f^{\mathrm{ann}}(r)$, where

$$f^{\mathrm{ann}}(r) := \log \mathbb{E}_{u, u_0 \sim \mu}\mathbb{E}_{Y \sim \mathbb{P}_{u_0}}\big[\mathbb{1}_{d(u, u_0) = r}\, L_u(Y)\big]$$
$$(8) \qquad = \log \mathbb{E}_{u, u_0 \sim \mu}\big[\mathbb{1}_{d(u, u_0) = r}\, \langle L_u, L_{u_0}\rangle_{\mathbb{Q}}\big]\,.$$

(We note that our notion of annealed FP is not quite the same as the one in [FP98].) We see a similarity between Eq. (8) and our Low-Overlap Likelihood Norm, Eq. (2), where the distance has been replaced by the inner product, or overlap of $u$ and $u_0$. This parametrization turns out to be convenient in the treatment of Gaussian models, as we do in this paper[4].

In many scenarios of interest, the annealed potential has the same qualitative properties as the quenched potential. We consider below the example of the spiked Wigner model and show that the annealed FP potential has the expected behavior as one changes the signal-to-noise ratio.

In this paper we are interested in the behavior of this annealed potential near overlap zero, see LO($\delta$). More specifically, we consider the annealed potential in a window of size $\delta$, where $\delta$ is related to the parameter $D$ via the entropy of the overlap of two copies from the prior: $D = -\log \mathbb{P}(|\langle u, v\rangle| \geq \delta)$; see Definition 1.5. As previously explained in the context of the low degree method, $D$ is a proxy for runtime (there are about $n^D$ terms in a multivariate polynomial in $n$ variables of degree $D$). A similar heuristic can be made on the FP side: one needs to draw on average $1/\mathbb{P}(|\langle u, v\rangle| \geq \delta) = e^D$-many samples $(u, v)$ from the prior distribution to realize the event $|\langle u, v\rangle| \geq \delta$. This means that for a typical choice of the true signal $u_0$, one needs to draw about $e^D$ samples $u$ from the prior before finding one whose overlap with the truth is $|\langle u, u_0\rangle| \geq \delta$. Once such an initialization is found then one can 'climb' the free energy curve to the closest local maximum

---

[3]For instance, if both $u$ and $Y$ have i.i.d. components, then computing $f$ boils down to a classical large deviation analysis.

[4]Of course, the two parametrizations are equivalent if $\mu$ is supported on a subset of a sphere.

to achieve higher overlap values. (See Figure 1 for an illustration.) So replacing the $D$ by $D \log n$ in the previous expression provides a rough correspondence of runtime between the LD and FP approaches, up to a logarithmic factor.

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

Bounds on $LD(D)$ have already been given for various special cases of the Gaussian additive model such as tensor PCA [HKP$^+$17, KWB19] and spiked Wigner models (including sparse PCA) [KWB19, DKWB19, BBK$^+$21].

As in Section 1.1 we use $\mathbb{P}_u$ to denote the distribution $\mathcal{N}(\lambda u, I_N)$, and write $L_u = \frac{\mathrm{d}\mathbb{P}_u}{\mathrm{d}\mathbb{Q}}$. To write explicit expressions for LD and FP, we will use the following facts, which are implicit in the proof of Theorem 2.6 in [KWB19].

**Proposition 2.3.** *In the Gaussian additive model, we have the formulas*

$$\langle L_u, L_v \rangle_{\mathbb{Q}} = \exp(\lambda^2 \langle u, v \rangle)$$

*and*

$$\langle L_{\tilde{u}}^{\leq D}, L_{\tilde{v}}^{\leq D} \rangle_{\mathbb{Q}} = \exp^{\leq D}(\lambda^2 \langle u, v \rangle)$$

*where* $\exp^{\leq D}(\cdot)$ *denotes the degree-D Taylor expansion of* $\exp(\cdot)$, *namely*

$$(11) \qquad \exp^{\leq D}(x) := \sum_{d=0}^{D} \frac{x^d}{d!}.$$

It will also be helpful to define the *overlap random variable*

$$(12) \qquad s = \langle u, v \rangle \qquad \text{where } u, v \sim \mu \text{ independently.}$$

With Proposition 2.3 in hand, we can rewrite (1) and (3) as (making the dependence on $\lambda$ explicit)

$$(13) \qquad \mathrm{LD}(D, \lambda) = \mathbb{E}_{u,v} \left[ \langle L_{\tilde{u}}^{\leq D}, L_{\tilde{v}}^{\leq D} \rangle_{\mathbb{Q}} \right] = \mathbb{E}_{s} \left[ \exp^{\leq D}(\lambda^2 s) \right]$$

and

$$(14) \qquad \mathrm{FP}(D, \lambda) = \mathbb{E}_{u,v} \left[ \mathbb{1}_{|\langle u,v \rangle| \leq \delta} \cdot \langle L_u, L_v \rangle_{\mathbb{Q}} \right] = \mathbb{E}_{s} \left[ \mathbb{1}_{|s| \leq \delta} \cdot \exp(\lambda^2 s) \right]$$

with $\delta = \delta(D)$ as defined in (3).

We note that both LD and FP are guaranteed to be finite for any fixed choice of $D, \lambda, \mu$: our assumption that $\mu$ has finite moments of all orders implies that $s$ also has finite moments of all orders, and so (13) is finite. Also, (14) is at most $\exp(\lambda^2 \delta) < \infty$.

### 2.1. **FP-LD Equivalence.**

The following two theorems show that in the Gaussian additive model, FP and LD are equivalent up to logarithmic factors in $D$ and $1 + \varepsilon$ factors in $\lambda$. Recall the notation $\mathrm{LD}(D, \lambda)$ and $\mathrm{FP}(D, \lambda)$ from (13) and (14).

**Theorem 2.4** (FP-hard implies LD-hard)**.** *Assume the Gaussian additive model (Definition 2.1) and suppose* $\|u\|^2 \leq M$ *for all* $u \in \mathrm{supp}(\mu)$, *for some* $M > 0$. *Then for any* $\lambda \geq 0$ *and any odd integer* $D \geq 1$,

$$\mathrm{LD}(D, \lambda) \leq \mathrm{FP}(\tilde{D}, \lambda) + e^{-D}$$

*where*

$$\tilde{D} := D \cdot (2 + \log(1 + \lambda^2 M)).$$

The proof can be found later in this section. While the result is non-asymptotic, we are primarily interested in the regime $D = \omega(1)$, in which case we have shown that LD can only exceed FP by an additive $o(1)$ term. We have lost logarithmic factors in passing from $D$ to $\tilde{D}$. For many applications, these log factors are not an issue because (in the "hard" regime) it is possible to prove FP is bounded for some $D = N^{\Omega(1)}$ while $\lambda, M$ are polynomial in $N$.

**Theorem 2.5** (LD-hard implies FP-hard)**.** *Assume the Gaussian additive model (Definition 2.1). For every* $\varepsilon \in (0, 1)$ *there exists* $D_0 = D_0(\varepsilon) > 0$ *such that for any* $\lambda \geq 0$ *and any even integer* $D \geq D_0$, *if*

$$(15) \qquad \mathrm{LD}(D, (1+\varepsilon)\lambda) \leq \frac{1}{eD}(1+\varepsilon)^D$$

*then*

$$(16) \qquad \mathrm{FP}(D, \lambda) \leq \mathrm{LD}(D, (1+\varepsilon)\lambda) + \varepsilon.$$

The proof can be found in Section 5.2. In the asymptotic regime of primary interest, we have the following consequence (also proved in Section 5.2).

**Corollary 2.6.** *Fix any constant $\varepsilon' > 0$ and suppose $D = D_n$, $\lambda = \lambda_n$, $N = N_n$, and $\mu = \mu_n$ are such that $D$ is an even integer, $D = \omega(1)$, and $\mathrm{LD}(D, (1 + \varepsilon')\lambda) = O(1)$. Then*

$$\mathrm{FP}(D, \lambda) \leq \mathrm{LD}(D, (1 + \varepsilon')\lambda) + o(1).$$

**Remark 2.7.** In the theorems above, we have taken the liberty to assume $D$ has a particular parity for convenience. Since FP and LD are both monotone in $D$ (see Lemma 5.1), we can deduce similar results for all integers $D$. For example, if $D$ is even, Theorem 2.4 implies

$$\mathrm{LD}(D, \lambda) \leq \mathrm{LD}(D + 1, \lambda) \leq \mathrm{FP}((D + 1)(2 + \log(1 + \lambda^2 M)), \lambda) + e^{-(D+1)}.$$

We now present the proof of Theorem 2.4 ("FP-hard implies LD-hard"), as it is conceptually simple and also instructive for highlighting the key reason why LD and FP are related. (Some of these ideas extend beyond the Gaussian additive model, as discussed in Remark 2.10 below.) We first need to establish two key ingredients. The first is an inequality for low-degree projections.

**Lemma 2.8.** *In the Gaussian additive model with $D$ odd, for any $u, v \in \mathrm{supp}(\mu)$,*

$$\langle L_{\bar{u}}^{\leq D}, L_{\bar{v}}^{\leq D} \rangle_{\mathbb{Q}} \leq \langle L_u, L_v \rangle_{\mathbb{Q}}.$$

*Proof.* Recalling from Proposition 2.3 the formulas $\langle L_{\bar{u}}^{\leq D}, L_{\bar{v}}^{\leq D} \rangle_{\mathbb{Q}} = \exp^{\leq D}(\lambda^2 \langle u, v \rangle)$ and $\langle L_u, L_v \rangle_{\mathbb{Q}} = \exp(\lambda^2 \langle u, v \rangle)$, the result follows because $\exp^{\leq D}(x) \leq \exp(x)$ for all $x \in \mathbb{R}$ when $D$ is odd (see Lemma 5.4). $\square$

Second, we will need a crude upper bound on $\|L_{\bar{u}}^{\leq D}\|_{\mathbb{Q}}$.

**Lemma 2.9.** *In the Gaussian additive model, for any $u \in \mathrm{supp}(\mu)$,*

$$\|L_{\bar{u}}^{\leq D}\|_{\mathbb{Q}}^2 \leq (D + 1)(1 + \lambda^2 M)^D.$$

*Proof.* Recall that $M$ is an upper bound on $\|u\|^2$, and recall from Proposition 2.3 that $\|L_{\bar{u}}^{\leq D}\|_{\mathbb{Q}}^2 = \langle L_{\bar{u}}^{\leq D}, L_{\bar{u}}^{\leq D} \rangle_{\mathbb{Q}} = \exp^{\leq D}(\lambda^2 \|u\|^2)$. The result follows because $\exp^{\leq D}(\lambda^2 \|u\|^2)$ is the sum of $D + 1$ terms (see (11)), each of which can be upper-bounded by $(1 + \lambda^2 M)^D$. $\square$

With the two lemmas above in hand, we can now prove Theorem 2.4 without using any additional properties specific to the Gaussian additive model.

*Proof of Theorem 2.4.* Recall $\tilde{D} := D \cdot (2 + \log(1 + \lambda^2 M))$. Let $u, v \sim \mu$ independently. Define $\delta = \delta(\tilde{D})$ as in (3), which implies $\Pr(|\langle u, v \rangle| > \delta) \leq e^{-\tilde{D}}$ (see Remark 1.6). Decompose LD into low- and high-overlap terms:

$$\mathrm{LD}(D, \lambda) = \mathop{\mathbb{E}}_{u,v}\left[\langle L_{\bar{u}}^{\leq D}, L_{\bar{v}}^{\leq D} \rangle_{\mathbb{Q}}\right] = \mathop{\mathbb{E}}_{u,v}\left[\mathbb{1}_{|\langle u,v \rangle| \leq \delta} \cdot \langle L_{\bar{u}}^{\leq D}, L_{\bar{v}}^{\leq D} \rangle_{\mathbb{Q}}\right] + \mathop{\mathbb{E}}_{u,v}\left[\mathbb{1}_{|\langle u,v \rangle| > \delta} \cdot \langle L_{\bar{u}}^{\leq D}, L_{\bar{v}}^{\leq D} \rangle_{\mathbb{Q}}\right].$$

The low-overlap term can be related to FP using Lemma 2.8:

$$\mathop{\mathbb{E}}_{u,v}\left[\mathbb{1}_{|\langle u,v \rangle| \leq \delta} \cdot \langle L_{\bar{u}}^{\leq D}, L_{\bar{v}}^{\leq D} \rangle_{\mathbb{Q}}\right] \leq \mathop{\mathbb{E}}_{u,v}\left[\mathbb{1}_{|\langle u,v \rangle| \leq \delta} \cdot \langle L_u, L_v \rangle_{\mathbb{Q}}\right] = \mathrm{FP}(\tilde{D}, \lambda).$$

For the high-overlap term, Lemma 2.9 implies (via Cauchy–Schwarz) the crude upper bound $\langle L_{\bar{u}}^{\leq D}, L_{\bar{v}}^{\leq D} \rangle_{\mathbb{Q}} \leq (D + 1)(1 + \lambda^2 M)^D$, and together with the tail bound $\Pr(|\langle u, v \rangle| > \delta) \leq e^{-\tilde{D}}$ this yields

$$\mathop{\mathbb{E}}_{u,v}\left[\mathbb{1}_{|\langle u,v \rangle| > \delta} \cdot \langle L_{\bar{u}}^{\leq D}, L_{\bar{v}}^{\leq D} \rangle_{\mathbb{Q}}\right] \leq e^{-\tilde{D}} \cdot (D + 1)(1 + \lambda^2 M)^D$$

$$= \exp\left(-\tilde{D} + \log(D + 1) + D \log(1 + \lambda^2 M)\right)$$

$$\leq \exp(-D)$$

where the final step used the definition of $\tilde{D}$ and the fact $\log(D + 1) \leq D$. $\square$

**Remark 2.10.** Many of the ideas in the above proof can potentially be extended beyond the Gaussian additive model. The only times we used the Gaussian additive model were in Lemmas 2.8 and 2.9. The main difficulty in generalizing this proof to other models seems to be establishing the inequality $\langle L_{\bar{u}}^{\leq D}, L_{\bar{v}}^{\leq D}\rangle_{\mathbb{Q}} \leq \langle L_u, L_v\rangle_{\mathbb{Q}}$ from Lemma 2.8. This inequality is *not* true in general: one counterexample is the Gaussian additive model with $D$ *even* and $\langle u, v\rangle < 0$ (combine Proposition 2.3 with Lemma 5.4); see also the counterexamples in Section 4. One of our contributions (see Section 3) is to identify another class of problems where this inequality is guaranteed to hold, allowing us to prove "FP-hard implies LD-hard" for such problems.

**Remark 2.11.** Some prior work has implicitly used FP in the proof of low-degree lower bounds, in a few specific settings where the FP-to-LD connection can be made quite easily [BKW20, KWB19, BBK+21]. Our contribution is to establish this in much higher generality, which requires some new ideas such as the symmetry argument in Section 3.1.

The proof of the converse bound "LD-hard implies FP-hard" (Theorem 2.5) is deferred to Section 5.2. The key step is to show that when $|\langle u, v\rangle| \leq \delta$, the inequality $\langle L_{\bar{u}}^{\leq D}, L_{\bar{v}}^{\leq D}\rangle_{\mathbb{Q}} \leq \langle L_u, L_v\rangle_{\mathbb{Q}}$ is almost an equality (see Lemma 5.7).

## 2.2. **FP-Hard Implies MCMC-Hard.**

In this section we show that bounds on FP imply that a natural class of local Markov chain Monte Carlo (MCMC) methods fail to recover the planted signal. Combining this with the results of the previous section, we also find that low-degree hardness implies MCMC-hardness. We note that in the setting of spin glass models (with no planted signal), the result of [BAJ18] is similar in spirit to ours: they relate a version of annealed FP to the spectral gap for Langevin dynamics.

We again restrict our attention to the additive Gaussian model, now with the additional assumption that the prior $\mu$ is uniform on some finite (usually of exponential size) set $S \subseteq \mathbb{R}^N$ with *transitive symmetry*, defined as follows.

**Definition 2.12.** We say $S \subseteq \mathbb{R}^N$ has *transitive symmetry* if for any $u, v \in S$ there exists an orthogonal matrix $R \in \mathrm{O}(N)$ such that $Ru = v$ and $RS = S$.

The assumption that $S$ is finite is not too restrictive because one can imagine approximating any continuous prior to arbitrary accuracy by a discrete set. Many applications of interest have transitive symmetry. For example, in sparse PCA we might take $u = x^{\otimes 2}$ (thought of as the flattening of a rank-1 matrix) where $x \in \{0, 1\}^n$ has exactly $k$ nonzero entries (chosen uniformly at random). In this case, the orthogonal matrix $R$ in Definition 2.12 is a permutation matrix. More generally, we could take $u = x^{\otimes p}$ where $x$ has any fixed empirical distribution of entries (ordered uniformly at random).

Note that transitive symmetry implies that every $u \in S$ has the same 2-norm. Without loss of generality we will assume this 2-norm is 1, i.e., $S$ is a subset of the unit sphere $\mathbb{S}^{N-1}$.

Given an observation $Y = \lambda u + Z$ drawn from the Gaussian additive model (with $\mu$ uniform on a finite, transitive-symmetric set $S \subseteq \mathbb{S}^{N-1}$), consider the associated *Gibbs measure* $\nu_\beta$ on $S$ defined by

$$(17) \qquad \nu_\beta(v) = \frac{1}{\mathcal{Z}_\beta} \exp(-\beta H(v))$$

where $\beta \geq 0$ is an inverse-temperature parameter (i.e., $1/\beta$ is the *temperature*),

$$H(v) = -\langle v, Y\rangle$$

is the *Hamiltonian*, and

$$\mathcal{Z}_\beta = \sum_{v \in S} \exp(-\beta H(v))$$

is the *partition function*. Note that $\nu_\beta, H, \mathcal{Z}_\beta$ all depend on $Y$, but we have supressed this dependence for ease of notation. When $\beta = \lambda$ (the "Bayesian temperature"), the Gibbs measure $\nu_\beta$ is precisely the posterior distribution for the signal $u$ given the observation $Y = \lambda u + Z$.

We will consider a (not necessarily reversible) Markov chain $X_0, X_1, X_2, \ldots$ on state space $S$ with stationary distribution $\nu_\beta$ (for some $\beta$), that is, if $X_t \sim \nu_\beta$ then $X_{t+1} \sim \nu_\beta$. We will assume a worst-case initial state, which may depend adversarially on $Y$. We will be interested in *hitting time lower bounds*, showing that such a Markov chain will take many steps before arriving at a "good" state that is close to the true signal $u$.

The core idea of our argument is to establish a *free energy barrier*, that is, a subset $B \subseteq S$ of small Gibbs mass that separates the initial state from the "good" states.[5] Such a barrier is well-known to imply a lower bound for the hitting time of the "good" states using conductance; see e.g. [LP17, Theorem 7.4]. In fact, establishing such barriers have been the main tool behind most statistical MCMC lower bounds [Jer92, BGJ20, GZ22, GZ19, GJS21, BWZ20], with the recent exception of [CMZ22]. More formally, we leverage the following result; see (the proof of) Proposition 2.2 in [BWZ20].

**Proposition 2.13** (Free Energy Barrier Implies Hitting Time Lower Bound). *Suppose $X_0, X_1, X_2, \ldots$ is a Markov chain on a finite state space $S$, with some stationary distribution $\nu$. Let $A$ and $B$ be two disjoint subsets of $S$ and define the hitting time $\tau_B := \inf\{t \in \mathbb{N} : X_t \in B\}$. If the initial state $X_0$ is drawn from the conditional distribution $\nu|A$, then for any $t \in \mathbb{N}$, $\Pr(\tau_B \leq t) \leq t \cdot \frac{\nu(B)}{\nu(A)}$. In particular, for any $t \in \mathbb{N}$ there exists a state $v \in A$ such that if $X_0 = v$ deterministically, then $\Pr(\tau_B \leq t) \leq t \cdot \frac{\nu(B)}{\nu(A)}$.*

We will need to impose some "locality" on our Markov chain so that it cannot jump from $A$ to a good state without first visiting $B$.

**Definition 2.14.** We say a Markov chain on a finite state space $S \subseteq \mathbb{S}^{N-1}$ is $\Delta$-*local* if for every possible transition $v \to v'$ we have $\|v - v'\|_2 \leq \Delta$.

We note that the use of local Markov chains is generally motivated and preferred in theory and practice for the, in principle, low computation time for implementing a single step. Indeed, a $\Delta$-local Markov chain sampling from a sufficiently low-temperature Gibbs measure may need to optimize over the whole $\Delta$-neighborhood to update a given point. For such reasons, in most (discrete-state) cases the locality parameter $\Delta > 0$ is tuned so that the $\Delta$-neighborhood of each point is at most of polynomial size; see e.g. [Jer92, BWZ20, GZ19, CMZ22].

Our results will be slightly stronger in the special case that $S$ satisfies the following property, which holds for instance if $u = x^{\otimes p}$ with $p$ even, or if $u \geq 0$ entrywise.

**Definition 2.15.** We say $S \subseteq \mathbb{S}^{N-1}$ has *nonnegative overlaps* if $\langle u, v \rangle \geq 0$ for all $u, v \in S$.

We now state the core result of this section, followed by various corollaries.

**Theorem 2.16** (FP-Hard Implies Free Energy Barrier). *Let $\mu$ be the uniform measure on $S$, where $S \subseteq \mathbb{S}^{N-1}$ is a finite, transitive-symmetric set. The following holds for any $\varepsilon \in (0, 1/2)$, $D \geq 2$, $\lambda \geq 0$, and $\beta \geq 0$. Fix a ground-truth signal $u \in S$ and let $Y = \lambda u + Z$ with $Z \sim \mathcal{N}(0, I_N)$. Define $\delta = \delta(D)$ as in (3). Let*

$$A = \{v \in S : |\langle u, v \rangle| \leq \delta\} \qquad and \qquad B = \{v \in S : \langle u, v \rangle \in (\delta, (1 + \varepsilon)\delta]\}.$$

*With probability at least $1 - e^{-\varepsilon D}$ over $Z$, the Gibbs measure (17) associated to $Y$ satisfies*

$$\frac{\nu_\beta(B)}{\nu_\beta(A)} \leq 2 \left( 2 \cdot \mathrm{FP}(D + \log 2, \tilde{\lambda}) \right)^{1-2\varepsilon} e^{-\varepsilon D}$$

*where*

$$(18) \qquad\qquad \tilde{\lambda} := \sqrt{\beta \lambda \cdot \frac{2 + \varepsilon}{1 - 2\varepsilon}}.$$

*Furthermore, if $S$ has nonnegative overlaps (in the sense of Definition 2.15) then the factor $\frac{2+\varepsilon}{1-2\varepsilon}$ in (18) can be replaced by $\frac{1+\varepsilon}{1-2\varepsilon}$.*

The proof is deferred to Section 5.3 and uses an argument based on [BGJ20] (and also used by [BWZ20] in the "high temperature" regime). This argument makes use of the rotational invariance of Gaussian measure, and we unfortunately do not know how to generalize it beyond the Gaussian additive model. We leave this as an open problem for future work.

---

[5]We note that depending on the temperature, a free energy barrier may arise due to entropy, not necessarily due to an increase of the Hamiltonian. Even when an Hamiltonian is monotonically decreasing along a direction to the desired solution, a free barrier may still exist, consisting of a small-volume set that must be crossed to reach good solutions — its small volume can still lead to a small Gibbs mass, for a sufficiently high temperature.

As mentioned above, one particularly natural choice of $\beta$ is the Bayesian temperature $\lambda$ (which corresponds to sampling from the posterior distribution). In this case $\beta = \lambda$, if $S$ has nonnegative overlaps and $\varepsilon$ is small, there is essentially no "loss" between $\tilde{\lambda}$ and $\lambda$. Without nonnegative overlaps, we lose a factor of $\sqrt{2}$ in $\lambda$.

**Corollary 2.17** (FP-Hard Implies Hitting Time Lower Bound). *In the setting of Theorem 2.16, suppose $X_0, X_1, X_2, \ldots$ is a $\Delta$-local Markov chain with state space $S$ and stationary distribution $\nu_\beta$, for some $\Delta \leq \varepsilon\delta$. Define the hitting time $\tau := \inf\{t \in \mathbb{N} : \langle u, X_t \rangle > \delta\}$. With probability at least $1 - e^{-\varepsilon D}$ over $Z$, there exists a state $v \in A$ such that for the initialization $X_0 = v$, with probability at least $1 - e^{-\varepsilon D/2}$ over the Markov chain,*

$$\tau \geq \frac{e^{\varepsilon D/2}}{2\left(2 \cdot \mathrm{FP}(D + \log 2, \tilde{\lambda})\right)^{1-2\varepsilon}}.$$

*Proof.* Due to $\Delta$-locality,

$$|\langle u, X_{t+1} \rangle - \langle u, X_t \rangle| = |\langle u, X_{t+1} - X_t \rangle| \leq \|u\|_2 \cdot \|X_{t+1} - X_t\|_2 \leq \Delta \leq \varepsilon\delta$$

since $\|u\|_2 = 1$. This means the Markov chain cannot "jump" over the region $B$. Formally, since $X_0 \in A$, we have $\tau \geq \tau_B$ (with $\tau_B$ defined as in Proposition 2.13). The result now follows by combining Proposition 2.13 and Theorem 2.16. $\qquad\square$

We note that Corollary 2.17 applies for all $\Delta \leq \varepsilon\delta(D)$. For various models of interest, we note that the range $\Delta \leq \varepsilon\delta(D)$ for the locality parameter $\Delta$ contains the "reasonable" range of values where the $\Delta$-neighborhoods are of polynomial size. For example, let us focus on the well-studied tensor PCA settng with a Rademacher signal and even tensor power, that is $S = \{u = x^{\otimes 2p} : x \in \{-n^{-p}, n^{-p}\}^n\}$ and the sparse PCA setting where $S = \{x \in \{0, 1/\sqrt{k}\}^n : \|x\|_0 = k\}$ where we focus for simplicity in the regime $k/\sqrt{n} = \omega(1)$. Then for any $D > 0$ and any $\varepsilon > 0$, for both the models, the $\varepsilon\delta(D)$-neighborhood of any point $x \in S$, contains $n^{\omega(1)}$ points. Indeed, it is a simple exercise that for tensor PCA (respectively, sparse PCA) $\delta(D) = \Omega(n^{-p})$ (respectively, $\delta(D) = \Omega(k/n)$), and as an implication, each neighborhood contains every vector $x$ at any Hamming distance $o(\sqrt{n})$ (respectively, $o(k^2/n)$) from the given reference point.

Combining Corollary 2.6 with Corollary 2.17 directly implies the following result, showing that for certain Gaussian additive models, a bounded low-degree likelihood norm implies a hitting time lower bound for local Markov chains. To the best of our knowledge this is the first result of its kind.

**Corollary 2.18** (LD-Hard Implies Hitting Time Lower Bound). *Suppose $D = D_n$ is a sequence with $D = \omega(1)$ such that $D + \log 2$ is an even integer. In the setting of Theorem 2.16, assume for some constant $B > 0$ that*

$$\mathrm{LD}(D + \log 2, (1 + \varepsilon)\tilde{\lambda}) \leq B.$$

*Suppose $X_0, X_1, X_2, \ldots$ is a $\Delta$-local Markov chain with state space $S$ and stationary distribution $\nu_\beta$, for some $\Delta \leq \varepsilon\delta$. Define the hitting time $\tau := \inf\{t \in \mathbb{N} : \langle u, X_t \rangle > \delta\}$. There is a constant $C = C(B, \varepsilon) > 0$ only depending on $B, \varepsilon$ such that the following holds for all sufficiently large $n$. With probability at least $1 - e^{-\varepsilon D}$ over $Z$, there exists a state $v \in A$ such that for the initialization $X_0 = v$, with probability at least $1 - e^{-\varepsilon D/2}$ over the Markov chain,*

$$\tau \geq C(B, \varepsilon)e^{\varepsilon D/2}.$$

**Remark 2.19.** Observe that under a bounded degree-$D$ likelihood norm, Corollary 2.18 not only implies a super-polynomial lower bound on the hitting time of large overlap $\tau$, but also an $e^{\Omega(D)}$-time lower bound, matching the exact "low-degree" time complexity predictions. This significantly generalizes to a wide class of Gaussian additive models a similar observation from [BWZ20] which was in the context of sparse PCA.

**Remark 2.20.** We note that the original work of [BGJ20], on which the proof of Theorem 2.16 is based, showed failure of MCMC methods in a strictly *larger* (by a power of $n$) range of $\lambda$ than the low-degree-hard regime for the tensor PCA problem. In contrast, our MCMC lower bound uses the same argument but matches the low-degree threshold (at the Bayesian temperature). This is because [BGJ20] only considers temperatures that are well above the Bayesian one: in their notation, their result is for constant $\beta$ whereas the Bayesian $\beta$ grows with $n$.

One might naturally wonder whether for every Gaussian additive model considered in Corollary 2.18, an appropriately designed MCMC method achieves a matching upper bound, that is it works all the way down to the low-degree threshold. While we don't know the answer to this question, we highlight the severe lack of tools in the literature towards proving the success of MCMC methods for inference, with only a few exceptions designed in the zero-temperature regime; see e.g. [GZ22]. A perhaps interesting indication of the lack of such tools and generic understanding, has been the recent proof that the classical MCMC method of Jerrum [Jer92] actually fails to recover even almost-linear sized planted cliques [CMZ22], that is it fails much above the $\sqrt{n}$-size low-degree threshold. We note though that [GZ19] suggested (but not proved) a way to "lift" certain free energy barriers causing the failure of MCMC methods, by an appropriate overparametrization of the state space. Finally, we note that non-rigorous statistical physics results have suggested the underperformance of MCMC for inference in various settings, see e.g. [AFUZ19].

Our Theorem 2.16 provides a free energy barrier which becomes larger as the temperature $1/\beta$ becomes larger (equivalently, as $\tilde{\lambda}$ becomes smaller). We note that there are bottleneck arguments which can establish the failure of low-temperature MCMC methods using the so-called Overlap Gap Property for inference, see e.g. [GZ22, BWZ20]. Yet these techniques are usually based on a careful second moment argument and appear more difficult to be applied in high generality and to connect with the technology built in the present work.

## 3. Planted Sparse Models

As discussed previously (see Remark 2.10), the main difficulty in generalizing our proof of "FP-hard implies LD-hard" from the Gaussian additive model to other models is establishing the inequality $\langle L_{\bar{u}}^{\leq D}, L_{\bar{v}}^{\leq D} \rangle_{\mathbb{Q}} \leq \langle L_u, L_v \rangle_{\mathbb{Q}}$. We showed previously that this inequality holds in the Gaussian additive model with $D$ odd (Lemma 2.8). Here we identify another class of problems (with a "sparse planted signal") where we can establish this inequality via a symmetry argument, allowing us to prove "FP-hard implies LD-hard" and thus use FP as a tool to prove low-degree lower bounds. As an application, we give new low-degree lower bounds for the detection problem in sparse linear regression (see Section 3.2).

**Assumption 3.1** (Distributional Assumptions)**.** In this section we focus on hypothesis testing between two distributions $\mathbb{P}, \mathbb{Q}$ on $\mathbb{R}^N$ of a specific form, where the signal corresponds to a planted subset of entries.

- Under $Y \sim \mathbb{Q}$, each entry $Y_i$ is drawn independently from some distribution $Q_i$ on $\mathbb{R}$.
- Under $Y \sim \mathbb{P}$, first a signal vector $u \in \mathbb{R}^n$ is drawn from some distribution $\mu$ on $\mathbb{R}^n$. The only role of the vector $u$ is to be a surrogate for a subset of "planted entries." To be more precise, to each $u \in \text{supp}(\mu)$ we associate a set of planted entries $\Phi_u \subseteq [N]$.[6] Conditioned on $u$, we draw $Y$ from the following distribution $\mathbb{P}_u$.
  - For entries $i \notin \Phi_u$, draw $Y_i$ independently from $Q_i$ (the same as in $\mathbb{Q}$).
  - The entries in $\Phi_u$ can have an arbitrary joint distribution (independent from the entries outside $\Phi_u$) subject to the following symmetry condition: for any subset $S \subseteq [N]$ and any $u \in \text{supp}(\mu)$ such that $S \subseteq \Phi_u$, the marginal distribution $\mathbb{P}_u|_S$ is equal to some distribution $P_S$ that may depend on $S$ but not on $u$.

We will further assume that $Q_i$ and $P_S$ have finite moments of all orders, and that $\mathbb{P}_u$ is absolutely continuous with respect to $\mathbb{Q}$. Finally, we assume $\mathbb{Q}$ has a complete basis of orthogonal polynomials in $L^2(\mathbb{Q})$. (This is for instance known to be the case if the marginals $Q_i$ are Gaussian or have bounded support. More generally, this can be guaranteed under mild conditions on the marginals $Q_i$. See Appendix B for further discussion.)

As in Section 1.1 we define $L_u = \frac{d\mathbb{P}_u}{d\mathbb{Q}}$, and define

$$\text{LD}(D) = \underset{u,v \sim \mu}{\mathbb{E}} \left[ \langle L_{\bar{u}}^{\leq D}, L_{\bar{v}}^{\leq D} \rangle_{\mathbb{Q}} \right]$$

---

[6]This seemingly involved way to identify the planted structure has some advantages as we will see, including providing a suitable measure of "overlap." It may be helpful to think about the special case $u \in \{0,1\}^n$, $N = n$, and $\Phi_u = \text{supp}(u)$ for intuition, although in many cases of interest $u$ will correspond to a subset of entries of vectors in a different dimension.

and

$$\mathrm{FP}(D) = \mathop{\mathbb{E}}_{u,v\sim\mu}\left[\mathbb{1}_{|\langle u,v\rangle|\leq\delta}\cdot\langle L_u, L_v\rangle_{\mathbb{Q}}\right]$$

where

$$(19) \qquad\qquad \delta = \delta(D) = \sup\left\{\varepsilon \geq 0 \text{ s.t. } \mathop{\Pr}_{u,v\sim\mu}\left(|\langle u,v\rangle| \geq \varepsilon\right) \geq e^{-D}\right\}.$$

**Remark 3.2.** Any distribution $\mathbb{P}$ can satisfy Assumption 3.1 by taking only one possible value for $u$ and taking the associated $\Phi_u$ to be all-ones. However, in this case the resulting FP degenerates to simply $\|L\|_{\mathbb{Q}}^2$ (which can't be $O(1)$ unless detection is information-theoretically impossible). In order to prove useful low-degree lower bounds (in the possible-but-hard regime) using the tools from this section, it will be important to have many possible $u$'s with different associated distributions $\mathbb{P}_u$, and so the $\Phi_u$'s must be sparse (not all ones).

We now give some motivating examples that satisfy the assumptions above. The first is a generic class of problems that includes the classical planted clique and planted dense subgraph problems (see e.g. [HWX15]).

**Example 3.3** (Planted Subgraph Problems). Let $N = \binom{n}{2}$ and observe $Y \in \mathbb{R}^N$, which we think of as the complete graph on $n$ vertices with a real-valued observation on each edge. Under $\mathbb{Q}$, each entry $Y_{ij}$ is drawn independently from some distribution $Q$ on $\mathbb{R}$. Under $\mathbb{P}$, a vertex subset $C \subseteq [n]$ of size $|C| = k$ is chosen uniformly at random. For edges $(i,j)$ whose endpoints both lie in $C$, we draw $Y_{ij}$ independently from some distribution $P$. For all other edges, $Y_{ij}$ is drawn independently from $Q$.

To see that this satisfies Assumption 3.1, let $u \in \{0,1\}^n$ be the indicator vector for $C$ and let $\Phi_u \subseteq [N]$ be the set of edges whose endpoints both lie in $C$. For the symmetry condition, note that for any edge subset $S \subseteq [N]$ and any $u$ such that $S \subseteq \Phi_u$, all edges in $S$ must have both endpoints in $C$, so the marginal distribution $\mathbb{P}_u|_S$ is simply the product distribution $P^{\otimes|S|}$.

The next example, *sparse generalized linear models* (sparse GLMs) includes various classical problems such as the sparse linear regression problem we will study in Section 3.2.

**Example 3.4** (Sparse GLMs). Fix an arbitrary activation function $\phi : \mathbb{R} \to \mathbb{R}$ and real-valued distributions $\nu, \Xi, Q, R$. The observation will be a pair $(X, Y) \in \mathbb{R}^{m\times n} \times \mathbb{R}^m$ generated as follows.

- Under $\mathbb{Q}$, $X$ has i.i.d. entries drawn from $Q$ and $Y$ has i.i.d. entries drawn from $R$.
- Under $\mathbb{P}$, $X$ again has i.i.d. entries drawn from $Q$. We draw a $k$-sparse signal vector $\beta \in \mathbb{R}^n$ by first choosing exactly $k$ distinct entries uniformly at random to be the support, then drawing these $k$ entries i.i.d. from $\nu$, and then setting all remaining entries to 0. We then let

$$Y = \phi(X\beta) + \xi,$$

where $\phi$ is applied entrywise and the noise $\xi \in \mathbb{R}^m$ is drawn i.i.d. from $\Xi$.

To see that this satisfies Assumption 3.1, let $u \in \{0,1\}^n$ be the support of $\beta$ (so $\mathbb{P}_u$ includes sampling the nonzero entries of $\beta$ from $\nu$). Define $\Phi_u \subseteq ([m]\times[n]) \sqcup [m]$ (where $\sqcup$ denotes disjoint union) to contain: (i) all entries of $Y$, and (ii) all "planted" columns of $X$, that is, the columns of $X$ indexed by the support of $\beta$. To see that the entries in $\Phi_u$ are independent (conditioned on $u$) from those outside $\Phi_u$, note that $Y$ depends only on the planted columns of $X$. For the symmetry condition, let $S \subseteq ([m]\times[n]) \sqcup [m]$ and let $J_S \subseteq [n]$ index the columns of $X$ that contain at least one entry of $S$. Suppose $u, v$ are such that $S \subseteq \Phi_u$ and $S \subseteq \Phi_v$; we will show $\mathbb{P}_u|_S = \mathbb{P}_v|_S$. From the definition of $\Phi_u$, we see that $u$ and $v$ must both contain the columns $J_S$ in their support, and each also contains $k - |J_S|$ additional columns in their support. However, the joint distribution of $(X_{[m], J_S}, Y)$ does not depend on which additional $k - |J_S|$ columns are the planted ones, due to symmetry in the model. Therefore $\mathbb{P}_u|_S = \mathbb{P}_v|_S$.

We give one more example, which is the one we will need for our application in Section 3.2. This is a variation on the previous example with an added ingredient: for technical reasons we will need to condition on a particular high-probability event.

**Example 3.5** (Sparse GLMs with Conditioning). Consider the setting of Example 3.4. Let $i_1, \ldots, i_k \in [n]$ denote the indices in the support of $\beta$, and let $X_{i_1}, \ldots, X_{i_k}$ denote the corresponding ("planted") columns of $X$. Let $A = A(X_{i_1}, \ldots, X_{i_k})$ be an event that depends only on the planted columns and is invariant under permutations of its $k$ inputs. (In other words, $A$ has access to a multi-set of $k$ vectors containing the values in the planted columns, but not the planted indices $i_1, \ldots, i_k$.) Under $\mathbb{Q}$, draw $(X, Y)$ as in Example 3.4. Under $\mathbb{P}$, draw $u$ as in Example 3.4 (equivalently, draw $i_1, \ldots, i_k$) and sample $(X, Y)$ from the conditional distribution $\mathbb{P}_u | A$.

This satisfies Assumption 3.1 by the same argument as in Example 3.4. Here it is crucial that the conditioning on $A$ does not affect the non-planted columns. (It would also be okay for $A$ to depend on $Y$, but we won't need this for our sparse regression example.)

### 3.1. FP-Hard Implies LD-Hard.
In the setting of Assumption 3.1, we now establish the key inequality that will allow us to prove "FP-hard implies LD-hard."

**Proposition 3.6.** *Under Assumption 3.1, for any integer $D \geq 0$ and any $u, v \in \mathrm{supp}(\mu)$,*
$$\langle L_u^{\leq D}, L_v^{\leq D} \rangle_{\mathbb{Q}} \leq \langle L_u, L_v \rangle_{\mathbb{Q}}.$$

*Proof.* We first construct an orthonormal basis of polynomials for $L^2(\mathbb{Q})$. For each $i \in [N]$, first construct an orthonormal basis of polynomials for $L^2(Q_i)$, that is, a collection $\{h_k^{(i)}\}_{k \in I_i}$ for some index set $I_i \subseteq \mathbb{N}$ (we use the convention $0 \in \mathbb{N}$), where $h_k^{(i)} : \mathbb{R} \to \mathbb{R}$ is a degree-$k$ polynomial, the set $\{h_k^{(i)}\}_{k \in I_i, \, k \leq D}$ spans all polynomials of degree at most $D$, and $\langle h_k^{(i)}, h_\ell^{(i)} \rangle_{Q_i} = \mathbb{1}_{k=\ell}$. Such a basis can be constructed by applying the Gram–Schmidt process to the monomial basis $\{1, x, x^2, \ldots\}$, discarding any monomials that are linearly dependent on the previous ones. In particular, $0 \in I_i$ and $h_0^{(i)}(x) = 1$. From orthonormality, we have the property

$$(20) \qquad \mathbb{E}_{x \sim Q_i}[h_k^{(i)}] = \langle h_k^{(i)}, 1 \rangle_{Q_i} = \langle h_k^{(i)}, h_0^{(i)} \rangle_{Q_i} = 0 \qquad \text{for all } k \geq 1,$$

which will be needed later.

Now an orthonormal basis of polynomials for $\mathbb{Q}$ is given by $\{H_\alpha\}_{\alpha \in I}$ where the index set $I \subseteq \mathbb{N}^N$ is the direct product $I := \prod_{i \in [N]} I_i$ and $H_\alpha(x) := \prod_{i \in [N]} h_{\alpha_i}^{(i)}(x_i)$. Letting $|\alpha| := \sum_{i \in [N]} \alpha_i$, these have the property that $\{H_\alpha\}_{\alpha \in I, \, |\alpha| \leq D}$ spans all polynomials $\mathbb{R}^N \to \mathbb{R}$ of degree at most $D$, and $\langle H_\alpha, H_\beta \rangle_{\mathbb{Q}} = \mathbb{1}_{\alpha=\beta}$.

Expanding in this basis, we have

$$(21) \qquad \langle L_u^{\leq D}, L_v^{\leq D} \rangle_{\mathbb{Q}} = \sum_{\alpha \in I, \, |\alpha| \leq D} \langle L_u, H_\alpha \rangle_{\mathbb{Q}} \langle L_v, H_\alpha \rangle_{\mathbb{Q}} = \sum_{\alpha \in I, \, |\alpha| \leq D} \mathbb{E}_{Y \sim \mathbb{P}_u}[H_\alpha(Y)] \mathbb{E}_{Y \sim \mathbb{P}_v}[H_\alpha(Y)],$$

where we have used the change-of-measure property $\mathbb{E}_{\mathbb{Q}}[L \cdot f] = \mathbb{E}_{\mathbb{P}}[f]$. We claim that for any $\alpha$,

$$(22) \qquad \mathbb{E}_{Y \sim \mathbb{P}_u}[H_\alpha(Y)] \mathbb{E}_{Y \sim \mathbb{P}_v}[H_\alpha(Y)] \geq 0.$$

This claim completes the proof because every term in (21) is nonnegative and so

$$\langle L_u^{\leq 0}, L_v^{\leq 0} \rangle_{\mathbb{Q}} \leq \langle L_u^{\leq 1}, L_v^{\leq 1} \rangle_{\mathbb{Q}} \leq \langle L_u^{\leq 2}, L_v^{\leq 2} \rangle_{\mathbb{Q}} \leq \cdots \leq \lim_{D \to \infty} \langle L_u^{\leq D}, L_v^{\leq D} \rangle_{\mathbb{Q}} = \langle L_u, L_v \rangle_{\mathbb{Q}},$$

where the final equality follows due to our assumption that $\mathbb{Q}$ has a *complete* orthonormal basis of polynomials, that is, the space of polynomials is dense in $L^2(\mathbb{Q})$. It remains to prove the claim (22). Let $S_\alpha \subseteq [N]$ be the support of $\alpha$, that is, $S_\alpha = \{i \in [N] : \alpha_i \geq 1\}$. Note that since $h_0^{(i)} = 1$, $H_\alpha(Y)$ depends only on the entries $Y|_{S_\alpha}$. If $S_\alpha \not\subseteq \Phi_u$, we will show that $\mathbb{E}_{Y \sim \mathbb{P}_u}[H_\alpha(Y)] = 0$, implying that (22) holds with equality. To see this, fix some $i \in S_\alpha \setminus \Phi_u$ (which exists because $S_\alpha \not\subseteq \Phi_u$) and note that under $Y \sim \mathbb{P}_u$, $Y_i$ is drawn from $Q_i$ independent from all other entries of $Y$. Letting $\bar{\alpha}$ be obtained from $\alpha$ by setting $\alpha_i$ to 0, $H_{\bar{\alpha}}(Y)$ depends only on $Y|_{S_\alpha \setminus \{i\}}$ and is therefore independent from $Y_i$. This means

$$\mathbb{E}_{Y \sim \mathbb{P}_u}[H_\alpha(Y)] = \mathbb{E}_{Y \sim \mathbb{P}_u}[H_{\bar{\alpha}}(Y) \cdot h_{\alpha_i}^{(i)}(Y_i)] = \mathbb{E}_{Y \sim \mathbb{P}_u}[H_{\bar{\alpha}}(Y)] \mathbb{E}_{x \sim Q_i}[h_{\alpha_i}^{(i)}(x)] = \mathbb{E}_{Y \sim \mathbb{P}_u}[H_{\bar{\alpha}}(Y)] \cdot 0 = 0,$$

where we have used (20) along with the fact $\alpha_i \geq 1$ (since $i \in S_\alpha$). The same argument also shows that if $S \not\subseteq \Phi_v$ then $\mathbb{E}_{Y \sim \mathbb{P}_v}[H_\alpha(Y)] = 0$.

It therefore remains to prove (22) in the case $S_\alpha \subseteq \Phi_u \cap \Phi_v$. In this case, the symmetry condition in Assumption 3.1 implies $\mathbb{P}_u|_{S_\alpha} = \mathbb{P}_v|_{S_\alpha}$. Since $H_\alpha(Y)$ depends only on $Y|_{S_\alpha}$, this means $\mathbb{E}_{Y \sim \mathbb{P}_u}[H_\alpha(Y)] = \mathbb{E}_{Y \sim \mathbb{P}_v}[H_\alpha(Y)]$, implying (22). $\qquad\square$

As a consequence of the above, we can now show "FP-hard implies LD-hard."

**Theorem 3.7** (FP-hard implies LD-hard). *Under Assumption 3.1, suppose*

$$\sup_{u \in \mathrm{supp}(\mu)} \|L_{\bar{u}}^{\leq D}\|_{\mathbb{Q}}^2 \leq M$$

*for some $M \geq 1$ and some integer $D \geq 0$. Then*

$$\mathrm{LD}(D) \leq \mathrm{FP}(D + \log M) + e^{-D}.$$

For intuition, it is typical to have $M = n^{O(D)}$ and so $\log M = O(D \log n)$. This will be the case in our sparse regression example.

*Proof.* The proof is nearly identical to that of Theorem 2.4. We recap the main steps here.

Let $u, v \sim \mu$ independently. Define $\tilde{D} := D + \log M$ and $\delta = \delta(\tilde{D})$ as in (19), which implies $\Pr\left(|\langle u, v \rangle| > \delta\right) \leq e^{-\tilde{D}}$ (see Remark 1.6). Decompose

$$\mathrm{LD}(D, \lambda) = \mathop{\mathbb{E}}_{u,v}\left[\langle L_{\bar{u}}^{\leq D}, L_{\bar{v}}^{\leq D}\rangle_{\mathbb{Q}}\right] = \mathop{\mathbb{E}}_{u,v}\left[\mathbb{1}_{|\langle u,v\rangle| \leq \delta} \cdot \langle L_{\bar{u}}^{\leq D}, L_{\bar{v}}^{\leq D}\rangle_{\mathbb{Q}}\right] + \mathop{\mathbb{E}}_{u,v}\left[\mathbb{1}_{|\langle u,v\rangle| > \delta} \cdot \langle L_{\bar{u}}^{\leq D}, L_{\bar{v}}^{\leq D}\rangle_{\mathbb{Q}}\right].$$

The low-overlap term can be related to FP using Proposition 3.6:

$$\mathop{\mathbb{E}}_{u,v}\left[\mathbb{1}_{|\langle u,v\rangle| \leq \delta} \cdot \langle L_{\bar{u}}^{\leq D}, L_{\bar{v}}^{\leq D}\rangle_{\mathbb{Q}}\right] \leq \mathop{\mathbb{E}}_{u,v}\left[\mathbb{1}_{|\langle u,v\rangle| \leq \delta} \cdot \langle L_u, L_v\rangle_{\mathbb{Q}}\right] = \mathrm{FP}(\tilde{D}, \lambda).$$

For the high-overlap term,

$$\mathop{\mathbb{E}}_{u,v}\left[\mathbb{1}_{|\langle u,v\rangle| > \delta} \cdot \langle L_{\bar{u}}^{\leq D}, L_{\bar{v}}^{\leq D}\rangle_{\mathbb{Q}}\right] \leq e^{-\tilde{D}} \cdot M \leq e^{-D}$$

using the choice of $\tilde{D}$. $\qquad\square$

3.2. **Application: Sparse Linear Regression.** In this section we give sharp low-degree lower bounds for the hypothesis testing version of a classical inference model: sparse linear regression with Gaussian covariates and Gaussian noise. This classical model, which shares many similarities with the well-studied setting of compressed sensing, admits an interesting information-computation gap for an appropriate choice of parameters (see [GZ22] and references therein). There is a sample regime where multiple polynomial-time algorithms can correctly infer the hidden coefficient vector, including convex programs such as LASSO [Wai09] or greedy compressed sensing algorithms such as Basis Pursuit [DT10]. Interestingly though, exponential-time algorithms are known to work for sample sizes which are an order of magnitude smaller as compared to the known polynomial-time ones (see [RXZ21] and references therein). All known polynomial-time algorithms for sparse regression are either believed or proven to fail in the intermediate "hard" regime. Motivated by such results, [GZ22] study this gap and prove that a low-temperature free-energy barrier, also called the Overlap Gap Property for inference, appears for a part of the hard regime. Their result implies that certain low-temperature MCMC methods fail, leaving open the question of whether more evidence, such as a low-degree lower bound, can also be established in the hard regime to support the presence of a gap.

In this section, we establish such a low-degree lower bound for the associated detection problem in sparse regression. As we discuss in Remark 3.14, the lower bound seems difficult to prove using existing approaches. We will instead prove this result in an indirect manner by leveraging the connection with FP developed in the previous section, illustrating that FP can be a powerful tool for proving low-degree lower bounds that otherwise seem out of reach.

Formally, we consider the following detection task, also studied by [ITV10, Arp21].

**Definition 3.8** (Sparse Linear Regression: Hypothesis Testing). Given a sample size $m \in \mathbb{N}$, feature size $n \in \mathbb{N}$, sparsity level $k \in \mathbb{N}$ with $k \leq n$ and noise level $\sigma > 0$, we consider hypothesis testing between the following two distributions over $(X, Y) \in \mathbb{R}^{m \times n} \times \mathbb{R}^m$.

- $\mathbb{Q}$ generates a pair $(X, Y)$ where both $X$ and $Y$ have i.i.d. $\mathcal{N}(0,1)$ entries.
- $\mathbb{P}$ generates a pair $(X, Y)$ as follows. First $X$ is drawn with i.i.d. $\mathcal{N}(0,1)$ entries. Then, for a planted signal $u \in \{0,1\}^n$ drawn uniformly from all binary vectors of sparsity exactly $k$, and independent noise $W \sim \mathcal{N}(0, \sigma^2 I_m)$, we set

$$Y = (k + \sigma^2)^{-1/2}(Xu + W).$$

This is a specific instance of a sparse GLM (see Example 3.4). As a remark, note that the marginal distribution of $Y$ is $\mathcal{N}(0, I_m)$ under both $\mathbb{Q}$ and $\mathbb{P}$ (but under $\mathbb{P}$, $Y$ is correlated with $X$).

We follow the parameter assumptions from [GZ22].

**Assumption 3.9** (Scaling of the parameters). For constants $\theta \in (0,1)$ and $R > 0$, consider a scaling regime where $n \to \infty$ and

- (Sublinear sparsity)
$$k = n^{\theta + o(1)},$$
- (High signal-to-noise ratio per sample)
$$\sigma^2 = o(k),$$
- (Scale of the sample size)
$$m = (1 + o(1))Rk\log(n/k) = (1 + o(1))R(1 - \theta)k\log n.$$

The scaling of the sample size is chosen because the low-degree threshold will occur at this scaling. The high-SNR assumption is for the sake of simplicity, and guarantees that the low-degree threshold will not depend on the value of $\sigma$. Our techniques can likely be generalized to the case where $\sigma$ is larger, but as [GZ22] consider this scaling, we focus on this regime for our result.

Under the above assumptions, multiple works in the literature have studied the closely related task of *approximate recovery* of $u$ given access to $m$ samples from the planted model $\mathbb{P}$, where the goal is to estimate the support of $u$ with $o(k)$ errors (false positives plus false negatives) with probability $1 - o(1)$. It is known that when given $m > 2k \log n$ samples, or equivalently when $R > 2/(1 - \theta)$, the LASSO convex program [Wai09] succeeds in achieving the even harder goal of exact recovery. We prove (see Section 6.2.5) that under the weaker condition $R > 2$ (for any $\theta \in (0,1)$), a simple thresholding algorithm achieves approximate recovery. (It was previously suggested by non-rigorous calculations, but not proven to the best of our knowledge, that AMP also achieves approximate recovery when $R > 2$ [RXZ19], and a similar result is expected to hold for LASSO as well [GZ22].) On the other hand, it is known [RXZ21] that the information-theoretic sample size is

$$(23) \qquad m_{\inf} = 2k \, \frac{\log(n/k)}{\log(k/\sigma^2 + 1)} = o(k \log n/k),$$

above which the exponential-time maximum-likelihood estimator approximately recovers the hidden signal, while no estimator succeeds below it. This line of work suggests the presence of a possible-but-hard regime for approximate recovery when $0 < R < 2$ (for any fixed $\theta \in (0,1)$). In [GZ22], rigorous evidence was provided for this gap, namely when $0 < R < c_0$ for a small constant $c_0 > 0$, the Overlap Gap Property appears and certain MCMC methods fail.

Turning back to the detection task, in sparse regression it holds that approximate recovery is formally at least as hard as strong detection in the sense that there is a polynomial-time reduction from strong detection to approximate recovery [Arp21]. In particular, the results mentioned above imply that there is a polynomial-time algorithm for strong detection whenever $R > 2$. Furthermore, any evidence of hardness for detection when $0 < R < 2$ would suggest that recovery should also be hard in the same regime. We note that the information-theoretic threshold for strong detection is the same as that for approximate recovery, namely $m_{\inf}$ as defined in (23) [RXZ21]. This makes strong detection information-theoretically possible in the entire regime $0 < R < 2$, but the algorithm achieving this has exponential-in-$k$ runtime due to a brute-force search over all possible $\binom{n}{k}$ signals.

In the following result, we provide rigorous "low-degree evidence" (and a matching upper bound) for the precise optimal trade-off between sparsity, sample complexity, and time complexity for the sparse linear

regression detection task. As compared to prior work, this is a sharper threshold than the low-degree lower bounds in [Arp21] (as we discuss below), and a different scaling regime than the one considered in [ITV10] (which is an information-theoretic result). The proof is deferred to Section 6.

**Theorem 3.10.** *Define*

$$(24) \qquad R_{\mathrm{LD}}(\theta) = \begin{cases} \frac{2(1-\sqrt{\theta})}{1+\sqrt{\theta}} & \text{if } 0 < \theta < \frac{1}{4}, \\ \frac{1-2\theta}{1-\theta} & \text{if } \frac{1}{4} \leq \theta < \frac{1}{2}, \\ 0 & \text{if } \frac{1}{2} \leq \theta < 1. \end{cases}$$

*Consider sparse linear regression (Definition 3.8) in the scaling regime of Assumption 3.9.*

(a) *(Hard regime) If $R < R_{\mathrm{LD}}(\theta)$ then no degree-$o(k)$ polynomial weakly separates $\mathbb{P}$ and $\mathbb{Q}$ (see Definition 1.8).*

(b) *(Easy regime) If $R > R_{\mathrm{LD}}(\theta)$ then there is a polynomial-time algorithm for strong detection between $\mathbb{P}$ and $\mathbb{Q}$ (see Definition 1.2).*

The threshold $R_{\mathrm{LD}}(\theta)$ is a continuous and monotone decreasing function of $\theta \in (0, 1)$, that is, the problem is expected to become easier the larger $\theta$ is. When $\theta \geq 1/2$ we have $R_{\mathrm{LD}}(\theta) = 0$ which means the testing problem is "easy" for any fixed $R > 0$.

The algorithm that gives the matching upper bound in part (b) is fairly simple but somewhat subtle: letting $X_j$ denote the $j$th column of $X$, the idea is to count the number of indices $j \in [n]$ for which $\langle X_j, Y \rangle / \|Y\|_2$ exceeds a particular (carefully chosen) threshold, and then threshold this count.

**Remark 3.11.** We expect that by approximating the algorithm from part (b) by a polynomial, it is possible to prove that for $R > R_{\mathrm{LD}}(\theta)$, there is a degree-$O(\log n)$ polynomial that strongly separates $\mathbb{P}$ and $\mathbb{Q}$.

**Remark 3.12** (Optimality of "brute-force search" in the hard regime). Part (a) suggests that for $R < R_{\mathrm{LD}}(\theta)$, weak detection requires runtime $\exp(\tilde{\Omega}(k))$, which matches the runtime of the brute-force search algorithm of [RXZ21]. This is in contrast to the related sparse PCA problem, where the low-degree analysis suggests a smoother tradeoff between runtime and SNR, with non-trivial subexponential-time algorithms existing in the hard regime [DKWB19, HSV20].

**Remark 3.13** (Implications for recovery). Recall that approximate recovery is formally at least as hard as strong detection [Arp21]. Thus, Theorem 3.10 suggests computational hardness of recovery when $R < R_{\mathrm{LD}}(\theta)$. In the limit $\theta \downarrow 0$, this is essentially tight: $\lim_{\theta \downarrow 0} R_{\mathrm{LD}}(\theta) = 2$, matching the threshold achieved by both LASSO [Wai09] and our thresholding algorithm (Section 6.2.5). For larger values of $\theta$, there appears to be a detection-recovery gap, so our lower bound (while sharp for detection) does not suggest a sharp recovery lower bound. An interesting open problem is to establish a direct low-degree lower bound for *recovery* (in the style of [SW20]) for all $R < 2$ and all $\theta \in (0, 1)$.

**Proof techniques.** Proving the sharp low-degree lower bound in the regime $\theta < 1/4$ requires a *conditional* low-degree calculation: instead of bounding LD for testing $\mathbb{P}$ versus $\mathbb{Q}$, we bound LD for testing the conditional distribution $\mathbb{P}|A$ versus $\mathbb{Q}$, for some high-probability event $A$ (see Section 6.1.1 for more details). This is necessary because the standard LD blows up at a sub-optimal threshold due to a rare "bad" event under $\mathbb{P}$. Conditioning arguments of this type are common for information-theoretic lower bounds (see e.g. [BMNN16, BMV$^+$17, PWB16, PWBM18]), but this is (to our knowledge) the first instance where conditioning has been needed for a low-degree lower bound (along with the concurrent work [CGHK$^+$22] by some of the same authors). We note that [Arp21] gave low-degree lower bounds for sparse regression by analyzing the standard (non-conditioned) LD, and our result improves the threshold when $\theta < 1/4$ via conditioning.

**Remark 3.14.** The standard approach to bounding LD involves direct moment calculations; see e.g. Section 2.4 of [Hop18] for a simple example, or [Arp21] for the case of sparse regression. It seems difficult to carry out our conditional low-degree calculation by this approach because it does not seem straightforward to directly analyze the moments of $\mathbb{P}|A$ for our event $A$. Instead, we bound FP for the $\mathbb{P}|A$ versus $\mathbb{Q}$ problem and then use the machinery from the previous section to conclude a bound on the conditional LD. Luckily, FP

is a somewhat simpler object that "plays well" with the conditioning, leading to a more tractable calculation. This illustrates that FP can be a powerful tool for proving low-degree lower bounds that may otherwise be out of reach.

**Remark 3.15.** Finally, we point out that the prior results and our low-degree hardness results supporting the information-computation gap for sparse linear regression are conjectured to be meaningful only under the assumption that the noise level $\sigma$ is *not exponentially small in n*. If $\sigma$ is exponentially small, it is known that polynomial-time lattice-based methods can exactly recover the hidden signal $u$ even with access to only $m = 1$ sample from the planted model [ZG18] (and in particular solve the detection task as well). We direct the interested reader to the discussion in [ZSWB21] for the importance of non-trivial noise in making computational predictions in inference.

## 4. Counterexamples

Our results suggest that one may hope for a formal FP-LD equivalence in much higher generality than what we have proven. However, in this section we discuss a few obstacles which a more widely-applicable free energy-based criterion for computational hardness will have to overcome. By "more widely-applicable," we mean that we would like a criterion which accurately predicts information-computation gaps for a wide range of problems outside the Gaussian additive model – after all, the low-degree approach appears to make accurate predictions for e.g. graph partitioning problems, constraint satisfaction, planted clique, tensor PCA, and more, and it remains to develop a rigorous mathematical theory based on free energy barriers that achieves the same.

We first demonstrate a simple hypothesis testing problem for distributions on the $n$-dimensional hypercube for which the FP criterion, as we have defined it, makes an obviously-incorrect prediction about computational hardness – predicting that an easy problem is hard. That is, $\text{FP}(D) = o(1)$ for the problems we demonstrate, but they are polynomial-time solvable (in particular, $\text{LD}(D) \gg 1$ for small $D$). This shows that either (a) the FP criterion does generalize accurately to a broad range of testing problems beyond the Gaussian additive model, but the particular problem we construct has to be "defined out" of that range, or (b) the FP criterion itself must be modified to successfully generalize beyond the Gaussian additive model.

We give some evidence against option (a) by showing that our construction of such "bad" problems is robust in two ways, making it seemingly hard to define a natural class of problems which avoids the issue. First, adding some noise to the alternative hypothesis $H_1$ does not fix the issue. And, second, the issue can appear even in a natural planted subgraph detection problem of the kind that we would expect a good theory of computational hardness to address.

This leaves option (b), that to generalize past the Gaussian additive model, we should look for a different free energy-based criterion for hardness. The intuitions from statistical physics which in the first place guided the definition of the Franz–Parisi criterion actually suggest that computational hardness should coincide with *non-monotonicity* of some one-parameter curve associated to an inference problem. For instance, the *replica heuristic* for predicting computational hardness associates to an inference problem (with a fixed SNR) a certain one-parameter curve called the *replica symmetric potential* which, according to the heuristic, is monotonically increasing if and only if the problem is computationally tractable at that SNR (see Figure 1 of [DMK+16] or Figure 1 of [BPW18]). By contrast, $\text{FP}(D)$ measures the *value* of a related curve near the typical overlap.

While the replica heuristic and others have been remarkably successful at predicting hardness,[7] we show that formalizing such a criterion will require overcoming some technical challenges. Free energy-based approaches to computational hardness we are aware of can all be understood to study some function $f(t)$ which tracks or approximates the free energy of a posterior distribution (or the corresponding Gibbs measure at a different "non-Bayesian" temperature) *restricted to overlap $\approx t$ with the some ground-truth signal*. (Exactly which function $f$ is used depends on which of many possible free energy-based hardness criteria is in question.) The

---

[7]We note, however, that the replica heuristic and the associated AMP algorithm do not predict the correct computational threshold for tensor PCA [RM14, LML+17, BGJ20] (see also [WEM19, BCRT20] for discussion), which was part of our initial motivation to search for a different criterion.

$LO(\delta)$ curve we study in this paper and the one-parameter curve studied in the replica method are both examples of such $f$.

We also show that many natural hypothesis testing problems – problems to which one would naturally hope a generic theory of computational hardness would apply – can straightforwardly be transformed into hypothesis testing problems where any reasonable curve $f(t)$ which measures the free energy of solutions "at overlap $t$" *must* be non-monotonic, regardless of computational complexity of the problem. We do this by introducing artificial "overlap gaps" – ranges of $t$ where no pair of solutions can have overlap $t$, but some pairs exist with both smaller and larger overlaps.

Since the low-degree criterion for hardness remains applicable even to these problems with manufactured overlap gaps, we take this to show that any criterion based on monotonicity of some free-energy curve must apply to a narrower set of inference problems than the corresponding low-degree criterion.

**Remark 4.1.** One use case for our results is to use FP as a tool for proving low-degree lower bounds, as in Section 3.2. While the counterexamples we give here show that we cannot hope for a formal FP-to-LD connection for general Boolean-valued problems, one strategy for proving low-degree lower bounds for Boolean-valued problems is to first compare to an associated Gaussian problem (see Proposition B.1 of [BBK+21]) and then use the FP-to-LD connection for Gaussian problems (Theorem 2.4). This type of strategy is used implicitly in [BBK+21] to give low-degree lower bounds for community detection. Also, some Boolean-valued problems (such as planted clique and planted dense subgraph) fall into the framework of Section 3 and can be handled using the FP-to-LD connection in Theorem 3.7.

4.1. **The Form of the Low-Overlap and Low-Degree Likelihood Ratios for Boolean Problems.**
Throughout this section, let $H_0 = \mathsf{Rad}(\frac{1}{2})^{\otimes n}$. We'll have $H_1$ as a mixture over *biased* distributions $H_1 = \mathbb{E}_{u \sim \mu} H_u$, where $\mu$ is a distribution over *bias vectors* $u \in [-1, 1]^n$, and we sample $x \sim H_u$ by independently sampling

$$x_i = \begin{cases} 1 & \text{with probability } \frac{1}{2} + \frac{u_i}{2} \\ -1 & \text{with probability } \frac{1}{2} - \frac{u_i}{2}. \end{cases}$$

**Claim 4.2.** *For* $u, v \in [-1, 1]^n$, $\langle L_u, L_v \rangle = \prod_{i=1}^{n}(1 + u_i v_i)$.

*Proof.* By definition,

$$L_u(x) = \prod_{i \leq n} \left( \mathbb{1}(x_i = 1) \cdot \frac{1/2 + u_i/2}{1/2} + \mathbb{1}(x_i = -1) \cdot \frac{1/2 - u_i/2}{1/2} \right) = \prod_{i \leq n} (1 + x_i u_i).$$

So,

$$\langle L_u, L_v \rangle = \mathbb{E}_{x \sim H_0} \prod_{i \leq n} (1 + x_i u_i) = \prod_{i \leq n} \mathbb{E}_{x \sim H_0} (1 + x_i u_i)(1 + x_i v_i) = \prod_{i \leq n} (1 + u_i v_i). \qquad \square$$

It is not completely clear which notion of overlap to take in defining the Franz–Parisi criterion in this Boolean setting. However, our examples below will rule out any reasonable notion of overlap.

**Claim 4.3.** *In the setting where* $H_0 = \mathsf{Rad}(\frac{1}{2})^{\otimes n}$ *and* $H_1 = \mathbb{E}_{u \sim \mu} H_u$,

$$\mathrm{LD}(D) = \sum_{\substack{S \subset [n] \\ |S| \leq D}} \mathbb{E}_{u, v \sim \mu} \left[ \prod_{i \in S} u_i v_i \right].$$

*Proof.* The *Walsh–Hadamard* characters are an orthonormal basis for $L^2(H_0)$. For each $S \subseteq [n]$, the character $\chi_S$ is given by $\chi_S(x) = \prod_{i \in S} x_i$. We can express $L_u(x) = \sum_{S \subseteq [n]} \widehat{L_u}(S) \chi_S(x)$, where $\widehat{L_u}(S) = \langle L_u, \chi_S \rangle$, and $L_u^{\leq D}(x) = \sum_{S \subseteq [n], |S| \leq D} \widehat{L_u}(S) \chi_S(x)$. (For proofs of these standard facts from Boolean analysis, see e.g. [O'D14].)

Taking the inner product in the Walsh–Hadamard basis, $\langle L_u^{\leq D}, L_v^{\leq D} \rangle = \sum_{S \subseteq [n], |S| \leq D} \widehat{L_u}(S) \widehat{L_v}(S)$. Computing $\widehat{L_u}(S)$, we get $\widehat{L_u}(S) = \mathbb{E}_{x \sim H_0} \prod_{i \in [n]} (1 + x_i u_i) \cdot \prod_{i \in S} x_i = \prod_{i \in S} u_i$. Since $LD(D) = \mathbb{E}_{u, v \sim \mu} \langle L_u^{\leq D}, L_v^{\leq D} \rangle$, the claim follows. $\qquad \square$

## 4.2. Examples of Problems where FP Fails to Predict the Computational Threshold.
Both of the examples presented in this section show that in the Boolean case, the form of the inner product of likelihood ratios (Claim 4.2) enables us to make $FP(D)$ small even for easy problems.

Our first simple example shows that it is possible to have $FP(D) = 0$ for all reasonable values of $D$, and for any reasonable definition of overlap between bias vectors $u, v$ even for an easy hypothesis testing problem. Consider any distribution $\mu$ over bias vectors in $\{\pm 1\}^n$ (rather than in $[-1, 1]^n$). Then whenever $u, v \sim \mu$ are such that $u \neq v$, they disagree on at least one coordinate, so there exists some $i \in [n]$ (depending on $u, v$) where $u_i = -v_i$. This means whenever $u \neq v$,

$$\langle L_u, L_v \rangle = \prod_{j \in [n]} (1 + u_j v_j) = (1 - u_i^2) \cdot \prod_{j \in [n] \setminus \{i\}} (1 + u_i v_i) = 0.$$

Hence, for any reasonable definition of overlap between $u, v$, for any $\delta$ small enough to exclude the $u = v$ case,

$$\mathrm{LO}(\delta) \leq \mathbb{E}_{u,v \sim \mu}[\mathbb{1}_{u \neq v} \cdot \langle L_u, L_v \rangle] = 0.$$

Thus, even if $H_1$ and $H_0$ are easy to distinguish (for example, $H_1$ is uniform over $\{u \in \{\pm 1\} : \sum_{i \in [n]} u_i = 0.9n\}$), the Franz–Parisi criterion will predict that the problem is hard for $D = n^{\Omega(1)}$.

The assumption of $u \in \{\pm 1\}$ is quite artificial, but next we will see that it is not necessary; a more "noisy" version of the problem, in which $H_1$ is not a mixture around point masses but rather a mixture of biased product measures with reasonable variance, will still exhibit the same qualitative behavior. After that, we'll show how to embed this problem into a natural planted problem: a variant of densest subgraph in the easy regime.

4.2.1. *Positively biased product measures.* For any $\varepsilon, \delta \in (0, 1)$, consider the following prior $\mu$ over $u$: sample $u_i = \varepsilon$ with probability $\frac{1}{2} + \frac{1}{2}\delta$ and $u_i = -\varepsilon$ otherwise. For sake of illustration, in the following lemmas we take $\langle u, v \rangle$ to be the definition of overlap of bias vectors $u, v$, so that FP agrees with the definition used in the rest of this paper. However, we believe that a qualitatively similar statement holds for any reasonable definition of overlap.

**Lemma 4.4.** *In the model above, for any $\alpha > 0$, if $\varepsilon = n^{-1/4 + 2\alpha}$ and $\delta = n^{-1/4 + \alpha}$ and $D \ll n^\alpha$, $FP(D) = \exp(-O(n^{8\alpha}))$ but $\mathrm{LD}(D) \geq n^{6\alpha}$. Furthermore, given $x$ sampled from either $H_0$ or $H_1$, the statistic $\langle x, \vec{1} \rangle$ distinguishes between the models with error probability $o(1)$.*

*Proof.* First, we show that in this parameter setting, a successful test statistic exists. The test statistic $\langle x, \vec{1} \rangle$ is distributed as $2 \left( \mathrm{Bin}(n, \frac{1}{2}) - \frac{n}{2} \right)$ for $x \sim H_0$, and it is distributed as $2\varepsilon \left( \mathrm{Bin}(n, \frac{1}{2} + \frac{1}{2}\delta) - \frac{n}{2} \right)$ for $x \sim H_1$. Since in our setting $\mathbb{E}_{H_1}[\langle x, 1 \rangle] = \varepsilon \delta n = n^{1/2 + 3\alpha} \gg \sqrt{n}$ and $\mathbb{E}_{H_0}[\langle x, 1 \rangle] = 0$, $\langle x, \vec{1} \rangle$ takes value at most $\sqrt{n \log n}$ under $H_0$ with probability $1 - O(1/n)$, and at least $n^{1/2 + 2\alpha}$ with probability $1 - o(1/n)$, so thresholding on $\langle x, 1 \rangle$ gives a hypothesis test which succeeds with high probability.

The value of $FP(D)$ and $\mathrm{LD}(D)$ follow as corollaries of Claims 4.5 and 4.6 below.

**Claim 4.5.** *If $\varepsilon, \delta \in (0, 1)$ satisfy $\varepsilon^2 \gg \max \left( \delta^2, \sqrt{\frac{D}{n}} \right)$, then $FP(D) \leq \exp \left( -O(n\varepsilon^4) \right)$.*

*Proof.* If $u, v$ agree on $\frac{n}{2} + \Delta$ coordinates (which is equivalent to $\langle u, v \rangle = \varepsilon^2 \cdot 2\Delta$), then by Claim 4.2,

$$\langle L_u, L_v \rangle = (1 + \varepsilon^2)^{\frac{n}{2} + \Delta} (1 - \varepsilon^2)^{\frac{n}{2} - \Delta}$$

$$= (1 - \varepsilon^4)^{n/2} \cdot \left( \frac{1 + \varepsilon^2}{1 - \varepsilon^2} \right)^\Delta.$$

For $u, v \sim \mu$, we have that $\langle u, v \rangle \sim 2\varepsilon^2 \left( \mathrm{Bin}(n, \frac{1}{2} + \frac{1}{2}\delta^2) - \frac{n}{2} \right)$. Now, applying the Chernoff bound for a sum of independent Bernoulli random variables,

$$\Pr_{u,v \sim \mu} \left[ \left| \frac{\langle u, v \rangle}{\varepsilon^2} - \delta^2 n \right| \geq C\sqrt{n} \right] = \Pr_{X \sim \mathrm{Bin}(n, \frac{1}{2} + \frac{1}{2}\delta^2)} \left[ |X - \mathbb{E}[X]| \geq \tfrac{1}{2} C\sqrt{n} \right] \leq \exp \left( -\frac{C^2}{2(1 - \delta^4)} \right).$$

Taking our notion of overlap to be $\langle u, v \rangle$, note that if $\delta_0$ satisfies $\Pr(|\langle u, v \rangle| > \delta_0) < e^{-D}$ then $\delta_0 > \delta(D)$ and hence $\mathrm{FP}(D) \leq \mathrm{LO}(\delta_0)$. So we have

$$\mathrm{FP}(D) \leq \mathbb{E}_{u,v\sim\mu} \left[ \mathbb{1}_{|\langle u,v\rangle| > \varepsilon^2 \delta^2 n + \varepsilon^2 \sqrt{2D(1-\delta^4)n}} \cdot \langle L_u, L_v \rangle \right] \leq (1-\varepsilon^4)^{n/2} \left( \frac{1+\varepsilon^2}{1-\varepsilon^2} \right)^{\delta^2 n + \sqrt{2Dn}},$$

and taking logarithms,

$$\log(\mathrm{FP}(D)) \leq n \cdot \left( \tfrac{1}{2} \log(1-\varepsilon^4) + \left( \delta^2 + \sqrt{\tfrac{2D}{n}} \right) \log \left( \frac{1+\varepsilon^2}{1-\varepsilon^2} \right) \right)$$

$$= n \cdot \left( -\Omega(\varepsilon^4) + O\left( \left( \delta^2 + \sqrt{\tfrac{D}{n}} \right) \varepsilon^2 \right) \right),$$

where we have used a first-order Taylor expansion to $\log(1+x)$. Thus so long as $\varepsilon^2 \gg \max\left( \delta^2, \sqrt{\tfrac{D}{n}} \right)$, $\mathrm{FP}(D) \leq \exp(-nO(\varepsilon^4))$. This completes the proof. $\qquad \square$

While the definition of the overlap of $u, v$ as $\langle u, v \rangle$ is just one of many possible choices in the Boolean setting, we note that any definition of overlap which would count only only "typical" pairs $u, v$ which agree on a $n(\tfrac{1}{2} + \tfrac{\delta^2}{2}) \pm O(\sqrt{n})$ fraction of coordinates as having small-enough overlap to be counted when computing $\mathrm{FP}(D)$ for small $D$ would have led to the same outcome.

**Claim 4.6.** *For the model specified above,* $\mathrm{LD}(D) \geq \mathrm{LD}(1) = n\delta^2\varepsilon^2$.

*Proof.* In our model, $\mathbb{E}_{u\sim\mu} u_i = \varepsilon\delta$ for every $i \in [n]$. So we compute directly from Claim 4.3,

$$\mathrm{LD}(1) = \sum_{i\in[n]} \mathbb{E}_{u,v\sim\mu} u_i v_i = n \cdot (\mathbb{E}_{u\sim\mu} u_1)^2 = n\delta^2\varepsilon^2. \qquad \square$$

This concludes the proof of the lemma. $\qquad \square$

4.2.2. *Planted dense-and-sparse subgraph.* Next we show that FP can mis-predict the computational threshold even for a familiar-looking planted subgraph problem. Consider the following problem: $H_0$ is uniform over $\{\pm 1\}^{\binom{n}{2}}$, i.e., $H_0$ is the Erdos-Renyi distribution $G(n, 1/2)$. The alternate distribution $H_1$ is uniform over signed adjacency matrices of $n$-vertex graphs containing a planted *dense* subgraph of size $\delta n$ and a planted *sparse* subgraph of size $c\delta n$ for $c < 1$. That is, we take $\mu$ uniform over the upper-triangular restriction of matrices of the form $u = \mathrm{upper}(0.9 \cdot 1_S 1_S^\top - 0.9 \cdot 1_T 1_T^\top)$, where $S$ is a subset of $[n]$ chosen by including every $i \in S$ independently with probability $\delta$, and $T$ is a subset of $[n]$ chosen by including every $i \in T$ independently with probability $c\delta$.

**Lemma 4.7.** *When* $c = 0.9$, $\delta = n^{-1/10}$, *and* $D \ll n^{0.2}$, *the testing problem* $H_0$ *vs* $H_1$ *is easy,* $\mathrm{LD}(D) = \Omega(n^{9/5})$, *but* $\mathrm{FP}(D) = \exp(-\Omega(n^{8/5}))$.

*Proof.* To see that the testing problem is easy in this regime, consider the test statistic given a sample $x$ from either $H_0$ or $H_1$ which is the maximum eigenvalue of the matrix $A(x)$ whose $(i,j)$ entry is given by $x_{ij}$ (or $x_{ji}$). Under $H_0$, $\lambda_{\max}(A(x)) = O(\sqrt{n})$ with high probability. However under $H_1$, $\lambda_{\max}(A(x)) \geq 1_S^\top A(x) 1_S / |S|$, and the final quantity is at least $\Omega((\delta n)^2) = \Omega(n^{9/5})$ with high probability using standard estimates. Hence the maximum eigenvalue of $A(x)$ furnishes a test that succeeds with high probability.

To bound the values of $\mathrm{LD}(D)$ and $\mathrm{FP}(D)$, we turn to the following claims:

**Claim 4.8.** *Suppose* $\delta = o(1)$, $c \in (0.43, 2.32)$ *and* $D \ll \delta^8 n$. *Then there exists a constant* $a > 0$ *such that* $\mathrm{FP}(D) \leq \exp\left(-a\delta^4 n^2\right)$.

*Proof.* Let $S'_u = S_u \setminus T_u$ and similarly for $T'_u$. For $u, v$ where $|S'_u \cap S'_v| = \alpha n$, $|T'_u \cap T'_v| = \beta n$, $|S'_u \cap T'_v| = \gamma n$, and $|T'_u \cap S'_v| = \eta n$,

$$\langle L_u, L_v \rangle = \prod_{(i,j)\in\binom{[n]}{2}} (1 + u_{ij} v_{ij}) = (1.81)^{\binom{\alpha n}{2} + \binom{\beta n}{2}} (0.19)^{\binom{\gamma n}{2} + \binom{\eta n}{2}}.$$

We have that $\mathbb{E}|S'_u \cap S'_v| = \delta^2(1-c\delta)^2 n$, $\mathbb{E}|T'_u \cap T'_v| = c^2\delta^2(1-\delta)^2 n$ and $\mathbb{E}|T'_u \cap S'_v| = \mathbb{E}|S'_v \cap T'_u| = c\delta^2(1-c\delta)(1-\delta)n$. From standard concentration arguments, the sizes of each of these sets is within an additive $\sqrt{Dn}$ with probability at least $1 - \exp(-O(D^2))$. Hence, for definition of overlap between pairs $(S_0, T_0), (S_1, T_1)$ such that the overlaps accounted for in $\mathrm{FP}(D)$ for small $D$ includes only pairs falling within this tolerance,

$$\log\left(\mathrm{FP}(D)\right) \leq \log\left((1.81)^{\frac{1}{2}(1-c\delta)^4\delta^4 n^2 + \frac{1}{2}c^4\delta^4(1-\delta)^4 + O(n^{3/2}\sqrt{D})}(0.19)^{c^2\delta^4(1-c\delta)^2(1-\delta)^2 n^2 - O(n^{3/2}\sqrt{D})}\right)$$

$$\leq \delta^4 n^2 \cdot \left(\frac{1}{2}\left((1-c\delta)^4 + c^4(1-\delta)^4 + O(\frac{\sqrt{D}}{\delta^4\sqrt{n}})\right)\log(1.81) + (c^2(1-c\delta)^2(1-\delta)^2 - O(\frac{\sqrt{D}}{\delta^4\sqrt{n}}))\log(0.19)\right),$$

and the quantity within the parenthesis is a negative constant so long as $D \ll \delta^8 n$ and $c \in (0.43, 2.32)$. This concludes the proof. $\square$

**Claim 4.9.** $\mathrm{LD}(D) = \Omega((\delta(1-c)n)^2)$.

*Proof.* We have that $\mathbb{E}_{u\sim\mu}u_{ij} = 0.9(1-c)\delta$. Hence,

$$\mathrm{LD}(D) \geq \mathrm{LD}(1) = \sum_{(i,j)\in\binom{[n]}{2}} \mathbb{E}_{u,v\sim\mu}u_{ij}v_{ij} = \binom{n}{2}0.81(1-c)^2\delta^2. \qquad \square$$

This completes the proof of the lemma. $\square$

**Remark 4.10.** We note that the planted dense-and-sparse subgraph problem can be put into the framework of Section 3 (specifically Assumption 3.1) and so we have the FP-to-LD implication from Theorem 3.7. However, to do this, we need to let $u$ encode the set of vertices in the union of the two subgraphs, but not the choice of which vertices belong to the dense subgraph and which belong to the sparse one — this choice is instead absorbed into $\mathbb{P}_u$. This alters the notion of overlap sufficiently that the FP curve accurately reflects the computational complexity of the problem.

### 4.3. Pruning to Achieve Sparse Support of the Overlap Distribution.
We turn to our last family of examples, constructing problems where the distribution of overlaps has "gaps" in its support, regardless of the computational complexity of the problem. In fact, this follows from a simple lemma:

**Lemma 4.11** (Subsampling prior distributions). *Let $D$ be a probability distribution and $E$ be an event in the probability space corresponding to $D \otimes D$, with $\Pr_{x,y\sim D}((x,y) \in E) \leq \delta$, and such that $(x,x) \notin E$ for all $x$ in the support of $D$. Then the uniform distribution $D'$ over $\Omega(1/\sqrt{\delta})$ samples from $D$ satisfies $\Pr_{x,y\sim D'}((x,y) \in E) = 0$ with probability at least $0.99$.*

*Proof.* There are at most $T^2$ distinct pairs of draws $x, y \sim D$ in a list of $T$ independent draws $x_1, \ldots, x_T$; by a union bound the probability that any $(x_i, x_j)$ is in $E$ is at most $\delta T^2$. $\square$

Now let us sketch an example hypothesis testing problems with a "manufactured" overlap gap, using Lemma 4.11.

*Planted clique with artificial overlap gap.* Here $H_0$ is $G(n, 1/2)$ and $H_1$ is $G(n, 1/2)$ with a randomly-added $\approx k$-clique. (For simplicity, consider the model where each vertex of the clique is added independently with probability $k/n$.) A natural measure of the overlap of two potential $k$-cliques $S, T \subseteq [n]$, $|S| = |T| = k$, is $|S \cap T|$.

Consider the event $E$ that $|S \cap T| \in [k^\delta, k)$. By standard Chernoff bounds,

$$\Pr(|S \cap T| > k^\delta) \leq e^{-\Omega(k^\delta)}$$

so long as $k \leq n^{1/2 + O(\delta)}$. Applying Lemma 4.11, we see that there is a distribution $D'$ on size $\approx k$ subsets of $[n]$ for which no pair has overlap between $k^\delta$ and $k$ and which has support size $e^{\Omega(k^\delta)}$. Then we can create a new planted problem, $H_0$ versus $H'_1$, where $H_0$ is as before and $H'_1$ plants a clique on a randomly-chosen set of vertices from $D'$.

We note a few features of this construction. (1) Since this construction allows for $k$ to be either smaller or larger than $\sqrt{n}$, overlap gaps like this can be introduced in both the computationally easy and hard regimes of planted clique. And, (2), for $k = \text{poly}(n)$, the size of the support of the prior distribution in $H_1'$ is still $2^{\text{poly}(n)}$, meaning that we have not trivialized the planted problem. Finally, (3), it is hopefully clear that there was nothing special here about planted clique; this approach applies easily to other planted subgraph problems, spiked matrix and tensor models, and so on.

## 5. Proofs for the Gaussian Additive Model

5.1. **Basic Facts.** First we prove Remark 1.6, which contains some basic facts about the quantity $\delta$ in the definition of FP.

*Proof of Remark 1.6.* For convenience, we recall the definition
$$\delta := \sup \{\varepsilon \geq 0 \text{ s.t. } \Pr_{u,v \sim \mu}(|\langle u, v\rangle| \geq \varepsilon) \geq e^{-D}\}.$$

By definition of supremum, for any $\delta' < \delta$ we have $\Pr(|\langle u, v\rangle| \geq \delta') \geq e^{-D}$. Using continuity of measure,
$$\Pr(|\langle u, v\rangle| \geq \delta) = \Pr\left(\cap_{\delta' < \delta}\{|\langle u, v\rangle| \geq \delta'\}\right) = \lim_{\delta' \uparrow \delta} \Pr(|\langle u, v\rangle| \geq \delta') \geq e^{-D},$$

as desired.

Now we prove the second statement. By definition of supremum, for any $\delta' > \delta$ we have $\Pr(|\langle u, v\rangle| \geq \delta') < e^{-D}$. Using continuity of measure,
$$\Pr(|\langle u, v\rangle| > \delta) = \Pr\left(\cup_{\delta' > \delta}\{|\langle u, v\rangle| \geq \delta'\}\right) = \lim_{\delta' \downarrow \delta} \Pr(|\langle u, v\rangle| \geq \delta') \leq e^{-D},$$

as desired. $\qquad\square$

Recall the quantities $\text{LD}(D, \lambda)$ and $\text{FP}(D, \lambda)$ from (13) and (14). We now state some associated monotonicity properties.

**Lemma 5.1.** *For any fixed $\lambda$, we have that $\text{LD}(D, \lambda)$ and $\text{FP}(D, \lambda)$ are both monotone increasing in $D$. For any fixed $D$, we have that $\text{LD}(D, \lambda)$ is monotone increasing in $\lambda$.*

*Proof.* To see that LD is increasing in $D$, recall the definition (1) and note that projecting onto a larger subspace can only increase the 2-norm of the projection.

To see that FP is increasing in $D$, recall the definition (3), note that $\delta(D)$ is increasing in $D$, and note that $\langle L_u, L_v\rangle_{\mathbb{Q}} \geq 0$ because (being likelihood ratios) $L_u$ and $L_v$ are nonnegative-valued functions.

To see that LD is increasing in $\lambda$, start with (13) and expand
$$\text{LD}(D, \lambda) = \sum_{d=0}^{D} \frac{\lambda^{2d}}{d!} \mathbb{E}_s[s^d],$$

where $s = \langle u, v\rangle$ is the overlap random variable from (12). Since $\mathbb{E}[s^d] \geq 0$ for all integers $d \geq 0$ (see Corollary 5.3 below), this is increasing in $\lambda$. $\qquad\square$

For the next result, we need to introduce some new notation. Let $V^{\leq D}$ denote the space of polynomials $\mathbb{R}^N \to \mathbb{R}$ of degree at most $D$. Also define $V^{=D} = V^{\leq D} \cap (V^{\leq(D-1)})^{\perp}$ where $\perp$ denotes orthogonal complement (with respect to $\langle \cdot, \cdot \rangle_{\mathbb{Q}}$). We have already defined $f^{\leq D}$ to mean the orthogonal projection of $f$ onto $V^{\leq D}$, and we similarly define $f^{=D}$ to be the orthogonal projection of $f$ onto $V^{=D}$. In the Gaussian additive model, $V^{=D}$ is spanned by the multivariate Hermite polynomials of degree exactly $D$. The following extension of Proposition 2.3 is implicit in the proof of Theorem 2.6 in [KWB19].

**Proposition 5.2.** *In the Gaussian additive model, we have the formula*
$$\langle L_u^{=D}, L_v^{=D}\rangle_{\mathbb{Q}} = \exp^{=D}(\lambda^2 \langle u, v\rangle),$$

*where $\exp^{=D}(x) := \frac{x^D}{D!}$*

The following corollary is not specific to the Gaussian additive model, yet curiously can be proved via degree-$D$ projections in the Gaussian additive model.

**Corollary 5.3.** *Let $s = \langle u, v \rangle$ denote the overlap random variable, with $u, v$ drawn independently from some distribution $\mu$ on $\mathbb{R}^N$ with all moments finite. For any integer $d \geq 0$, we have $\mathbb{E}[s^d] \geq 0$.*

*Proof.* Consider the Gaussian additive model with prior $\mu$ and some SNR $\lambda > 0$. Using Proposition 5.2,

$$0 \leq \left\| \mathbb{E}_{u \sim \mu} L_u^{=d} \right\|_{\mathbb{Q}}^2 = \mathbb{E}_{u,v \sim \mu} \left[ \langle L_u^{=d}, L_v^{=d} \rangle_{\mathbb{Q}} \right] = \mathbb{E}_s \left[ \exp^{=d}(\lambda^2 s) \right] = \frac{\lambda^{2d}}{d!} \mathbb{E}_s[s^d],$$

which yields the result. $\square$

Next we state some basic properties of the function $\exp^{\leq D}(\cdot)$.

**Lemma 5.4.** *Let $\exp^{\leq D}(\cdot)$ be defined as in (11) for some integer $D \geq 0$.*

- *If $D$ is odd then $\exp^{\leq D}(x) \leq \exp(x)$ for all $x \in \mathbb{R}$.*
- *If $D$ is even then*

$$\exp^{\leq D}(x) \leq \exp(x) \ \text{ for all } x \geq 0, \qquad \text{and} \qquad \exp^{\leq D}(x) > \exp(x) \ \text{ for all } x < 0.$$

*Proof.* For $x \geq 0$, both results are immediate because every term in the Taylor expansion (11) is nonnegative and the series converges to $\exp(x)$.

For $x < 0$, we will prove the following statement by induction on $D$: $\exp^{\leq D}(x) < \exp(x)$ for all $x < 0$ when $D$ is odd, and $\exp^{\leq D}(x) > \exp(x)$ for all $x < 0$ when $D$ is even. The base case $D = 0$ is easily verified. The induction step can be deduced from the fact

$$\frac{d}{dx} \left[ \exp(x) - \exp^{\leq D}(x) \right] = \exp(x) - \exp^{\leq (D-1)}(x)$$

along with the fact $\exp(0) = \exp^{\leq D}(0)$. $\square$

**Corollary 5.5.** *If $D$ is even then $\exp^{\leq D}(x) \geq 0$ for all $x \in \mathbb{R}$.*

*Proof.* For $x \geq 0$ this is clear because every term in the Taylor expansion (11) is nonnegative. For $x \leq 0$, Lemma 5.4 implies $\exp^{\leq D}(x) \geq \exp(x) \geq 0$. $\square$

Finally, we will need the following standard bounds on the factorial. These appeared in [Knu97] (Section 1.2.5, Exercise 24), and the proof can be found in [Pro].

**Proposition 5.6.** *For any integer $n \geq 1$,*

$$\frac{n^n}{e^{n-1}} \leq n! \leq \frac{n^{n+1}}{e^{n-1}}.$$

5.2. **Proof of Theorem 2.5: LD-Hard Implies FP-Hard.** We first prove Corollary 2.6, a straightforward consequence of Theorem 2.5 under certain asymptotic assumptions.

*Proof of Corollary 2.6.* Fix any constant $\varepsilon \in (0, \varepsilon']$. Recall LD is monotone increasing in $\lambda$ (see Lemma 5.1). For all sufficiently large $n$, our assumptions on the scaling regime imply that $D \geq D_0(\varepsilon)$ and (15) holds, so (16) holds. In other words, $\limsup_n [\mathrm{FP}(D, \lambda) - \mathrm{LD}(D, (1 + \varepsilon')\lambda)] \leq \varepsilon$. Since $\varepsilon$ was arbitrary, $\limsup_n [\mathrm{FP}(D, \lambda) - \mathrm{LD}(D, (1 + \varepsilon')\lambda)] \leq 0$ as desired. $\square$

The remainder of this section is devoted to the proof of Theorem 2.5.

*Proof of Theorem* 2.5. Define $\hat{\lambda} := (1+\varepsilon)\lambda$, $\tilde{\lambda} := (1+\varepsilon^2/4)\lambda$, and $C := \mathrm{LD}(D,\hat{\lambda})$. Define $\delta = \delta(D)$ as in (3), which implies $\Pr(|s| \geq \delta) \geq e^{-D}$ (see Remark 1.6). Recall the overlap random variable $s$ from (12). We will first prove an upper bound on $\delta$ in terms of $\mathrm{LD}(D,\hat{\lambda})$. Since $\mathbb{E}_s[\exp^{=d}(\hat{\lambda}^2 s)] \geq 0$ for all $d$ (see the proof of Corollary 5.3),

$$C = \mathrm{LD}(D,\hat{\lambda}) = \underset{s}{\mathbb{E}}[\exp^{\leq D}(\hat{\lambda}^2 s)] \geq \underset{s}{\mathbb{E}}[\exp^{=D}(\hat{\lambda}^2 s)] = \underset{s}{\mathbb{E}}\frac{1}{D!}(\hat{\lambda}^2 s)^D.$$

Using $\Pr(|s| \geq \delta) \geq e^{-D}$ and the fact that $D$ is even,

$$\underset{s}{\mathbb{E}}\frac{1}{D!}(\hat{\lambda}^2 s)^D \geq e^{-D}\frac{1}{D!}(\hat{\lambda}^2 \delta)^D.$$

Combining this with the above yields $(\hat{\lambda}^2 \delta)^D \leq Ce^D D!$ and so, using the factorial bound in Proposition 5.6,

$$\delta \leq \hat{\lambda}^{-2} C^{1/D} e(D!)^{1/D} \leq \hat{\lambda}^{-2} C^{1/D} e \left(\frac{D^{D+1}}{e^{D-1}}\right)^{1/D} = \frac{D}{\hat{\lambda}^2}(CeD)^{1/D}.$$

Using (15),

$$(CeD)^{1/D} \leq 1+\varepsilon \leq \frac{(1+\varepsilon)^2}{(1+\varepsilon^2/4)^2} = \frac{\hat{\lambda}^2}{\tilde{\lambda}^2},$$

and so we conclude

(25) $$\delta \leq \frac{D}{\tilde{\lambda}^2}.$$

In Lemma 5.7 below, we establish for all $s \in [-\delta, \delta]$,

$$\exp(\lambda^2 s) \leq \exp^{\leq D}(\tilde{\lambda}^2 s) + \varepsilon.$$

As a result,

$$\mathrm{FP}(D,\lambda) = \underset{s}{\mathbb{E}}\left[\mathbb{1}_{|s| \leq \delta}\exp(\lambda^2 s)\right] \leq \underset{s}{\mathbb{E}}\left[\exp^{\leq D}(\tilde{\lambda}^2 s)\right] + \varepsilon = \mathrm{LD}(D,\tilde{\lambda}) + \varepsilon \leq \mathrm{LD}(D,\hat{\lambda}) + \varepsilon,$$

where we have used the fact $\exp^{\leq D}(x) \geq 0$ for $D$ even (Corollary 5.5) and monotonicity of LD in $\lambda$ (Lemma 5.1). □

**Lemma 5.7.** *For an appropriate choice of $D_0 = D_0(\varepsilon)$, we have for all $s \in [-\delta, \delta]$,*

$$\exp(\lambda^2 s) \leq \exp^{\leq D}(\tilde{\lambda}^2 s) + \varepsilon.$$

*Proof.* We will split into various cases depending on the value of $s$.

**Case I**: $s \leq -\lambda^{-2}\log(1/\varepsilon)$. We have

$$\exp(\lambda^2 s) \leq \exp[\lambda^2 \cdot (-\lambda^{-2}\log(1/\varepsilon))] = \varepsilon,$$

which suffices because $\exp^{\leq D}(x) \geq 0$ for even $D$ (Corollary 5.5).

**Case II**: $-\lambda^{-2}\log(1/\varepsilon) < s \leq 0$. We have

$$\exp(\lambda^2 s) = \exp(\tilde{\lambda}^2 s) + \exp(\tilde{\lambda}^2 s)[\exp(\lambda^2 s - \tilde{\lambda}^2 s) - 1].$$

Since $s \leq 0$ and $D$ is even, Lemma 5.4 gives $\exp(\tilde{\lambda}^2 s) \leq \exp^{\leq D}(\tilde{\lambda}^2 s)$. For the second term, recalling $\tilde{\lambda} = (1+\varepsilon^2/4)\lambda$ and $-\lambda^2 s \leq \log(1/\varepsilon)$,

$$\exp(\tilde{\lambda}^2 s)[\exp(\lambda^2 s - \tilde{\lambda}^2 s) - 1] \leq 1 \cdot [\exp(-\lambda^2 s(\varepsilon^2/2 + \varepsilon^4/16)) - 1]$$
$$\leq \exp[\log(1/\varepsilon)(9\varepsilon^2/16)] - 1$$
$$\leq \exp(9\varepsilon/16) - 1.$$

Using the bound $\exp(x) \leq 1 + (e-1)x$ for $x \in [0,1]$, the above is at most $\frac{9}{16}(e-1)\varepsilon \leq \varepsilon$.

**Case III**: $0 < s \leq D/(2e\tilde{\lambda}^2)$. Using the Taylor series for exp,

$$\exp(\lambda^2 s) \leq \exp(\tilde{\lambda}^2 s) = \exp^{\leq D}(\tilde{\lambda}^2 s) + \sum_{d=D+1}^{\infty} \frac{(\tilde{\lambda}^2 s)^d}{d!}.$$

Using the factorial bound (Proposition 5.6) and $s \leq D/(2e\tilde{\lambda}^2)$,

$$\sum_{d=D+1}^{\infty} \frac{(\tilde{\lambda}^2 s)^d}{d!} \leq \sum_{d=D+1}^{\infty} \frac{1}{e}\left(\frac{e\tilde{\lambda}^2 s}{d}\right)^d \leq \frac{1}{e}\sum_{d=D+1}^{\infty} \left(\frac{1}{2}\right)^d = \frac{1}{e \cdot 2^D},$$

which can be made smaller than $\varepsilon$ by choosing $D_0$ sufficiently large.

**Case IV**: $D/(2e\tilde{\lambda}^2) < s \leq \delta$. Let $d = \lceil \tilde{\lambda}^2 s \rceil$ and note that $\frac{D}{2e} \leq d \leq D$ due to (25) and the assumption on $s$. Again using the factorial bound (Proposition 5.6),

$$\exp^{\leq D}(\tilde{\lambda}^2 s) \geq \frac{1}{d!}(\tilde{\lambda}^2 s)^d \geq \frac{1}{ed}\left(\frac{e\tilde{\lambda}^2 s}{d}\right)^d = \frac{1}{ed}\left(\frac{e\tilde{\lambda}^2 s}{\lceil \tilde{\lambda}^2 s \rceil}\right)^{\lceil \tilde{\lambda}^2 s \rceil} \geq \frac{1}{eD}\left(\frac{e\tilde{\lambda}^2 s}{\tilde{\lambda}^2 s + 1}\right)^{\tilde{\lambda}^2 s}$$

$$= \frac{1}{eD}\left(\frac{e}{1 + \frac{1}{\tilde{\lambda}^2 s}}\right)^{\tilde{\lambda}^2 s} = \exp\left[\tilde{\lambda}^2 s\left(1 - \frac{1 + \log D}{\tilde{\lambda}^2 s} - \log\left(1 + \frac{1}{\tilde{\lambda}^2 s}\right)\right)\right].$$

Since $\tilde{\lambda}^2 s \geq D/(2e)$ by assumption, we conclude

$$\exp^{\leq D}(\tilde{\lambda}^2 s) \geq \exp\left[\tilde{\lambda}^2 s\left(1 - 2e \cdot \frac{1 + \log D}{D} - \log\left(1 + \frac{2e}{D}\right)\right)\right].$$

Since $\tilde{\lambda} > \lambda$, this can be made larger than $\exp(\lambda^2 s)$ by choosing $D_0$ sufficiently large. □

## 5.3. **Proof of Theorem 2.16: FP-Hard Implies Free Energy Barrier.**

*Proof of Theorem 2.16.* Let $b = (1 + \varepsilon)\delta$ denote the maximum possible value of $\langle u, v \rangle$ for $v \in B$, and let $a = -\sigma\delta$ denote the minimum possible value of $\langle u, v \rangle$ for $v \in A$, where $\sigma = 0$ if $S$ has nonnegative overlaps and $\sigma = 1$ otherwise). Since the Hamiltonian decomposes as

$$-H(v) = \langle v, Y \rangle = \langle v, \lambda u + Z \rangle = \lambda\langle u, v \rangle + \langle v, Z \rangle,$$

we can write

$$\frac{\nu_\beta(B)}{\nu_\beta(A)} = \frac{\sum_{v \in B} \exp(-\beta H(v))}{\sum_{v \in A} \exp(-\beta H(v))} = \frac{\sum_{v \in B} \exp(\beta\lambda\langle u, v \rangle + \beta\langle v, Z \rangle)}{\sum_{v \in A} \exp(\beta\lambda\langle u, v \rangle + \beta\langle v, Z \rangle)}$$

$$\leq \exp(\beta\lambda(b - a))\frac{\sum_{v \in B} \exp(\beta\langle v, Z \rangle)}{\sum_{v \in A} \exp(\beta\langle v, Z \rangle)} = \exp(\beta\lambda\delta(1 + \sigma + \varepsilon))\frac{\tilde{\nu}_\beta(B)}{\tilde{\nu}_\beta(A)}$$

where $\tilde{\nu}_\beta(v) \propto \exp(-\beta\tilde{H}(v))$ is the Gibbs measure associated with the "pure noise" Hamiltonian $\tilde{H}(v) = -\langle v, Z \rangle$. Letting $A^c = S \setminus A$ denote the complement of $A$, we have

$$\frac{\tilde{\nu}_\beta(B)}{\tilde{\nu}_\beta(A)} \leq \frac{\tilde{\nu}_\beta(A^c)}{1 - \tilde{\nu}_\beta(A^c)},$$

so it remains to bound $\tilde{\nu}_\beta(A^c)$.

We next claim that

$$(26) \qquad \mathbb{E}_Z[\tilde{\nu}_\beta(v)] = \mathbb{E}_Z[\tilde{\nu}_\beta(v')] \qquad \text{for all } v, v' \in S.$$

To see this, let $R \in \mathrm{O}(N)$ be the orthogonal matrix such that $Rv = v'$ and $RS = S$ (guaranteed by transitive symmetry) and write

$$(27) \qquad \tilde{\nu}_\beta(v) = \frac{\exp(\beta\langle v, Z \rangle)}{\sum_{w \in S} \exp(\beta\langle w, Z \rangle)},$$

$$(28) \qquad \tilde{\nu}_\beta(v') = \frac{\exp(\beta\langle v', Z \rangle)}{\sum_{w \in S} \exp(\beta\langle w, Z \rangle)} = \frac{\exp(\beta\langle Rv, Z \rangle)}{\sum_{w \in S} \exp(\beta\langle Rw, Z \rangle)} = \frac{\exp(\beta\langle v, R^\top Z \rangle)}{\sum_{w \in S} \exp(\beta\langle w, R^\top Z \rangle)}.$$

By rotational invariance of $Z$, (27) and (28) have the same distribution, which proves (26). Since $\tilde{\nu}_\beta$ is a normalized measure, we must in fact have $\mathbb{E}_Z[\tilde{\nu}_\beta(v)] = 1/|S|$ for every $v \in S$, and so (using linearity of expectation)

$$\mathbb{E}_Z[\tilde{\nu}_\beta(A^c)] = \frac{|A^c|}{|S|}.$$

Note that $|A^c|/|S|$ is simply $\Pr_{v \sim \mu}(|\langle u, v \rangle| > \delta)$, which by transitive symmetry is the same as $\Pr_{v,v' \sim \mu}(|\langle v, v' \rangle| > \delta)$, which by Remark 1.6 is at most $e^{-D}$. By Markov's inequality,

$$\Pr_Z\left(\tilde{\nu}_\beta(A^c) \geq e^{-(1-\varepsilon)D}\right) \leq e^{-\varepsilon D}.$$

Putting it all together, we have now shown that with probability at least $1 - e^{-\varepsilon D}$ over $Z$,

$$(29) \qquad \frac{\nu_\beta(B)}{\nu_\beta(A)} \leq \exp(\beta\lambda\delta(1 + \sigma + \varepsilon))\frac{\tilde{\nu}_\beta(A^c)}{1 - \tilde{\nu}_\beta(A^c)} \leq \exp(\beta\lambda\delta(1 + \sigma + \varepsilon)) \cdot 2e^{-(1-\varepsilon)D}.$$

The next step is to relate this to FP. Define $\tilde{D} = D + \log 2$ and $\tilde{\delta} = \delta(\tilde{D})$ as in (3) so that (by Remark 1.6) $\Pr_{v,v' \sim \mu}(|\langle v, v' \rangle| > \tilde{\delta}) \leq e^{-\tilde{D}} = \frac{1}{2}e^{-D}$. Also from Remark 1.6 we have $\Pr_{v,v' \sim \mu}(|\langle v, v' \rangle| \geq \delta) \geq e^{-D}$, so we conclude $\Pr_{v,v' \sim \mu}(|\langle v, v' \rangle| \in [\delta, \tilde{\delta}]) \geq \frac{1}{2}e^{-D}$. This means

$$(30) \qquad \mathrm{FP}(\tilde{D}, \tilde{\lambda}) = \mathbb{E}_{v,v' \sim \mu}\left[\mathbb{1}_{|\langle v,v' \rangle| \leq \tilde{\delta}} \cdot \exp(\tilde{\lambda}^2 \langle v, v' \rangle)\right] \geq \frac{1}{2}e^{-D} \cdot \exp(\tilde{\lambda}^2 \delta).$$

Now comparing (29) with (30) and using the choice $\tilde{\lambda}^2 = \beta\lambda(1 + \sigma + \varepsilon)/(1 - 2\varepsilon)$, we have

$$\frac{\nu_\beta(B)}{\nu_\beta(A)} \leq 2\exp[\beta\lambda\delta(1 + \sigma + \varepsilon) - (1 - \varepsilon)D]$$

$$= 2\exp[(1 - 2\varepsilon)(\tilde{\lambda}^2\delta - D) - \varepsilon D]$$

$$\leq 2\left(2 \cdot \mathrm{FP}(\tilde{D}, \tilde{\lambda})\right)^{1-2\varepsilon} e^{-\varepsilon D}$$

as desired. $\qquad \square$

## 6. Proofs for Sparse Regression

### 6.1. **Proof of Theorem 3.10(a): Lower Bound.**

6.1.1. *Conditional Low-Degree Calculation.* As discussed in Section 3.2, our low-degree lower bound will involve a *conditional* low-degree calculation where we bound LD for a modified testing problem $\mathbb{P}|A$ versus $\mathbb{Q}$, for a particular high-probability event $A$. In this section, we lay down some of the basic foundations for this approach.

Our ultimate goal will be to rule out weak separation (see Definition 1.8) for the original (non-conditioned) testing problem $\mathbb{P}$ versus $\mathbb{Q}$ (see Definition 3.8). Note that in particular, this also rules out strong separation. To motivate why weak separation is a natural notion of success for low-degree tests, we first show that weak separation by a polynomial $f$ implies that $f$'s output can be used to achieve weak detection (see Definition 1.2). Unlike the analogous result "strong separation implies strong detection" (which follows immediately from Chebyshev's inequality), the testing procedure here may be more complicated than simply thresholding $f$.

**Proposition 6.1.** *Suppose $\mathbb{P} = \mathbb{P}_n$ and $\mathbb{Q} = \mathbb{Q}_n$ are distributions over $\mathbb{R}^N$ for some $N = N_n$. If there exists a polynomial $f = f_n$ that weakly separates $\mathbb{P}$ and $\mathbb{Q}$ then weak detection is possible.*

*Proof.* It suffices to show that the random variable $P := f(Y)$ for $Y \sim \mathbb{P}$ has non-vanishing total variation (TV) distance from the random variable $Q := f(Y)$ for $Y \sim \mathbb{Q}$. By shifting and scaling we can assume $\mathbb{E}[Q] = 0$, $\mathbb{E}[P] = 1$, and that $\mathrm{Var}[Q]$ and $\mathrm{Var}[P]$ are both $O(1)$. This implies that $\mathbb{E}[Q^2]$ and $\mathbb{E}[P^2]$ are both $O(1)$. Assume on the contrary that the TV distance is vanishing, that is, $P$ and $Q$ can be coupled

so that $P = Q$ except on a "bad" event $B$ of probability $o(1)$. Using Cauchy–Schwarz and the inequality $(a - b)^2 \leq 2(a^2 + b^2)$,

$$1 = \mathbb{E}[P] - \mathbb{E}[Q] = \mathbb{E}[(P - Q)\mathbb{1}_B] \leq \sqrt{\mathbb{E}(P - Q)^2} \cdot \sqrt{\Pr(B)}$$
$$\leq \sqrt{2(\mathbb{E}[P^2] + \mathbb{E}[Q^2])} \cdot \sqrt{\Pr(B)}$$
$$= O(1) \cdot o(1) = o(1),$$

a contradiction. $\qquad\square$

The next result is the key to our approach: it shows that to rule out weak separation for the original (non-conditioned) testing problem, it suffices to bound LD for a *conditional* testing problem $\mathbb{P}|A$ versus $\mathbb{Q}$, where $A$ is any high-probability event under $\mathbb{P}$. More precisely, $A$ is allowed to depend both on the sample $Y \sim \mathbb{P}$ but also any latent randomness used to generate $Y$; notably, in our case, $A$ will depend on $u$.

**Proposition 6.2.** *Suppose $\mathbb{P} = \mathbb{P}_n$ and $\mathbb{Q} = \mathbb{Q}_n$ are distributions over $\mathbb{R}^N$ for some $N = N_n$. Let $A = A_n$ be a high-probability event under $\mathbb{P}$, that is, $\mathbb{P}(A) = 1 - o(1)$. Define the conditional distribution $\tilde{\mathbb{P}} = \mathbb{P}|A$. Suppose $\tilde{\mathbb{P}}$ is absolutely continuous with respect to $\mathbb{Q}$, let $L = \frac{d\tilde{\mathbb{P}}}{d\mathbb{Q}}$ and define $\mathrm{LD}(D) = \|L^{\leq D}\|_{\mathbb{Q}}^2$ accordingly. For any $D = D_n$,*

- *if $\mathrm{LD}(D) = O(1)$ as $n \to \infty$ then no degree-$D$ polynomial strongly separates $\mathbb{P}$ and $\mathbb{Q}$ (in the sense of Definition 1.8);*
- *if $\mathrm{LD}(D) = 1 + o(1)$ as $n \to \infty$ then no degree-$D$ polynomial weakly separates $\mathbb{P}$ and $\mathbb{Q}$ (in the sense of Definition 1.8).*

*Proof.* We will need the following variational formula (see e.g. [Hop18, Theorem 2.3.1]) for LD: letting $V^{\leq D}$ denote the space of polynomials $\mathbb{R}^N \to \mathbb{R}$ of degree at most $D$,

$$(31) \qquad \mathrm{LD}(D) - 1 = \|L^{\leq D}\|_{\mathbb{Q}}^2 - 1 = \|L^{\leq D} - 1\|_{\mathbb{Q}}^2 = \sup_{\substack{f \in V^{\leq D} \\ \mathbb{E}_{\mathbb{Q}}[f] = 0}} \frac{(\mathbb{E}_{\tilde{\mathbb{P}}}[f])^2}{\mathbb{E}_{\mathbb{Q}}[f^2]}.$$

We now begin the proof, which will be by contrapositive. Suppose a degree-$D$ polynomial $f = f_n$ strongly (respectively, weakly) separates $\mathbb{P}$ and $\mathbb{Q}$. By shifting and scaling we can assume $\mathbb{E}_{\mathbb{Q}}[f] = 0$ and $\mathbb{E}_{\mathbb{P}}[f] = 1$, and that $\mathrm{Var}_{\mathbb{Q}}[f]$ and $\mathrm{Var}_{\mathbb{P}}[f]$ are both $o(1)$ (resp., $O(1)$). Note that $\mathbb{E}_{\mathbb{Q}}[f^2] = \mathrm{Var}_{\mathbb{Q}}[f]$. It suffices to show $\mathbb{E}_{\tilde{\mathbb{P}}}[f] \geq 1 - o(1)$ so that, using (31),

$$\mathrm{LD}(D) - 1 \geq \frac{(\mathbb{E}_{\tilde{\mathbb{P}}}[f])^2}{\mathbb{E}_{\mathbb{Q}}[f^2]} \geq \frac{1 - o(1)}{\mathrm{Var}_{\mathbb{Q}}[f]}$$

which is $\omega(1)$ (resp., $\Omega(1)$), contradicting the assumption on $\mathrm{LD}(D)$ and completing the proof.

To prove $\mathbb{E}_{\tilde{\mathbb{P}}}[f] \geq 1 - o(1)$, we have

$$1 = \mathbb{E}_{\mathbb{P}}[f] = \mathbb{P}(A) \, \mathbb{E}_{\tilde{\mathbb{P}}}[f] + \mathbb{P}(A^c) \, \mathbb{E}_{\mathbb{P}}[f \mid A^c]$$

and so

$$\mathbb{E}_{\tilde{\mathbb{P}}}[f] = \mathbb{P}(A)^{-1}(1 - \mathbb{P}(A^c) \, \mathbb{E}_{\mathbb{P}}[f \mid A^c]).$$

Since $\mathbb{P}(A) = 1 - o(1)$, it suffices to show $\mathbb{P}(A^c)\mathbb{E}_{\mathbb{P}}[f \mid A^c] = o(1)$. As above,

$$\mathbb{E}_{\mathbb{P}}[f^2] = \mathbb{P}(A) \, \mathbb{E}_{\tilde{\mathbb{P}}}[f^2] + \mathbb{P}(A^c) \, \mathbb{E}_{\mathbb{P}}[f^2 \mid A^c]$$

and so

$$(32) \qquad \mathbb{E}_{\mathbb{P}}[f^2 \mid A^c] \leq \mathbb{P}(A^c)^{-1} \, \mathbb{E}_{\mathbb{P}}[f^2] = \mathbb{P}(A^c)^{-1}(\mathrm{Var}_{\mathbb{P}}[f] + 1).$$

Now using Jensen's inequality and (32),

$$
\begin{aligned}
\left| \mathbb{P}(A^c) \underset{\mathbb{P}}{\mathbb{E}}[f \mid A^c] \right| &\leq \mathbb{P}(A^c) \sqrt{\underset{\mathbb{P}}{\mathbb{E}}[f^2 \mid A^c]} \\
&\leq \mathbb{P}(A^c) \sqrt{\mathbb{P}(A^c)^{-1}(\mathrm{Var}_{\mathbb{P}}[f] + 1)} \\
&= \sqrt{\mathbb{P}(A^c)(\mathrm{Var}_{\mathbb{P}}[f] + 1)} \\
&= \sqrt{o(1) \cdot O(1)}
\end{aligned}
$$

which is $o(1)$ as desired. $\qquad\square$

6.1.2. *The "Good" Event.* We now specialize to the sparse regression problem (Definition 3.8). Under $\mathbb{P}$, the signal $u$ is drawn from $\mu$ where $\mu$ is the uniform prior over $k$-sparse binary vectors, and then the observation $(X, Y)$ is drawn from the appropriate distribution $\mathbb{P}_u$ as described in Definition 3.8. As described in the previous section, we will need to condition $\mathbb{P}$ on a particular "good" event $A$, which we define in this section. This event $A = A(u)$ will depend on the signal vector $u$, and (by symmetry) the probability $\mathbb{P}_u(A)$ will not depend on $u$, so our conditional distribution will take the form $\tilde{\mathbb{P}} = \mathbb{E}_{u \sim \mu} \tilde{\mathbb{P}}_u$ where $\tilde{\mathbb{P}}_u := \mathbb{P}_u | A$. Importantly, to fit the framework of Section 3 (specifically Example 3.5), the event $A$ will depend only on the columns of $X$ indexed by the support of the signal vector $u$.

**Definition 6.3** (Good Event). For a ground truth signal $u \in \{0,1\}^n$ with $\|u\|_0 = k$ and a sequence $\Delta = \Delta(\ell) > 0$ to be chosen later (see Lemma 6.4), let $A = A(u, \Delta)$ be the following event: for all integers $\ell$ in the range $1 \leq \ell \leq k/2$ and all subsets $S \subseteq \mathrm{supp}(u)$ of size $|S| = \ell$,

$$
\tag{33} \left\langle \frac{1}{\sqrt{\ell}} \sum_{j \in S} X_j , \; \frac{1}{\sqrt{k - \ell}} \sum_{j \in \mathrm{supp}(u) \setminus S} X_j \right\rangle \leq \Delta(\ell),
$$

where $(X_j)_{j \in [n]}$ denote the columns of $X$.

The following lemma gives us a choice of $\Delta$ such that the event $A = A(u, \Delta)$ occurs with high probability under $\mathbb{P}_u$.

**Lemma 6.4.** *Define $t = t(\ell) := \log[2^\ell \binom{k}{\ell} \log k]$ and $\Delta = \Delta(\ell) := \sqrt{2mt} + 10t$. Then under our scaling assumptions (Assumption 3.9), the following hold.*

- *For any fixed $\delta > 0$, for all sufficiently large $n$, for all integers $\ell$ with $1 \leq \ell \leq k/2$,*

$$
\tag{34} \Delta(\ell) \leq (1 + \delta)\sqrt{2\ell m \log k}.
$$

- *Under $\mathbb{P}_u$,*

$$
\Pr(A) \geq 1 - \frac{1}{\log k} = 1 - o(1).
$$

The proof uses standard concentration tools and is deferred to Appendix A.4.

6.1.3. *Proof Overview.* Throughout, we work with the *conditional* likelihood ratio $L = \frac{\mathrm{d}\tilde{\mathbb{P}}}{\mathrm{d}\mathbb{Q}} = \mathbb{E}_{u \sim \mu} L_u$ where $L_u = \frac{\mathrm{d}\tilde{\mathbb{P}}_u}{\mathrm{d}\mathbb{Q}}$ and where $\tilde{\mathbb{P}}, \tilde{\mathbb{P}}_u$ are as defined in Section 6.1.2. We define LD and FP accordingly, for the $\tilde{\mathbb{P}}$ versus $\mathbb{Q}$ problem. By Proposition 6.2, our goal is to bound LD. We will do this by exploiting the FP-to-LD connection for sparse planted models from Section 3. To apply Theorem 3.7, the first ingredient we need is a crude upper bound on $\|L_u^{\leq D}\|_{\mathbb{Q}}^2$.

**Lemma 6.5.** *For sufficiently large $n$, for any $u \in \{0,1\}^n$ with $\|u\|_0 = k$, for any integer $D \geq 1$,*

$$
\|L_u^{\leq D}\|_{\mathbb{Q}}^2 \leq 9(6mnD)^{4D}.
$$

The proof is deferred to Section 6.1.4. Now, recall from Example 3.5 that our conditioned sparse regression problem satisfies Assumption 3.1, and so we can apply Theorem 3.7 to conclude

$$
\tag{35} \mathrm{LD}(D) \leq \mathrm{FP}(D + \log M) + e^{-D} \qquad \text{where} \qquad M = 9(6mnD)^{4D}.
$$

It therefore remains to bound $\mathrm{FP}(\tilde{D})$ where $\tilde{D} = D + \log M$. Recall from (3) that $\mathrm{FP}(\tilde{D})$ is defined as $\mathrm{LO}(\delta)$ for a certain choice of $\delta$. To choose the right $\delta$, we need a tail bound on the overlap $\langle u, v \rangle$ where $u, v \sim \mu$ independently. In our case $\langle u, v \rangle$ follows the *hypergeometric distribution* $\mathrm{Hypergeom}(n, k, k)$, which has the following basic tail bounds.

**Lemma 6.6** (Hypergeometric tail bound). *For any integers $1 \leq \ell \leq k \leq n$,*

$$\Pr\{\mathrm{Hypergeom}(n, k, k) = \ell\} \leq \left( \frac{k^2}{n - k} \right)^\ell.$$

*Furthermore, if $k^2/(n - k) \leq 1$ then*

$$\Pr\{\mathrm{Hypergeom}(n, k, k) \geq \ell\} \leq k \left( \frac{k^2}{n - k} \right)^\ell.$$

*Proof.* For the first statement,

$$\Pr\{\mathrm{Hypergeom}(n, k, k) = \ell\} = \frac{\binom{k}{\ell}\binom{n-k}{k-\ell}}{\binom{n}{k}} \leq k^\ell \frac{\binom{n}{k-\ell}}{\binom{n}{k}} = k^\ell \frac{k!(n-k)!}{(k-\ell)!(n-k+\ell)!} \leq k^\ell \frac{k^\ell}{(n-k)^\ell}.$$

The second statement follows from the first by a union bound, noting that $k$ is the largest possible value for $\mathrm{Hypergeom}(n, k, k)$, and the bound $(\frac{k^2}{n-k})^\ell$ is decreasing in $\ell$. $\square$

It will end up sufficing to consider $\delta = \varepsilon k$ for a small constant $\varepsilon > 0$. The last ingredient we will need is the following bound on LO. This is the main technical heart of the argument, and the proof is deferred to Section 6.1.5.

**Proposition 6.7.** *Consider the setting of Theorem 3.10. If $R < R_{\mathrm{LD}}(\theta)$ then there exists a constant $\varepsilon = \varepsilon(\theta, R) > 0$ such that*

$$\mathrm{LO}(\varepsilon k) = 1 + o(1).$$

We now show how to combine the above ingredients to complete the proof.

*Proof of Theorem 3.10(a).* Suppose $0 < R < R_{\mathrm{LD}}(\theta)$ and $D = o(k)$. Also assume $D = \omega(1)$ without loss of generality, so that $e^{-D} = o(1)$. Recapping the arguments from this section, it suffices (by Proposition 6.2) to show $\mathrm{LD}(D) = 1 + o(1)$, where $\mathrm{LD}(D)$ denotes the *conditional* LD defined above. Let $\delta = \varepsilon k$ with $\varepsilon = \varepsilon(\theta, R)$ as defined in Proposition 6.7. From (35) and Proposition 6.7, it now suffices to show $\mathrm{FP}(D + \log M) \leq \mathrm{LO}(\delta)$. Recalling the definitions of LO and FP from (2) and (3), this holds provided

$$(36) \qquad \Pr(\langle u, v \rangle \geq \delta) < e^{-(D + \log M)}$$

where $u, v$ are uniformly random $k$-sparse binary vectors, i.e., $\langle u, v \rangle \sim \mathrm{Hypergeom}(n, k, k)$. (Note that $\langle u, v \rangle \geq 0$, so $|\langle u, v \rangle|$ can be replaced with $\langle u, v \rangle$ in this case.)

To complete the proof, we will establish that (36) holds for all sufficiently large $n$. We start by bounding the left-hand side. Since $R_{\mathrm{LD}}(\theta) = 0$ when $\theta \geq 1/2$ (see (24)), the assumption $0 < R < R_{\mathrm{LD}}(\theta)$ implies $\theta < 1/2$. Recalling $k = n^{\theta + o(1)}$ for $\theta \in (0, 1)$, this implies $k^2/(n - k) = n^{2\theta - 1 + o(1)} \leq 1$ (for sufficiently large $n$) and so by Lemma 6.6 we have

$$\Pr(\langle u, v \rangle \geq \delta) \leq k \left( n^{2\theta - 1 + o(1)} \right)^\delta$$

and so

$$(37) \qquad \log \Pr(\langle u, v \rangle \geq \delta) \leq \log k + (2\theta - 1 + o(1))\varepsilon k \log n = -\Omega(k \log n).$$

Now, for the right-hand side of (36),

$$(38) \qquad \log e^{-(D + \log M)} = -D - \log M = -D - \log 9 - 4D \log(4mnD) = -o(k \log n)$$

since $D = o(k)$ and $4mnD = n^{O(1)}$. Comparing (37) and (38) establishes (36) for sufficiently large $n$. $\square$

6.1.4. *Proof of Lemma 6.5.*

*Proof of Lemma 6.5.* Since $\mathbb{Q}$ is the standard Gaussian measure in $N = m(n+1)$ dimensions, the space $L^2(\mathbb{Q})$ admits the orthonormal basis of *Hermite polynomials* (see e.g. [Sze39] for a standard reference). We denote these by $(H_\alpha)_{\alpha \in \mathbb{N}^N}$ (where $0 \in \mathbb{N}$ by convention), where $H_\alpha(Z) = \prod_{i=1}^N h_{\alpha_i}(Z_i)$ for univariate Hermite polynomials $(h_j)_{j \in \mathbb{N}}$, and $Z = (X, Y) \in \mathbb{R}^N$. We adopt the normalization where $\|H_\alpha\|_{\mathbb{Q}} = 1$, which is not usually the standard convention in the literature. This basis is graded in the sense that for any $D \in \mathbb{N}$, $(H_\alpha)_{\alpha \in \mathbb{N}^N, |\alpha| \le D}$ is an orthonormal basis for the polynomials of degree at most $D$, where $|\alpha| := \sum_{i=1}^N \alpha_i$. Expanding in the orthonormal basis $\{H_\alpha\}$, we have for any $u$,

$$\|L_{\tilde{u}}^{\le D}\|_{\mathbb{Q}}^2 = \sum_{|\alpha| \le D} \langle L_u, H_\alpha \rangle_{\mathbb{Q}}^2 = \sum_{|\alpha| \le D} \left( \mathbb{E}_{Z \sim \mathbb{P}_u | A} H_\alpha(Z) \right)^2 \le (N+1)^D \max_{\alpha : |\alpha| \le D} \left( \mathbb{E}_{Z \sim \mathbb{P}_u | A} H_\alpha(Z) \right)^2.$$

Let $\mathbb{P}(A)$ denote the probability of $A$ under $\mathbb{P}_u$ (which, by symmetry, does not depend on $u$). Now we have

$$\left| \mathbb{E}_{Z \sim \mathbb{P}_u | A} H_\alpha(Z) \right| \le \mathbb{E}_{Z \sim \mathbb{P}_u | A} |H_\alpha(Z)|$$
$$\le \mathbb{P}(A)^{-1} \mathbb{E}_{Z \sim \mathbb{P}_u} |H_\alpha(Z)| \qquad \text{by Lemma A.6}$$
$$=: \mathbb{P}(A)^{-1} \|H_\alpha(Z)\|_1$$

where $L^p$ norms are with respect to $Z \sim \mathbb{P}_u$

$$= \mathbb{P}(A)^{-1} \left\| \prod_{i=1}^N h_{\alpha_i}(Z_i) \right\|_1$$
$$\le \mathbb{P}(A)^{-1} \prod_{i : \alpha_i > 0} \|h_{\alpha_i}(Z_i)\|_{d/\alpha_i},$$

where $d := |\alpha| \le D$ and where the last step used Proposition A.4, an extension of Hölder's inequality. Under $\mathbb{P}_u$, the marginal distribution of $Z_i$ is $\mathcal{N}(0, 1)$, so

$$\|h_{\alpha_i}(Z_i)\|_{d/\alpha_i} = \left( \mathbb{E}|h_{\alpha_i}(z)|^{d/\alpha_i} \right)^{\alpha_i/d}$$

where $z \sim \mathcal{N}(0, 1)$.

Now we have for $a \in \mathbb{N}$, $h_a(z) = \frac{1}{\sqrt{a!}} \sum_{j=0}^a c_j z^j$ where the coefficients satisfy $\sum_{j=0}^a |c_j| = T(a)$. Here $T(a)$ is known as the *telephone number* which counts the number of permutations on $a$ elements which are involutions [BBMD$^+$02]. In particular, we have the trivial upper bound $T(a) \le a!$. This means for any $a \ge 1$ and $q \in [1, \infty)$,

$$\mathbb{E}|h_a(z)|^q = \mathbb{E} \left| \frac{1}{\sqrt{a!}} \sum_{j=0}^a c_j z^j \right|^q$$
$$\le \mathbb{E} \left( \sqrt{a!} \max_{0 \le j \le a} |z|^j \right)^q$$
$$= (a!)^{q/2} \mathbb{E} \left( \max\{1, |z|^a\} \right)^q$$
$$= (a!)^{q/2} \mathbb{E} \max\{1, |z|^{aq}\}$$
$$\le (a!)^{q/2} (1 + \mathbb{E}|z|^{aq}).$$

Using the formula for Gaussian moments, and that for all $x \geq 1$, $\Gamma(x) \leq x^x$ (see e.g. [LC07]) the above becomes

$$= (a!)^{q/2} \left( 1 + \pi^{-1/2} 2^{aq/2} \Gamma \left( \frac{aq+1}{2} \right) \right)$$

$$\leq a^{aq/2} \left( 1 + 2^{aq/2} \left( \frac{aq+1}{2} \right)^{(aq+1)/2} \right)$$

$$\leq a^{aq/2} (1 + 2^{aq/2} (aq)^{aq}) \qquad \text{since } \frac{aq+1}{2} \leq aq$$

$$\leq 2 a^{aq/2} 2^{aq/2} (aq)^{aq}$$

$$= 2 (2a)^{aq/2} (aq)^{aq}$$

$$\leq 2 (2aq)^{2aq}.$$

Putting it together, and recalling $d = \sum_i \alpha_i$,

$$\left| \mathbb{E}_{Z \sim \mathbb{P}_u | A} H_\alpha(Z) \right| \leq \mathbb{P}(A)^{-1} \prod_{i \,:\, a_i > 0} \left( 2(2d)^{2d} \right)^{\alpha_i/d} = 2\mathbb{P}(A)^{-1} (2d)^{2d} \leq 3(2D)^{2D},$$

since $\mathbb{P}(A) \geq 2/3$ for sufficiently large $n$, and $d \leq D$. Finally, using the bound

$$N + 1 = m(n+1) + 1 \leq 3mn,$$

we have

$$\|L_u^{\leq D}\|_{\mathbb{Q}}^2 \leq (N+1)^D \left[ 3(2D)^{2D} \right]^2 \leq 9(3mn)^D (2D)^{4D} \leq 9(6mnD)^{4D},$$

completing the proof. $\qquad\square$

6.1.5. *Proof of Proposition 6.7.* A key step in bounding LO is to establish the following bound on $\langle L_u, L_v \rangle_{\mathbb{Q}}$.

**Proposition 6.8.** *Let $u, v \in \{0,1\}^n$ with $\|u\|_0 = \|v\|_0 = k$ and $\langle u, v \rangle = \ell$. Let $A = A(u, \Delta)$ be the "good" event from Definition 6.3, for some sequence $\Delta$. Also suppose $\sigma^2 \leq \varepsilon k$ and $\ell \leq \varepsilon k$ for some $\varepsilon \in (0,1)$. It holds that*

$$(39) \qquad \langle L_u, L_v \rangle_{\mathbb{Q}} \leq \mathbb{P}(A)^{-2} \exp \left( \left( \frac{\ell}{k-\ell} \right) m \right).$$

*Furthermore, if there exists some $q = q(\ell) > 0$ satisfying*

$$(40) \qquad \Delta(\ell) \leq \left( (1-\varepsilon)^2 \sqrt{\frac{\ell}{k}} - \frac{\sqrt{2(1+3\varepsilon)}}{1-\varepsilon} q - \frac{10}{(1-\varepsilon)^{3/2}} q^2 \right) m,$$

*then for this $q = q(\ell)$ it holds that*

$$(41) \qquad \langle L_u, L_v \rangle \leq \mathbb{P}(A)^{-2} \exp \left( \left( \frac{\ell}{k-\ell} - q^2 \right) m \right).$$

The proof of Proposition 6.8 is deferred to Section 6.1.6. We now show how to use this result to bound LO.

*Proof of Proposition 6.7.* We start with the case $\frac{1}{4} \leq \theta < \frac{1}{2}$ and $R < \frac{1-2\theta}{1-\theta}$. Fix a constant $\varepsilon = \varepsilon(\theta, R) > 0$ to be chosen later. Let $u, v \in \{0,1\}^n$ be independent uniformly random binary vectors of sparsity exactly $k$, and note that $\langle u, v \rangle$ follows the hypergeometric distribution $\mathrm{Hypergeom}(n, k, k)$. Therefore Proposition 6.8 and Lemma 6.4 imply

$$\mathrm{LO}(\varepsilon k) := \mathbb{E}_{u,v} \left[ \mathbb{1}_{\langle u,v \rangle \leq \varepsilon k} \cdot \langle L_u, L_v \rangle_{\mathbb{Q}} \right]$$

$$\leq \mathbb{P}(A)^{-2} \mathbb{E}_{\ell \sim \mathrm{Hypergeom}(n,k,k)} \left[ \mathbb{1}_{\ell \leq \varepsilon k} \exp \left( \frac{\ell m}{k-\ell} \right) \right]$$

$$(42) \qquad = (1 + o(1)) \sum_{0 \leq \ell \leq \varepsilon k} \Pr\{\mathrm{Hypergeom}(n, k, k) = \ell\} \exp \left( \frac{\ell m}{k-\ell} \right).$$

We bound the two terms in the product separately.

**First term.** Using the hypergeometric tail bound (Lemma 6.6), since $k = n^{\theta+o(1)}$ we have

$$(43) \qquad \Pr\{\text{Hypergeom}(n, k, k) = \ell\} \leq \left(\frac{k^2}{n-k}\right)^\ell = \left(n^{2\theta-1+o(1)}\right)^\ell.$$

**Second term.** Recalling $k = n^{\theta+o(1)}$ and $m = (1+o(1))(1-\theta)Rk \log n$ we have for every $0 \leq \ell \leq \varepsilon k$,

$$\exp\left(\frac{\ell m}{k-\ell}\right) \leq \exp\left(\frac{\ell m}{(1-\varepsilon)k}\right)$$
$$= \exp\left(\ell \cdot (1+o(1))(1-\varepsilon)^{-1}R(1-\theta)\log n\right)$$
$$(44) \qquad\qquad = \left(n^{(1-\varepsilon)^{-1}R(1-\theta)+o(1)}\right)^\ell.$$

Plugging (43) and (44) back into (42) we have

$$\text{LO}(\varepsilon k) \leq (1+o(1)) \left[1 + \sum_{1 \leq \ell \leq \varepsilon k} \left(n^{2\theta-1+o(1)}\right)^\ell \left(n^{(1-\varepsilon)^{-1}R(1-\theta)+o(1)}\right)^\ell\right]$$

$$\leq (1+o(1)) \left[1 + \sum_{1 \leq \ell \leq \varepsilon k} \left(n^{2\theta-1+(1-\varepsilon)^{-1}R(1-\theta)+o(1)}\right)^\ell\right].$$

Provided $2\theta - 1 + R(1-\theta) < 0$, i.e., $R < \frac{1-2\theta}{1-\theta}$, it is possible to choose $\varepsilon = \varepsilon(R, \theta) > 0$ small enough so that $2\theta - 1 + (1-\varepsilon)^{-1}R(1-\theta) < 0$ and therefore $\text{LO}(\varepsilon k) \leq 1 + o(1)$ as we wanted.

Now we focus on the second case where $0 < \theta < \frac{1}{4}$ and $R < \frac{2(1-\sqrt{\theta})}{1+\sqrt{\theta}}$. We are also free to assume

$$(45) \qquad\qquad R \geq \frac{1-2\theta}{1-\theta}$$

or else we can immediately conclude the result using the same argument as above. Similar to the first case, using now the second part of Proposition 6.8, for any sequence $q = q(\ell)$ satisfying (40) we have

$$\text{LO}(\varepsilon k) \leq \mathbb{P}(A)^{-2} \mathop{\mathbb{E}}_{\ell \sim \text{Hypergeom}(n,k,k)} \left[\mathbb{1}_{\ell \leq \varepsilon k} \exp\left(\left(\frac{\ell}{k-\ell} - q^2\right)m\right)\right].$$

Hence, by Lemma 6.4 and similar reasoning to the previous case it also holds for any sequence $q = q(\ell)$ satisfying (40) that

$$\text{LO}(\varepsilon k) \leq (1+o(1)) \sum_{0 \leq \ell \leq \varepsilon k} \Pr\{\text{Hypergeom}(n, k, k) = \ell\} \exp\left(\frac{\ell m}{k-\ell}\right) \exp\left(-q^2 m\right)$$

$$(46) \qquad \leq (1+o(1)) \left[1 + \sum_{1 \leq \ell \leq \varepsilon k} \left(n^{2\theta-1+(1-\varepsilon)^{-1}R(1-\theta)+o(1)}\right)^\ell \exp\left(-q^2 m\right)\right].$$

Now we choose $q = q(\ell) = c\sqrt{\ell(\log n)/m}$ for a constant $c = c(\theta, R) > 0$ to be chosen later. To satisfy (40), it suffices (using (34) from Lemma 6.4) to have for some constant $\delta = \delta(\theta, R) > 0$,

$$(1+\delta)\sqrt{2\ell m \log k} \leq (1-\varepsilon)^2 \sqrt{\frac{\ell}{k}} m - \frac{\sqrt{1+3\varepsilon}}{1-\varepsilon}\sqrt{2}qm - 10(1-\varepsilon)^{-3/2}q^2 m,$$

or since $\ell \leq \varepsilon k$ and therefore $q \leq c\sqrt{\sqrt{\varepsilon \ell k}(\log n)/m}$, it suffices to have

$$(1+\delta)\sqrt{2\ell m \log k} \leq (1-\varepsilon)^2 \sqrt{\frac{\ell}{k}} m - \frac{\sqrt{1+3\varepsilon}}{1-\varepsilon}\sqrt{2}qm - 10(1-\varepsilon)^{-3/2}c^2 \sqrt{\varepsilon \ell k} \log n.$$

Using the asymptotics of $k, m, q$ and dividing both sides by $\sqrt{2\ell R(1-\theta)k} \log n$, it suffices to have

$$(1+\delta)^2(1+o(1))\sqrt{\theta} \leq (1-\varepsilon)^2(1+o(1))\sqrt{\frac{1}{2}R(1-\theta)} - \frac{\sqrt{1+3\varepsilon}}{1-\varepsilon}(1+o(1))c - \frac{10\sqrt{\varepsilon}(1-\varepsilon)^{-3/2}c^2}{\sqrt{2R(1-\theta)}}.$$

But now there exist sufficiently small constants $\varepsilon = \varepsilon(\theta, R) > 0$ and $\delta = \delta(\theta, R) > 0$ satisfying the above so long as

$$(47) \qquad c < \sqrt{\frac{1}{2} R(1 - \theta)} - \sqrt{\theta}.$$

Notice that such a $c > 0$ exists since $\sqrt{\theta} < \sqrt{\frac{1}{2} R(1 - \theta)}$, which holds by our assumptions that $\theta < \frac{1}{4}$ and therefore $R \geq \frac{1 - 2\theta}{1 - \theta} > \frac{2\theta}{1 - \theta}$ (see (45)).

Returning to (46), we have

$$\mathrm{LO}(\varepsilon k) \leq (1 + o(1)) \left[ 1 + \sum_{1 \leq \ell \leq \varepsilon k} \left( n^{2\theta - 1 + (1 - \varepsilon)^{-1} R(1 - \theta) - c^2 + o(1)} \right)^{\ell} \right],$$

which concludes the result $\mathrm{LO}(\varepsilon k) = 1 + o(1)$ for sufficiently small $\varepsilon > 0$, provided that

$$(48) \qquad 2\theta - 1 + R(1 - \theta) - c^2 < 0.$$

We can choose $c > 0$ to satisfy both (47) and (48) simultaneously, provided

$$2\theta - 1 + R(1 - \theta) - \left( \sqrt{\frac{1}{2} R(1 - \theta)} - \sqrt{\theta} \right)^2 < 0,$$

which can be simplified (by expanding the square) to

$$(1 - R/2)(1 - \theta) > \sqrt{2R\theta(1 - \theta)}.$$

Squaring both sides yields the equivalent condition

$$\frac{(1 - R/2)^2}{2R} > \frac{\theta}{1 - \theta} \qquad \text{and} \qquad R < 2.$$

Solving for $R$ via the quadratic formula yields the equivalent condition $R < \frac{2(1 - \sqrt{\theta})}{1 + \sqrt{\theta}}$ as desired. $\qquad \square$

6.1.6. *Proof of Proposition 6.8.*

*Proof of Proposition 6.8.* Throughout we denote for simplicity, $\lambda = \sqrt{k/\sigma^2 + 1}$. By definition and Bayes' rule,

$$L_u(X, Y) = \frac{\mathbb{P}(X, Y | u, A)}{\mathbb{Q}(X, Y)} = \frac{\mathbb{P}(Y | X, u, A)}{\mathbb{Q}(Y)} \cdot \frac{\mathbb{P}(X | u, A)}{\mathbb{Q}(X)} = \frac{\mathbb{P}(Y | X, u)}{\mathbb{Q}(Y)} \cdot \frac{\mathbb{1}\{(X, u) \in A\}}{\mathbb{P}(A)},$$

where we have used the fact $\mathbb{P}(Y | X, u, A) = \mathbb{P}(Y | X, u)$ since $Y$ depends on $A$ only through $(X, u)$, and the fact $\mathbb{Q}(X) = \mathbb{P}(X)$. Under $\mathbb{Q}$ we have $\lambda \sigma Y \sim \mathcal{N}(0, \lambda^2 \sigma^2 I_m)$, while under $\mathbb{P}$ conditional on $(X, u)$ we have $\lambda \sigma Y = \sqrt{k + \sigma^2} Y \sim \mathcal{N}(Xu, \sigma^2 I_m)$, and so

$$\frac{\mathbb{P}(Y | X, u)}{\mathbb{Q}(Y)} = \lambda^m \exp \left( -\frac{1}{2\sigma^2} \| \lambda \sigma Y - Xu \|_2^2 + \frac{1}{2\lambda^2 \sigma^2} \| \lambda \sigma Y \|_2^2 \right)$$

$$= \lambda^m \exp \left( -\frac{\lambda^2 - 1}{2} \| Y \|_2^2 + \frac{\lambda}{\sigma} \langle Y, Xu \rangle - \frac{1}{2\sigma^2} \| Xu \|_2^2 \right).$$

This means

$$\langle L_u, L_v \rangle_{\mathbb{Q}} = \mathop{\mathbb{E}}_{(X, Y) \sim \mathbb{Q}} [L_u(X, Y) L_v(X, Y)]$$

$$(49) \qquad = \mathbb{P}(A)^{-2} \mathop{\mathbb{E}}_{(X, Y) \sim \mathbb{Q}} \left[ \mathbb{1}\{(X, u), (X, v) \in A\} \cdot \frac{\mathbb{P}(Y | X, u)}{\mathbb{Q}(Y)} \cdot \frac{\mathbb{P}(Y | X, v)}{\mathbb{Q}(Y)} \right]$$

where

$$\frac{\mathbb{P}(Y|X,u)}{\mathbb{Q}(Y)} \cdot \frac{\mathbb{P}(Y|X,v)}{\mathbb{Q}(Y)}$$

$$= \lambda^{2m} \exp\left(-(\lambda^2 - 1)\|Y\|_2^2 + \frac{\lambda}{\sigma}\langle Y, X(u+v)\rangle - \frac{1}{2\sigma^2}\left(\|Xu\|_2^2 + \|Xv\|_2^2\right)\right)$$

$$= \lambda^{2m} \exp\left(-\frac{\lambda^2 - 1}{\sigma^2\lambda^2}\left\|\lambda\sigma Y - \frac{\lambda^2 X(u+v)}{2(\lambda^2 - 1)}\right\|_2^2 + \frac{\lambda^2\|X(u+v)\|_2^2}{4(\lambda^2 - 1)\sigma^2} - \frac{1}{2\sigma^2}\left(\|Xu\|_2^2 + \|Xv\|_2^2\right)\right).$$

In the last step we have "completed the square" so that we can now explicitly compute the expectation over $Y \sim \mathbb{Q}$ using the (noncentral) chi-squared moment-generating function: for $t < 1/(2\nu^2)$ and $z \sim \mathcal{N}(\mu, \nu^2)$, $\mathbb{E}[\exp(tz^2)] = (1 - 2t\nu^2)^{-1/2}\exp[\mu^2 t/(1 - 2t\nu^2)]$. This yields

$$\mathop{\mathbb{E}}_{Y\sim\mathbb{Q}} \exp\left(-\frac{\lambda^2 - 1}{\sigma^2\lambda^2}\left\|\lambda\sigma Y - \frac{\lambda^2 X(u+v)}{2(\lambda^2 - 1)}\right\|_2^2\right) = \frac{1}{(2\lambda^2 - 1)^{m/2}}\exp\left(-\frac{\lambda^2\|X(u+v)\|_2^2}{4(2\lambda^2 - 1)(\lambda^2 - 1)\sigma^2}\right).$$

Plugging these results back into (49),

$$\langle L_u, L_v\rangle_{\mathbb{Q}} = \mathbb{P}(A)^{-2}\mathop{\mathbb{E}}_{X\sim\mathbb{Q}} \mathbb{1}\{(X,u),(X,v) \in A\}\frac{\lambda^{2m}}{(2\lambda^2 - 1)^{m/2}}$$

$$\cdot \exp\left(-\frac{\lambda^2\|X(u+v)\|_2^2}{4(2\lambda^2 - 1)(\lambda^2 - 1)\sigma^2} + \frac{\lambda^2\|X(u+v)\|_2^2}{4(\lambda^2 - 1)\sigma^2} - \frac{1}{2\sigma^2}\left(\|Xu\|_2^2 + \|Xv\|_2^2\right)\right)$$

$$= \mathbb{P}(A)^{-2}\mathop{\mathbb{E}}_{X\sim\mathbb{Q}} \mathbb{1}\{(X,u),(X,v) \in A\}\frac{\lambda^{2m}}{(2\lambda^2 - 1)^{m/2}}$$

(50)
$$\cdot \exp\left\{\frac{1}{2\sigma^2(2\lambda^2 - 1)}\left[(1 - \lambda^2)\left(\|Xu\|_2^2 + \|Xv\|_2^2\right) + 2\lambda^2\langle Xu, Xv\rangle\right]\right\}.$$

Let $T = \mathrm{supp}(u)$ and $T' = \mathrm{supp}(v)$. Let $X_i$ denote the $i$-th column of $X$. Define

$$Z_0 = \sum_{i\in T\cap T'} X_i, \quad Z_1 = \sum_{i\in T\backslash T'} X_i, \quad Z_2 = \sum_{i\in T'\backslash T} X_i.$$

Then under $\mathbb{Q}$ (with $u, v$ fixed), the values $Z_0, Z_1, Z_2$ are mutually independent and

$$Z_0 \sim \mathcal{N}(0, \ell I_m), \quad Z_1 \sim \mathcal{N}(0, (k - \ell)I_m), \quad Z_2 \sim \mathcal{N}(0, (k - \ell)I_m),$$

where $\ell = |T \cap T'| = \langle u, v\rangle$. Moreover, $Xu$ and $Xv$ can be expressed in terms of $Z_0, Z_1, Z_2$ simply by

(51)
$$Xu = Z_0 + Z_1 \quad \text{and} \quad Xv = Z_0 + Z_2.$$

Finally, notice that for any $X$ satisfying $(X,u),(X,v) \in A$ it necessarily holds (using the definition of $A$ in Section 6.1.2) that

$$\left\langle \frac{1}{\sqrt{\ell}}Z_0, \frac{1}{\sqrt{k-\ell}}Z_1\right\rangle \le \Delta, \quad \left\langle \frac{1}{\sqrt{\ell}}Z_0, \frac{1}{\sqrt{k-\ell}}Z_2\right\rangle \le \Delta,$$

where $\Delta = \Delta(\ell)$ is as in Section 6.1.2 (and in particular, satisfies the bound (34)).

We will next use the above to rewrite (50) in terms of $Z \in \mathbb{R}^{m\times 3}$ with i.i.d. $\mathcal{N}(0,1)$ entries, where the columns of $Z$ are $\frac{1}{\sqrt{\ell}}Z_0, \frac{1}{\sqrt{k-\ell}}Z_1, \frac{1}{\sqrt{k-\ell}}Z_2$. For symmetric $U \in \mathbb{R}^{3\times 3}$, define the event $B(U) = \{U_{12} \le \Delta \text{ and } U_{13} \le \Delta\}$. Also let $\ell := \langle u, v\rangle$ and

(52)
$$t := \frac{1}{2\sigma^2(2\lambda^2 - 1)} = \frac{1}{2\sigma^2(2k/\sigma^2 + 1)} = \frac{1}{4k + 2\sigma^2}.$$

This yields

$$\langle L_u, L_v\rangle_{\mathbb{Q}} \le \mathbb{P}(A)^{-2}\frac{\lambda^{2m}}{(2\lambda^2 - 1)^{m/2}}\mathop{\mathbb{E}}_Z \mathbb{1}_{B(Z^\top Z)}$$

$$\cdot \exp\left\{t\left[(1 - \lambda^2)\left(\|Z_0 + Z_1\|_2^2 + \|Z_0 + Z_2\|_2^2\right) + 2\lambda^2\langle Z_0 + Z_1, Z_0 + Z_2\rangle\right]\right\}$$

(53)
$$= \mathbb{P}(A)^{-2}\frac{\lambda^{2m}}{(2\lambda^2 - 1)^{m/2}}\mathop{\mathbb{E}}_Z \mathbb{1}_{B(Z^\top Z)}\exp\left(t\langle M, Z^\top Z\rangle\right)$$

where

$$M = M(\ell) := \begin{pmatrix} 2\ell & \sqrt{\ell(k-\ell)} & \sqrt{\ell(k-\ell)} \\ \sqrt{\ell(k-\ell)} & (1-\lambda^2)(k-\ell) & \lambda^2(k-\ell) \\ \sqrt{\ell(k-\ell)} & \lambda^2(k-\ell) & (1-\lambda^2)(k-\ell) \end{pmatrix}.$$

The eigendecomposition of $M$ is $\sum_{i=1}^3 \lambda_i \frac{u_i u_i^\top}{\|u_i\|^2}$ where

(54)
$$\begin{aligned} u_1^\top &= (0 \quad 1 \quad -1) & \lambda_1 &= (1-2\lambda^2)(k-\ell) \\ u_2^\top &= (\sqrt{k-\ell} \quad -\sqrt{\ell} \quad -\sqrt{\ell}) & \lambda_2 &= 0 \\ u_3^\top &= (2\sqrt{\ell} \quad \sqrt{k-\ell} \quad \sqrt{k-\ell}) & \lambda_3 &= k+\ell. \end{aligned}$$

We will evaluate the expression in (53) via some direct manipulations with the Wishart density function that are deferred to Appendix A.3. To apply Lemma A.5, we need to first verify $tM \prec \frac{1}{2}I_3$. This follows from the fact that the maximum eigenvalue of $M$ is $k+\ell \le 2k$ (see (54)) and the fact $t < \frac{1}{4k}$ (see (52)). Applying Lemma A.5 to (53) we conclude

(55)
$$\langle L_u, L_v \rangle_{\mathbb{Q}} \le \mathbb{P}(A)^{-2} \frac{\lambda^{2m}}{(2\lambda^2 - 1)^{m/2}} \det(I_3 - 2tM)^{-m/2} \Pr_{U \sim W_3((I_3-2tM)^{-1}, m)} \{B(U)\}$$

where the Wishart distribution $U \sim W_3((I_3 - 2tM)^{-1}, m)$ means $U = Z^\top Z$ where $Z \in \mathbb{R}^{m \times 3}$ has independent rows drawn from $\mathcal{N}(0, (I_3 - 2tM)^{-1})$.

We now focus on bounding $\det(I_3 - 2tM)^{-m/2}$. Using (52) and (54), the eigenvalues of the matrix $I_3 - 2tM$ are

$$\{1, 1 - 2t(k+\ell), 1 - 2t(1-2\lambda^2)(k-\ell)\} = \left\{1, 1 - \frac{k+\ell}{\sigma^2(2\lambda^2 - 1)}, 1 + \frac{k-\ell}{\sigma^2}\right\}.$$

Since $\lambda^2 = k/\sigma^2 + 1$ we conclude

$$\begin{aligned} \frac{\lambda^2}{\sqrt{2\lambda^2 - 1}} \det(I_3 - 2tM)^{-1/2} &= \lambda^2 \left[\left(2\lambda^2 - 1 - \frac{k+\ell}{\sigma^2}\right)\left(1 + \frac{k-\ell}{\sigma^2}\right)\right]^{-1/2} \\ &= \frac{\frac{k}{\sigma^2} + 1}{1 + \frac{k-\ell}{\sigma^2}} \\ &= \left(1 - \frac{\ell}{k + \sigma^2}\right)^{-1}. \end{aligned}$$

Hence, it holds that

$$\begin{aligned} \frac{\lambda^{2m}}{(2\lambda^2 - 1)^{m/2}} \det(I_3 - 2tM)^{-m/2} &= \left(1 - \frac{\ell}{k + \sigma^2}\right)^{-m} \\ &= \exp\left[-m \log\left(1 - \frac{\ell}{k + \sigma^2}\right)\right] \end{aligned}$$

(56)
$$\le \exp\left(\frac{\ell m}{k - \ell}\right)$$

where we have used the bound $\log(x) \ge 1 - 1/x$ in the last step. Combining (55) and (56),

(57)
$$\langle L_u, L_v \rangle_{\mathbb{Q}} \le \mathbb{P}(A)^{-2} \exp\left(\frac{\ell m}{k - \ell}\right) \Pr_{U \sim W_3((I_3-2tM)^{-1}, m)} \{B(U)\}.$$

At this point, we can conclude the first claim (39) by simply taking the trivial bound $\Pr\{B(U)\} \le 1$ on the last term above (this argument does not exploit the conditioning on the "good" event $A$). To prove the second claim (41), we will need a better bound on $\Pr\{B(U)\}$.

Using the eigendecomposition (54), we can directly compute the entries of

$$V := (I_3 - 2tM)^{-1} = \sum_{i=1}^3 (1 - 2t\lambda_i)^{-1} \frac{u_i u_i^\top}{\|u_i\|^2}$$

and in particular deduce

$$V_{11} = \frac{k + \sigma^2 + \ell}{k + \sigma^2 - \ell} \qquad V_{12} = V_{13} = \frac{\sqrt{\ell(k - \ell)}}{k + \sigma^2 - \ell} \qquad V_{22} + V_{23} = 2 \qquad V_{22} = V_{33}.$$

Since $U \sim W_3(V, m)$, we have $U_{12} + U_{13} = 2 \sum_{i=1}^m s_i$ where $s_i$ are i.i.d. and distributed as $s = \frac{1}{2} g_1 (g_2 + g_3)$ where $g \sim \mathcal{N}(0, V)$. Equivalently, we can write

$$g_1 = \sqrt{V_{11}} \, z_1 \qquad \text{and} \qquad g_2 + g_3 = \frac{2 V_{12}}{\sqrt{V_{11}}} z_1 + \sqrt{2 V_{22} + 2 V_{23} - 4 V_{12}^2 / V_{11}} \, z_2$$

where $z_1, z_2$ are independent $\mathcal{N}(0, 1)$, and so

(58) $$s = \frac{1}{2} g_1 (g_2 + g_3) = V_{12} z_1^2 + \sqrt{\frac{1}{2} V_{11} (V_{22} + V_{23}) - V_{12}^2} \, z_1 z_2 = V_{12} z_1^2 + \sqrt{V_{11} - V_{12}^2} \, z_1 z_2.$$

Hence, recalling $B(U) = \{U_{12} \leq \Delta \text{ and } U_{13} \leq \Delta\}$,

$$\Pr_{U \sim W_3(V, m)} \{B(U)\} \leq \Pr_{U \sim W_3(V, m)} \{U_{12} + U_{13} \leq 2\Delta\} = \Pr \left\{ \sum_{i=1}^m s_i \leq \Delta \right\}.$$

Now using Corollary A.2 for $y = q^2 m$, along with the representation (58) for $s$, we conclude that for all $q > 0$,

$$\Pr \left\{ \sum_{i=1}^m s_i \leq \left( a - \sqrt{2(3a^2 + b^2)} \, q - 10\sqrt{a^2 + b^2} \, q^2 \right) m \right\} \leq \exp(-q^2 m)$$

where $a = V_{12}$ and $b^2 = V_{11} - V_{12}^2$.

We now claim that for $q > 0$ satisfying (40) it must also hold that

(59) $$\Delta \leq \left( a - \sqrt{2(3a^2 + b^2)} \, q - 10\sqrt{a^2 + b^2} \, q^2 \right) m.$$

Notice that upon establishing (59) we can conclude

$$\Pr_{U \sim W_3(V, m)} \{B(U)\} \leq \Pr \left\{ \sum_{i=1}^m s_i \leq \Delta \right\} \leq \exp(-q^2 m),$$

and therefore combining with (57),

$$\langle L_u, L_v \rangle_{\mathbb{Q}} \leq \mathbb{P}(A)^{-2} \exp \left( \frac{\ell m}{k - \ell} - q^2 m \right)$$

which concludes the proof of (41).

Now we focus on establishing (59) as the final step of the proof. Now since $0 \leq \ell \leq \varepsilon k$ and $\sigma^2 \leq \varepsilon k$, we have

(60) $$a = V_{12} = \frac{\sqrt{\ell(k - \ell)}}{k + \sigma^2 - \ell} \geq \frac{\sqrt{\ell(1 - \varepsilon)k}}{(1 + \varepsilon)k} \geq \frac{1 - \varepsilon}{1 + \varepsilon} \sqrt{\frac{\ell}{k}} \geq (1 - \varepsilon)^2 \sqrt{\frac{\ell}{k}}.$$

Also, elementary algebra gives

$$a = \frac{\sqrt{\ell(k - \ell)}}{k + \sigma^2 - \ell} \leq \frac{\sqrt{\ell k}}{(1 - \varepsilon)k} = \frac{1}{1 - \varepsilon} \sqrt{\frac{\ell}{k}} \leq \frac{\sqrt{\varepsilon}}{1 - \varepsilon},$$

$$b^2 = V_{11} - V_{12}^2 \leq V_{11} = \frac{k + \sigma^2 + \ell}{k + \sigma^2 - \ell} \leq \frac{k + \ell}{k - \ell} \leq \frac{1 + \varepsilon}{1 - \varepsilon} \leq (1 - \varepsilon)^{-2},$$

and therefore

(61) $$a^2 + b^2 \leq \frac{1 + \varepsilon}{(1 - \varepsilon)^2} \leq (1 - \varepsilon)^{-3},$$

(62) $$3a^2 + b^2 \leq \frac{1 + 3\varepsilon}{(1 - \varepsilon)^2}.$$

Therefore since $q > 0$, combining (60), (61), (62) we have

$$
(63) \qquad a - \sqrt{2(3a^2 + b^2)}\, q - 10\sqrt{a^2 + b^2}\, q^2 \geq (1 - \varepsilon)^2 \sqrt{\frac{\ell}{k}} - \frac{\sqrt{2(1 + 3\varepsilon)}}{1 - \varepsilon}\, q - \frac{10}{(1 - \varepsilon)^{3/2}}\, q^2.
$$

But now combining (40) and (63) we conclude (59) and the proof is complete. $\qquad\square$

6.2. **Proof of Theorem 3.10(b): Upper Bound.** Given $(X, Y)$ drawn from either $\mathbb{Q}$ or $\mathbb{P}$, we will distinguish via the statistic $T$ that counts the number of indices $j \in [n]$ such that $\langle X_j, Y \rangle / \|Y\|_2 \geq \tau$ where $X_j$ denotes column $j$ of $X$ and $\tau > 0$ is a threshold to be chosen later. Specifically, we set

$$
T_\tau = \left| \left\{ j \in [n] : \frac{\langle X_j, Y \rangle}{\|Y\|_2} \geq \tau \right\} \right|.
$$

We will choose $\tau = c\sqrt{\log n}$ for an appropriate constant $c = c(\theta, R) > 0$ to be chosen later.

6.2.1. *Null Model.* Define

$$
(64) \qquad q = q(\tau) := \Pr\{\mathcal{N}(0, 1) \geq \tau\}.
$$

**Proposition 6.9.** *Let* $\tau = c\sqrt{\log n}$ *for a constant* $c > 0$. *Then under the null model* $\mathbb{Q}$ *we have* $\mathbb{E}[T_\tau] = qn = n^{1 - c^2/2 + o(1)}$ *and* $\mathrm{Var}(T_\tau) \leq qn = n^{1 - c^2/2 + o(1)}$.

We need the following Lemma.

**Lemma 6.10.** *Let* $\tau = c\sqrt{\log n}$ *for a constant* $c > 0$ *and define* $q = q(\tau)$ *as in* (64). *Then* $q = n^{-c^2/2 + o(1)}$.

*Proof.* Using a standard Gaussian tail bound,

$$
q = \Pr\{\mathcal{N}(0, 1) \geq \tau\} \leq \exp\left(-\frac{\tau^2}{2}\right) = \exp\left(-\frac{c^2}{2}\log n\right) = n^{-c^2/2}.
$$

For the reverse bound, use a standard *lower* bound on the Gaussian tail:

$$
q = \Pr\{\mathcal{N}(0, 1) \geq \tau\} \geq \frac{1}{\sqrt{2\pi}} \cdot \frac{\tau}{\tau^2 + 1} \exp\left(-\frac{\tau^2}{2}\right) = \frac{1}{\sqrt{2\pi}} \cdot \frac{\tau}{\tau^2 + 1}\, n^{-c^2/2} = n^{-c^2/2 + o(1)},
$$

completing the proof. $\qquad\square$

*Proof of Proposition 6.9.* Suppose $(X, Y) \sim \mathbb{Q}$. In this case, the values $z_j := \langle X_j, Y \rangle / \|Y\|_2$ are independent and each distributed as $\mathcal{N}(0, 1)$. The test statistic can be rewritten as

$$
T_\tau = \sum_{j=1}^{n} \mathbb{1}\{z_j \geq \tau\}.
$$

Therefore $\mathbb{E}[T] = qn$ and $\mathrm{Var}(T) = q(1 - q)n \leq qn$ where $q := \Pr\{\mathcal{N}(0, 1) \geq \tau\}$. The result then follows from Lemma 6.10. $\qquad\square$

6.2.2. *Planted Model.* We decompose the test statistic into two parts $T_\tau = T_\tau^+ + T_\tau^-$, depending on whether the index $j \in [n]$ lies in $S = \mathrm{supp}(u)$, the support of the signal vector $u$, or not. That is,

$$
T_\tau^+ := \sum_{j \in S} \mathbb{1}\{z_j \geq \tau\},
$$

and

$$
T_\tau^- := \sum_{j \notin S} \mathbb{1}\{z_j \geq \tau\}.
$$

The following analysis of $T_\tau^-$ follows by the same argument as Proposition 6.9, so we omit the proof.

**Proposition 6.11.** *Let* $\tau = c\sqrt{\log n}$ *for a constant* $c > 0$, *and define* $q = q(\tau)$ *as in* (64). *Then under the planted model* $\mathbb{P}$ *we have* $\mathbb{E}[T_\tau^-] = (n - k)q$ *and* $\mathrm{Var}(T_\tau^-) \leq n^{1 - c^2/2 + o(1)}$.

Focusing now on $T_\tau^+$, we will establish the following result in the next section.

**Proposition 6.12.** *Let $\tau = c\sqrt{\log n}$ for a constant $c > 0$. Fix any constant $\tilde{c} > 0$ such that*

$$\max\{c - \sqrt{R(1 - \theta)}, 0\} < \tilde{c} < \min\{\sqrt{2\theta}, c\}.$$

*Then under the planted model $\mathbb{P}$ we have that with probability $1 - o(1)$,*

$$T_\tau^+ \geq n^{\theta - \tilde{c}^2/2 - o(1)}.$$

6.2.3. *Proof of Proposition 6.12.* Recall $Y = (k + \sigma^2)^{-1/2} \left( \sum_{j \in S} X_j + W \right)$ and so for $j \in S$ we compute

$$\langle X_j, Y \rangle = (k + \sigma^2)^{-1/2} \left( \sum_{\ell \in S} \langle X_j, X_\ell \rangle + \langle X_j, W \rangle \right)$$

$$= (k + \sigma^2)^{-1/2} \left( \|X_j\|_2^2 + \langle X_j, Z_j \rangle \right)$$

where

$$Z_j = \sum_{\ell \in S, \, \ell \neq j} X_\ell + W.$$

We define the counting random variable

(65) $$\tilde{T} := \sum_{j \in S} I_j,$$

where $I_j = \mathbb{1}\{\tilde{z}_j \geq \tilde{c}\sqrt{\log n}\}$ where $\tilde{z}_j = \langle X_j, Z_j \rangle / \|Z_j\|_2$ and $\tilde{c} \in (0, c)$ is the constant defined in the statement of the proposition. The following lemma will allow us to analyze $\tilde{T}$ instead of $T_\tau^+$.

**Lemma 6.13.** *With probability $1 - n^{-\omega(1)}$,*

$$T_\tau^+ \geq \tilde{T}.$$

*Proof.* With probability $1 - n^{-\omega(1)}$, using standard concentration of the $\chi^2$ random variable we have the following norm bounds for some $\delta = n^{-\Omega(1)}$:

$$\|Y\|_2 \leq (1 + \delta)\sqrt{m}, \qquad \|X_j\|_2^2 \geq (1 - \delta)m, \qquad \|Z_j\|_2 \geq (1 - \delta)\sqrt{(k + \sigma^2)m}.$$

Suppose the above bounds hold and that $\tilde{z}_j \geq \tilde{c}\sqrt{\log n}$ holds for some $j$. It suffices to show $z_j \geq \tau$. We have, recalling $\sigma^2 = o(k)$,

$$z_j = \langle X_j, Y \rangle / \|Y\|_2$$

$$= (k + \sigma^2)^{-1/2} \|Y\|_2^{-1} \left( \|X_j\|_2^2 + \langle X_j, Z_j \rangle \right)$$

$$\geq \frac{1}{(1 + \delta)\sqrt{(k + \sigma^2)m}} \left( (1 - \delta)m + \tilde{z}_j \cdot \|Z_j\|_2 \right)$$

$$\geq \frac{1}{(1 + \delta)\sqrt{(k + \sigma^2)m}} \left( (1 - \delta)m + \tilde{c}\sqrt{\log n} \cdot (1 - \delta)\sqrt{(k + \sigma^2)m} \right)$$

$$= (1 - o(1)) \left( \sqrt{\frac{m}{k}} + \tilde{c}\sqrt{\log n} \right)$$

$$= (1 - o(1)) \left( \sqrt{R(1 - \theta)} + \tilde{c} \right) \sqrt{\log n}.$$

Since by assumption $\sqrt{R(1 - \theta)} + \tilde{c} > c$, we have for sufficiently large $n$ that $z_j \geq c\sqrt{\log n} = \tau$, as we wanted. $\qquad\square$

Note that the $\tilde{z}_j$ defining $\tilde{T}$ are distributed as $\tilde{z}_j \sim \mathcal{N}(0, 1)$ but they are not independent. Yet, by linearity of expectation, $\mathbb{E}[\tilde{T}] = \tilde{q}k$ where $\tilde{q} := \Pr\{\mathcal{N}(0, 1) \geq \tilde{c}\sqrt{\log n}\}$. By Lemma 6.10 we have $\tilde{q} = n^{-\tilde{c}^2/2 + o(1)}$. We now bound $\text{Var}(\tilde{T})$ by first establishing the following lemma.

**Lemma 6.14.** *Fix $j, \ell \in S$ with $j \neq \ell$. We have*

$$\Pr\left\{\tilde{z}_j \geq \tilde{c}\sqrt{\log n} \ \text{ and } \ \tilde{z}_\ell \geq \tilde{c}\sqrt{\log n}\right\} \leq (1 + n^{-\Omega(1)})\tilde{q}^2.$$

*Proof.* Let $Z_{j\ell} = \sum_{i \in S \setminus \{j,\ell\}} X_i + W$ and write

$$\tilde{z}_j = \frac{1}{\|Z_j\|_2} \left( \langle X_j, X_\ell \rangle + \langle X_j, Z_{j\ell} \rangle \right), \qquad \tilde{z}_\ell = \frac{1}{\|Z_\ell\|_2} \left( \langle X_j, X_\ell \rangle + \langle X_\ell, Z_{j\ell} \rangle \right).$$

The purpose of the above decomposition is to exploit the fact that $\langle X_j, Z_{j\ell} \rangle / \|Z_{j\ell}\|_2$ and $\langle X_\ell, Z_{j\ell} \rangle / \|Z_{j\ell}\|_2$ are independent, and the other terms are small in magnitude in comparison.

By standard concentration, the following events all occur with probability $1 - n^{-\omega(1)}$, for some $\delta = n^{-\Omega(1)}$:

- $\|Z_j\|_2, \|Z_\ell\|_2 \geq (1 - \delta)\sqrt{(k + \sigma^2)m}$,
- $\|Z_{j\ell}\|_2 \leq (1 + \delta)\sqrt{(k + \sigma^2)m}$,
- $\langle X_j, X_\ell \rangle \leq \sqrt{m} \log n$.

The first two properties follow by standard concentration of the $\chi^2$ distribution, and the third property follows from the observation $\langle X_j, X_\ell \rangle / \|X_\ell\|_2 \sim \mathcal{N}(0,1)$ and that with probability $1 - n^{-\omega(1)}$, $\|X_\ell\|_2 \leq (1 + \delta)\sqrt{m}$. The above events imply

$$\tilde{z}_j \leq \frac{\sqrt{m} \log n}{(1 - \delta)\sqrt{(k + \sigma^2)m}} + \frac{\|Z_{j\ell}\|_2}{\|Z_j\|_2} \cdot \frac{\langle X_j, Z_{j\ell} \rangle}{\|Z_{j\ell}\|_2} = \frac{\log n}{(1 - \delta)\sqrt{k + \sigma^2}} + \frac{\|Z_{j\ell}\|_2}{\|Z_j\|_2} \cdot \frac{\langle X_j, Z_{j\ell} \rangle}{\|Z_{j\ell}\|_2}$$

and similarly for $\tilde{z}_\ell$. Using the fact that $\langle X_j, Z_{j\ell} \rangle / \|Z_{j\ell}\|_2$ and $\langle X_\ell, Z_{j\ell} \rangle / \|Z_{j\ell}\|_2$ are independent and distributed as $\mathcal{N}(0,1)$,

$$\Pr\left\{ \tilde{z}_j \geq \tilde{c}\sqrt{\log n} \ \text{ and } \ \tilde{z}_\ell \geq \tilde{c}\sqrt{\log n} \right\}$$

$$\leq n^{-\omega(1)} + \Pr\left\{ \frac{\langle X_j, Z_{j\ell} \rangle}{\|Z_{j\ell}\|_2} \wedge \frac{\langle X_\ell, Z_{j\ell} \rangle}{\|Z_{j\ell}\|_2} \geq \frac{1 - \delta}{1 + \delta} \left( \tilde{c}\sqrt{\log n} - \frac{\log n}{(1 - \delta)\sqrt{k + \sigma^2}} \right) \right\}$$

$$= n^{-\omega(1)} + \Pr\left\{ \mathcal{N}(0,1) \geq \frac{1 - \delta}{1 + \delta} \left( \tilde{c}\sqrt{\log n} - \frac{\log n}{(1 - \delta)\sqrt{k + \sigma^2}} \right) \right\}^2$$

$$= n^{-\omega(1)} + \Pr\left\{ \mathcal{N}(0,1) \geq \tilde{c}\sqrt{\log n} - n^{-\Omega(1)} \right\}^2.$$

Using Lemma A.7, this is

$$\leq n^{-\omega(1)} + \left[ \Pr\left\{ \mathcal{N}(0,1) \geq \tilde{c}\sqrt{\log n} \right\} \left( 1 + n^{-\Omega(1)} \right) \right]^2 = \left( 1 + n^{-\Omega(1)} \right) \tilde{q}^2,$$

where for the last equality we used Lemma 6.10. The proof is complete. $\qquad\square$

We now bound the variance of $\tilde{T}$ using Lemma 6.14:

$$\mathrm{Var}(\tilde{T}) = \mathbb{E}\left[ \left( \sum_{j \in S} I_j \right)^2 \right] - (\tilde{q}k)^2$$

$$= \sum_{j \in S} \mathbb{E}[I_j] + \sum_{j, \ell \in S, \, j \neq \ell} \mathbb{E}[I_j I_\ell] - (\tilde{q}k)^2$$

$$\leq \tilde{q}k + (1 + n^{-\Omega(1)}) \tilde{q}^2 k(k - 1) - (\tilde{q}k)^2$$

$$\leq \tilde{q}k + n^{-\Omega(1)} \tilde{q}^2 k^2.$$

Recall using Lemma 6.10 that $\mathbb{E}[\tilde{T}] = \tilde{q}k = n^{\theta - \tilde{c}^2/2 + o(1)}$. Using now that $\theta > \tilde{c}^2/2$, the variance bound above implies $\tilde{T} \geq (1 - o(1))\tilde{q}k = n^{\theta - \tilde{c}^2/2 + o(1)}$ with probability $1 - o(1)$. In particular, using Lemma 6.13, the proof of Proposition 6.12 is complete.

6.2.4. *Putting it Together.* We now combine the previous results to conclude Theorem 3.10(b).

*Proof of Theorem 3.10(b).* We first recap the conclusions of Propositions 6.9, 6.11, 6.12. Under $\mathbb{Q}$, we have $\mathbb{E}[T_\tau] = qn$ and $\mathrm{Var}(T_\tau) \leq n^{1-c^2/2+o(1)}$. Under $\mathbb{P}$, we have $T_\tau = T_\tau^+ + T_\tau^-$ with $\mathbb{E}[T_\tau^-] = q(n-k)$ and $\mathrm{Var}(T_\tau^-) \leq n^{1-c^2/2+o(1)}$. We need to choose constants $c > 0$ and $\tilde{c} > 0$ satisfying

$$(66) \qquad \max\{c - \sqrt{R(1-\theta)}, 0\} < \tilde{c} < \min\{\sqrt{2\theta}, c\},$$

in which case we have $T_\tau^+ \geq n^{\theta - \tilde{c}^2/2 + o(1)}$ with probability $1 - o(1)$.

To successfully distinguish, it suffices by Chebyshev's inequality to choose $c, \tilde{c} > 0$ satisfying (66) such that

$$\sqrt{\mathrm{Var}_\mathbb{Q}(T_\tau) + \mathrm{Var}_\mathbb{P}(T_\tau^-)} = o(n^{\theta - \tilde{c}^2/2 + o(1)} + \mathbb{E}_\mathbb{P}[T_\tau^-] - \mathbb{E}_\mathbb{Q}[T_\tau]).$$

Plugging in the bounds stated above and noting $\mathbb{E}_\mathbb{P}[T_\tau^-] - \mathbb{E}_\mathbb{Q}[T_\tau] = -qk = -n^{\theta - c^2/2 + o(1)}$ by Lemma 6.10, it suffices to have

$$n^{(1-c^2/2)/2+o(1)} = o(n^{\theta - \tilde{c}^2/2 + o(1)} - n^{\theta - c^2/2 + o(1)}),$$

or since $0 < \tilde{c} < c$,

$$n^{(1-c^2/2)/2+o(1)} = o(n^{\theta - \tilde{c}^2/2 + o(1)}),$$

i.e.,

$$(1 - c^2/2)/2 < \theta - \tilde{c}^2/2.$$

Therefore it suffices to choose (under the assumption $R > R_{\mathrm{LD}}$) $c, \tilde{c} > 0$ satisfying the following conditions:

  (i)  $0 < \tilde{c} < c$,
  (ii) $\theta - \tilde{c}^2/2 > (1 - c^2/2)/2$,
  (iii) $\theta > \tilde{c}^2/2$,
  (iv) $\sqrt{R(1-\theta)} + \tilde{c} > c$.

First consider the case $R > \frac{2(1-\sqrt{\theta})}{1+\sqrt{\theta}}$ for arbitrary $\theta \in (0, 1)$. Then we choose $\tilde{c} = \sqrt{2\theta} - \eta$ and $c = \sqrt{R(1-\theta)} + \sqrt{2\theta} - 2\eta$ for a sufficiently small constant $\eta = \eta(\theta, R) > 0$. This choice immediately satisfies conditions (i), (iii), (iv). To satisfy (ii), it suffices to have $c^2 > 2$ because of condition (iii). Thus, there exists $\eta > 0$ satisfying (ii) provided that $\sqrt{R(1-\theta)} + \sqrt{2\theta} > \sqrt{2}$, which simplifies to $R > \frac{2(1-\sqrt{\theta})}{1+\sqrt{\theta}}$. This completes the proof in the case $\theta \leq \frac{1}{4}$.

Now consider the remaining case where $\theta > \frac{1}{4}$ and $\frac{1-2\theta}{1-\theta} < R \leq \frac{2(1-\sqrt{\theta})}{1+\sqrt{\theta}}$. (This covers the case $\theta \geq 1/2$ because $\frac{1-2\theta}{1-\theta} \leq 0$ when $\theta \geq 1/2$.) Since $\frac{2(1-\sqrt{\theta})}{1+\sqrt{\theta}} < \frac{2\theta}{1-\theta}$ for all $\theta > \frac{1}{4}$, we have $R < \frac{2\theta}{1-\theta}$, i.e., $R(1-\theta)/2 < \theta$. This means we can choose $\tilde{c} = \sqrt{R(1-\theta)}$ to satisfy (iii). We also choose $c = 2\sqrt{R(1-\theta)} - \eta$ for sufficiently small $\eta > 0$, which satisfies (i) and (iv). Finally, for this choice of $c, \tilde{c}$, (ii) reduces to $R > \frac{1-2\theta}{1-\theta}$ which holds by assumption. This completes the proof. $\qquad \square$

6.2.5. *Approximate Recovery.* By a simple adaptation of the above proof, we can also prove the following guarantee for approximate recovery.

**Theorem 6.15** (Algorithm for Approximate Recovery)**.** *Consider sparse linear regression (Definition 3.8) in the scaling regime of Assumption 3.9. If $R > 2$ then there is a polynomial-time algorithm for approximate recovery, that is: given $(X, Y)$ drawn from $\mathbb{P}$, the algorithm outputs $\hat{u} \in \{0, 1\}^n$ such that*

$$\|\hat{u} - u\|_2^2 = o(k) \qquad \text{with probability } 1 - o(1).$$

*Proof.* Since $R > 2$, it is possible to choose a constant $c > 0$ such that

$$(67) \qquad \sqrt{2(1-\theta)} < c < \sqrt{R(1-\theta)}.$$

The estimator will take the form

$$\hat{u}_j = \mathbb{1}\left\{\frac{\langle X_j, Y\rangle}{\|Y\|_2} \geq \tau\right\} \qquad \text{where} \qquad \tau = c\sqrt{\log n}.$$

Note that $\|\hat{u} - u\|_2^2$ is simply the number of false positives $E^+ := |\operatorname{supp}(\hat{u}) \setminus \operatorname{supp}(u)|$ plus the number of false negatives $E^- := |\operatorname{supp}(u) \setminus \operatorname{supp}(\hat{u})|$. We will consider these two terms separately and show that both are $o(k)$ with high probability.

**False positives.** This case follows by an adaptation of the calculations in Section 6.2.1. Noting that the values $\langle X_j, Y \rangle / \|Y\|_2$ for $j \notin \operatorname{supp}(u)$ are i.i.d. $\mathcal{N}(0,1)$, we have $E^+ \sim \operatorname{Bin}(n-k, q)$ where $q := \Pr\{\mathcal{N}(0,1) \geq \tau\}$. This means $\mathbb{E}[E^+] = q(n-k) \leq qn$ and $\operatorname{Var}[E^+] = q(1-q)(n-k) \leq qn$. Using Lemma 6.10, $qn = n^{1-c^2/2+o(1)}$. Recalling $k = n^{\theta+o(1)}$ and $c > \sqrt{2(1-\theta)}$ (from (67)), this means $\mathbb{E}[E^+] = o(k)$ and $\operatorname{Var}[E^+] = o(k)$, and so Chebyshev's inequality implies $E^+ = o(k)$ with probability $1 - o(1)$.

**False negatives.** This case follows by an adaptation of the calculations in Section 6.2.2. Note that $E^-$ is equal to $k - T_\tau^+$ for $T_\tau^+$ as defined in Section 6.2.2. Therefore, the proof is complete by the following analogue of Proposition 6.12. $\qquad\square$

**Proposition 6.16.** *Let $\tau = c\sqrt{\log n}$ for a constant $c > 0$ satisfying (67). Then under the planted model $\mathbb{P}$ we have that with probability $1 - o(1)$,*

$$T_\tau^+ \geq (1 - o(1))k.$$

*Proof.* The proof is similar to that of Proposition 6.12, so we explain here the differences. Fix a constant $\tilde{c} > 0$ such that

$$c - \sqrt{R(1-\theta)} < \tilde{c} < 0,$$

which is possible due to (67). Define $\tilde{T}$ and $I_j$ as in the original proof (of Proposition 6.12), using this value of $\tilde{c}$; see (65). The main difference is that now we have $\tilde{c} < 0$ instead of $\tilde{c} > 0$. The result of Lemma 6.13 remains true, namely $T_\tau^+ \geq \tilde{T}$ with probability $1 - n^{-\omega(1)}$; the proof is essentially the same, except now (since $\tilde{c} < 0$) we need to use the upper bound $\|Z_j\|_2 \leq (1+\delta)\sqrt{(k+\sigma^2)m}$ instead of a lower bound.

It remains to compute the mean and variance of $\tilde{T}$ in order to establish $\tilde{T} \geq (1-o(1))k$ with high probability. As in the original proof, $\mathbb{E}[\tilde{T}] = \tilde{q}k$ where $\tilde{q} := \Pr\{\mathcal{N}(0,1) \geq \tilde{c}\sqrt{\log n}\}$, but now since $\tilde{c} < 0$, the result diverges from the original and we instead have $\tilde{q} = 1 - n^{-\tilde{c}^2/2+o(1)} = 1 - n^{-\Omega(1)}$ (see Lemma 6.10).

The variance calculation is much easier than in the original proof: we can do away with Lemma 6.14 entirely and instead directly bound

$$\operatorname{Var}(\tilde{T}) = \mathbb{E}\left[\left(\sum_{j \in S} I_j\right)^2\right] - (\tilde{q}k)^2 \leq k^2 - (\tilde{q}k)^2 = k^2(1-\tilde{q})^2 = k^2 \cdot n^{-\Omega(1)},$$

since $\tilde{q} = 1 - n^{-\Omega(1)}$ from above. We have now shown $\mathbb{E}[\tilde{T}] = (1-o(1))k$ and $\operatorname{Var}(\tilde{T}) = o(k^2)$, so Chebyshev's inequality implies $\tilde{T} \geq (1-o(1))k$ with probability $1 - o(1)$ as desired. $\qquad\square$

## APPENDIX A. APPENDIX FOR SPARSE REGRESSION

### A.1. **Bernstein's Inequality.**

**Theorem A.1** (see [BLM13], Theorem 2.10)**.** *For $\nu, c > 0$, let $X_1, \ldots, X_n$ be independent with $\sum_{i=1}^n \mathbb{E}[X_i^2] \leq \nu$ and*

$$\sum_{i=1}^n \mathbb{E}|X_i|^q \leq \frac{q!}{2}\nu c^{q-2} \qquad \text{for all integers } q \geq 3.$$

*Then for all $y > 0$,*

$$\Pr\left\{\sum_{i=1}^n (X_i - \mathbb{E}X_i) \geq \sqrt{2\nu y} + cy\right\} \leq e^{-y}.$$

**Corollary A.2.** *For $a, b \in \mathbb{R}$, let $X_1, \ldots, X_n$ be i.i.d. and distributed as $X = ag^2 + bgg'$ where $g, g'$ are independent $\mathcal{N}(0, 1)$. Then for all $y > 0$,*

$$\Pr\left\{\sum_{i=1}^n X_i \geq an + \sqrt{2(3a^2 + b^2)ny} + 10\sqrt{a^2 + b^2}\, y\right\} \leq e^{-y} \qquad and$$

$$\Pr\left\{\sum_{i=1}^n X_i \leq an - \sqrt{2(3a^2 + b^2)ny} - 10\sqrt{a^2 + b^2}\, y\right\} \leq e^{-y}.$$

*Proof.* We will apply Theorem A.1. Set

$$\nu = \sum_{i=1}^n \mathbb{E}[X_i^2] = (3a^2 + b^2)n.$$

For any integer $q \geq 3$,

$$\begin{aligned}
\mathbb{E}|X|^q &= \mathbb{E}|g(ag + bg')|^q \\
&\leq \left(\mathbb{E}|g|^{2q} \cdot \mathbb{E}|ag + bg'|^{2q}\right)^{1/2} \\
&= (a^2 + b^2)^{q/2}\, \mathbb{E}|g|^{2q} \\
&= \pi^{-1/2}\, (a^2 + b^2)^{q/2}\, 2^q\, \Gamma\left(q + \frac{1}{2}\right) \\
&\leq \pi^{-1/2}\, (a^2 + b^2)^{q/2}\, 2^q\, \Gamma(q + 1) \\
&= \pi^{-1/2}\, (a^2 + b^2)^{q/2}\, 2^q\, q!
\end{aligned}$$

and so

$$\sum_{i=1}^n \mathbb{E}|X_i|^q \leq \frac{q!}{2}\nu \cdot \pi^{-1/2}\, (a^2 + b^2)^{q/2}\, (3a^2 + b^2)^{-1}\, 2^{q+1} \leq \frac{q!}{2}\nu \cdot \pi^{-1/2}\, (a^2 + b^2)^{q/2-1}\, 2^{q+1}.$$

Set $c = 10\sqrt{a^2 + b^2}$ so that $\pi^{-1/2}\, (a^2 + b^2)^{q/2-1}\, 2^{q+1} \leq c^{q-2}$ for all $q \geq 3$. This completes the proof.    $\square$

## A.2. Hölder's Inequality.

**Proposition A.3** (Hölder's inequality). *Let $r \geq 1$ and $p, q \in [1, \infty]$ with $\frac{1}{p} + \frac{1}{q} = \frac{1}{r}$, and let $X, Y$ be random variables. Then $\|XY\|_r \leq \|X\|_p \|Y\|_q$.*

**Proposition A.4.** *Let $r \geq 1$ and $p_1, \ldots, p_n \in [1, \infty]$ with $\sum_i \frac{1}{p_i} = \frac{1}{r}$, and let $X_1, \ldots, X_n$ be random variables. Then $\|\prod_i X_i\|_r \leq \prod_i \|X_i\|_{p_i}$.*

*Proof.* Proceed by induction on $n$. The base case $n = 2$ is Hölder's inequality. For $n \geq 3$, we have by Hölder that

$$\left\|\prod_{i=1}^n X_i\right\|_r \leq \|X_n\|_{p_n} \left\|\prod_{i=1}^{n-1} X_i\right\|_{\left(\frac{1}{r} - \frac{1}{p_n}\right)^{-1}}.$$

Since $\frac{1}{r} - \frac{1}{p_n} = \sum_{i=1}^{n-1} \frac{1}{p_i}$, the result follows using the induction hypothesis.    $\square$

## A.3. Wishart Distribution.

Recall that for an $m \times m$ matrix $V \succ 0$, the *Wishart distribution* $W_m(V, n)$ is the distribution of $Z^\top Z$ where $Z \in \mathbb{R}^{n \times m}$ has each row independently distributed as $\mathcal{N}(0, V)$. For $n \geq m$, the density of $U \sim W_m(V, n)$ (more precisely, the density of the diagonal and upper-triangular entries of $U$) is given by

$$f(U) = \frac{\det(U)^{(n-m-1)/2} \exp(-\frac{1}{2}\langle V^{-1}, U\rangle)}{2^{nm/2} \det(V)^{n/2} \Gamma_m(n/2)}$$

when $U \succ 0$ (and $f(U) = 0$ when $U \not\succ 0$), where $\Gamma_m$ is the multivariate gamma function [Wis28].

**Lemma A.5.** *Fix $t \in \mathbb{R}$ and a symmetric matrix $M \in \mathbb{R}^{m \times m}$ such that $tM \prec \frac{1}{2}I_m$. Then for $Z \in \mathbb{R}^{n \times m}$ with i.i.d. $\mathcal{N}(0,1)$ entries and an event $B(U)$ defined on symmetric matrices $U \in \mathbb{R}^{m \times m}$, it holds that*

$$\underset{Z}{\mathbb{E}} \, \mathbb{1}_{B(Z^\top Z)} \exp\left(t\langle M, Z^\top Z\rangle\right) = \det(I_m - 2tM)^{-n/2} \underset{U \sim W_m((I_m - 2tM)^{-1}, n)}{\Pr} \{B(U)\}$$

*where $W_m$ denotes the Wishart distribution defined above.*

*Proof.* Since $Z^\top Z \sim W_m(I_m, n)$, we can use the Wishart density from above to write

$$\underset{Z}{\mathbb{E}} \, \mathbb{1}_{B(Z^\top Z)} \exp(t\langle M, Z^\top Z\rangle) = \int_{U \succ 0} \mathbb{1}_{B(U)} \frac{\det(U)^{(n-m-1)/2} \exp(-\frac{1}{2}\langle I_m, U\rangle)}{2^{nm/2} \Gamma_m(n/2)} \exp(t\langle M, U\rangle) \, dU.$$

We will rewrite this in terms of a different Wishart distribution. Choose $V \in \mathbb{R}^{m \times m}$ so that $-\frac{1}{2}V^{-1} = -\frac{1}{2}I_m + tM$, that is, $V = (I_m - 2tM)^{-1}$. Then we have

$$\begin{aligned}
\underset{Z}{\mathbb{E}} \, \mathbb{1}_{B(Z^\top Z)} \exp(t\langle M, Z^\top Z\rangle) &= \int_{U \succ 0} \mathbb{1}_{B(U)} \frac{\det(U)^{(n-m-1)/2} \exp(-\frac{1}{2}\langle V^{-1}, U\rangle)}{2^{nm/2} \Gamma_m(n/2)} \, dU \\
&= \det(V)^{n/2} \int_{U \succ 0} \mathbb{1}_{B(U)} \frac{|U|^{(n-m-1)/2} \exp(-\frac{1}{2}\langle V^{-1}, U\rangle)}{2^{nm/2} \det(V)^{n/2} \Gamma_m(n/2)} \, dU \\
&= \det(V)^{n/2} \underset{U \sim W_m(V, n)}{\Pr} \{B(U)\}.
\end{aligned}$$

The conclusion follows. $\qquad\square$

### A.4. **Proof of Auxiliary Lemmas.**

*Proof of Lemma 6.4.* For the first part, notice

$$t = \log\log k + \ell \log 2 + \log\binom{k}{\ell} \leq \log\log k + \ell \log 2 + \ell \log k.$$

Hence for sufficiently large $k$, we have for all $\ell$ that $t \leq (1+\delta)\ell \log k$. Since $\binom{k}{\ell} \leq 2^k$, we also have $t \leq \log\log k + k \cdot 2\log 2 = O(k)$ while $m = \Theta(k \log k)$. As a result, the first term in $\Delta$ dominates: for sufficiently large $k$, we have for all $\ell$ that $10t \leq \delta\sqrt{2mt}$, and so

$$(68) \qquad \Delta \leq (1+\delta)\sqrt{2mt} \leq (1+\delta)\sqrt{(1+\delta)2\ell m \log k} \leq (1+\delta)^2 \sqrt{2\ell m \log k}.$$

The result follows since $\delta > 0$ was arbitrary.

For the second part, notice that for any fixed $\ell$ and $S$, the probability that (33) fails is

$$\Pr\left\{\sum_{i=1}^m s_i > \Delta\right\}$$

where $s_1, \ldots, s_m$ are i.i.d. with distribution $s = gg'$ where $g$ and $g'$ are independent $\mathcal{N}(0,1)$. By Corollary A.2 (with $a = 0$, $b = 1$), for all $t > 0$,

$$\Pr\left\{\sum_{i=1}^m s_i > \sqrt{2mt} + 10t\right\} \leq e^{-t}.$$

Plugging in $t = t(\ell)$ and taking a union bound over the choices of $\ell, S$, the probability that $A$ fails is at most

$$\sum_{\ell=1}^{k/2} \binom{k}{\ell} e^{-t} = \frac{1}{\log k} \sum_{\ell=1}^{k/2} 2^{-\ell} \leq \frac{1}{\log k},$$

completing the proof. $\qquad\square$

**Lemma A.6.** *If $X \geq 0$ is a nonnegative random variable and $A$ is an event of positive probability,*

$$\mathbb{E}[X \mid A] \leq \Pr(A)^{-1} \mathbb{E}[X].$$

*Proof.* Write

$$\mathbb{E}[X \mid A] = \frac{\mathbb{E}[X \cdot \mathbb{1}_A]}{\Pr(A)},$$

and using Hölder's inequality (Proposition A.3),

$$\mathbb{E}[X \cdot \mathbb{1}_A] = \|X \cdot \mathbb{1}_A\|_1 \leq \|X\|_1 \cdot \|\mathbb{1}_A\|_\infty = \mathbb{E}[X] \cdot 1,$$

completing the proof. $\qquad\square$

**Lemma A.7.** *For any $0 \leq \varepsilon < t$,*

$$\frac{\Pr\{\mathcal{N}(0,1) \geq t - \varepsilon\}}{\Pr\{\mathcal{N}(0,1) \geq t\}} \leq 1 + \frac{\varepsilon(t^2+1)}{t}\exp(\varepsilon t).$$

*Proof.* Letting $\Phi(t) = \Pr\{\mathcal{N}(0,1) \geq t\}$, we have $\Phi'(t) = -\frac{1}{\sqrt{2\pi}}\exp(-t^2/2)$ and so

$$\Phi(t - \varepsilon) \leq \Phi(t) + \frac{\varepsilon}{\sqrt{2\pi}}\exp(-(t-\varepsilon)^2/2).$$

Using the Gaussian tail lower bound $\Phi(t) \geq \frac{1}{\sqrt{2\pi}}\frac{t}{t^2+1}\exp(-t^2/2)$,

$$\frac{\Phi(t-\varepsilon)}{\Phi(t)} \leq 1 + \frac{1}{\Phi(t)}\frac{\varepsilon}{\sqrt{2\pi}}\exp(-(t-\varepsilon)^2/2) \leq 1 + \frac{\varepsilon(t^2+1)}{t}\exp(-(t-\varepsilon)^2/2 + t^2/2)$$

and the result follows. $\qquad\square$

## Appendix B. Orthogonal Polynomials

We provide here a sufficient condition for $L^2(\mathbb{Q})$ to admit a complete basis of orthonormal polynomials. We refer to [Akh20, Chapter 2] for further background. For a product measure $\mathbb{Q} = \prod_{i=1}^N Q_i$, it suffices that each $Q_i$ has finite moments of all orders, and that $Q_i$ is uniquely determined by its moment sequence. This in turn is guaranteed under various assumptions. For instance, $Q_i$ is determined by its moments if the characteristic function of $Q_i$ is analytic near zero, or under the weaker *Carleman's condition* [Akh20, Addendum 11, p. 85]:

$$(69) \qquad \sum_{k=1}^\infty m_{2k}^{-1/2k} = \infty \qquad \text{where} \qquad m_k = \mathbb{E}_{Y \sim Q_i}[Y^k].$$

Indeed we can first treat the univariate case $N = 1$, and then generalize to arbitrary $N$ by induction.

In the case $N = 1$, we can construct an orthonormal basis $(h_k)_{k \geq 0}$ in $\mathbb{R}[Y]$ by the Gram–Schmidt orthonormalization process. It remains to verify that the space of polynomials is dense in $L^2(\mathbb{Q})$. According to [Akh20, Theorem 2.3.3], a sufficient condition is that $\mathbb{Q}$ be determined by its moment sequence, i.e., no other probability measure has the same sequence of moments as $\mathbb{Q}$.

Generalizing to arbitrary $N$, we assume that each $Q_i$ is determined by its moments (e.g., satisfies Carleman's condition (69)). Since $\mathbb{Q} = \prod_{i=1}^N Q_i$, a basis of orthonormal polynomials with respect to $\langle \cdot, \cdot \rangle_\mathbb{Q}$ in $\mathbb{R}[Y_1, \ldots, Y_N]$ is given by $(h_\alpha)_{\alpha \in \mathbb{N}^N}$ (with $0 \in \mathbb{N}$ by convention) where $h_\alpha(Y) = \prod_{i=1}^N h_{\alpha_i}^{(i)}(Y_i)$ and for each $i \in [N]$, $(h_k^{(i)})_{k \geq 0}$ is a complete orthonormal basis of polynomials for $L^2(Q_i)$. It remains to show that such a basis is complete in $L^2(\mathbb{Q})$, i.e., the closure of $\text{span}\{h_\alpha : \alpha \in \mathbb{N}^N\}$ is $L^2(\mathbb{Q})$. Since we are dealing with linear spaces, it suffices to show that for any $f \in L^2(\mathbb{Q})$, if

$$(70) \qquad \langle f, h_\alpha \rangle_\mathbb{Q} = 0 \qquad \text{for all } \alpha \in \mathbb{N}^N,$$

then $f = 0$ ($\mathbb{Q}$-almost surely). We proceed by induction, the base case $N = 1$ having already been verified. Assume the above to be true for dimension up to $N - 1$. Let $\mathbb{Q} = \prod_{i=1}^N Q_i$ and let $f \in L^2(\mathbb{Q})$ such that (70) holds. This can be equivalently written as

$$(71) \qquad \int \tilde{f}(Y)h_{\alpha_N}^{(N)}(Y)Q_N(\mathrm{d}Y) = 0 \qquad \text{for all } \alpha_N \in \mathbb{N},$$

$$(72) \qquad \tilde{f}(y) := \mathbb{E}\Big[f(Y_1, \ldots, Y_{N-1}, y)\prod_{i=1}^{N-1} h_{\alpha_i}^{(i)}(Y_i)\Big],$$

where the above expectation is over $(Y_1, \cdots, Y_{N-1}) \sim \prod_{i=1}^{N-1} Q_i$. Since $(h_k^{(N)})_{k \geq 0}$ is a complete basis of $L^2(Q_N)$, Eq. (71) implies that $\tilde{f} = 0$ ($Q_N$-almost surely). Applying the induction hypothesis to Eq. (72), $f = 0$ ($\mathbb{Q}$-almost surely). This concludes the argument.

## Acknowledgements

AEA: Part of this work was done while this author was supported by the Richard M. Karp Fellowship at the Simons Institute for the Theory of Computing (Program on Probability, Geometry and Computation in High Dimensions). This author is grateful to Florent Krzakala for introducing him to the work of Franz and Parisi.

SBH: Parts of this work were done while this author was supported by a Microsoft Research PhD Fellowship, by a Miller Postdoctoral Fellowship, and by the Microsoft Fellowship at the Simons Institute for the Theory of Computing.

ASW: Part of this work was done at Georgia Tech, supported by NSF grants CCF-2007443 and CCF-2106444. Part of this work was done while visiting the Simons Institute for the Theory of Computing. Part of this work was done while with the Courant Institute at NYU, partially supported by NSF grant DMS-1712730 and by the Simons Collaboration on Algorithms and Geometry.

IZ: Supported by the Simons-NSF grant DMS-2031883 on the Theoretical Foundations of Deep Learning and the Vannevar Bush Faculty Fellowship ONR-N00014-20-1-2826. Part of this work was done while visiting the Simons Institute for the Theory of Computing. Part of this work was done while with the Center for Data Science at NYU, supported by a Moore-Sloan CDS postdoctoral fellowship.

The authors thank Cris Moore and the Santa Fe Institute for hosting the 2018 "Santa Fe Workshop on Limits to Inference in Networks and Noisy Data," where the initial ideas in this paper were formulated. The authors thank Aukosh Jagannath and anonymous reviewers for helpful comments on an earlier version of this work.

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

*Email address*: aswein@ucdavis.edu

(Zadik) Department of Mathematics, MIT

*Email address*: izadik@mit.edu