# OpenReview forum: "The Franz-Parisi Criterion and Computational Trade-offs in High Dimensional Statistics"
_NeurIPS.cc/2022/Conference — NeurIPS 2022 Accept_

### Official Review · Reviewer_sVoz · 2022-07-05

**Rating:** 7
**Confidence:** 3
**Soundness:** 3 good
**Presentation:** 3 good
**Contribution:** 3 good

**Summary:**

The authors consider the computational hardness of hypothesis testing in planted models.
Consider a "null" probability distribution and a "planted" distribution, i.e. a distribution depending on a hidden random signal with a given prior distribution.
Given a sample from one of the two distributions, how computationally hard is it to decide from which distribution it comes from?

The authors introduce a novel criterion, the Franz-Parisi (FP) criterion, that depends on local geometric properties of the planted/null distributions, and they define a related notion of FP hardness.
They show that FP hardness is related to an already established kind of computational hardness, namely low-degree (LD) hardness, in Gaussian additive models and some planted sparse models.
In particular, in the former class of models they prove that FP hardness is equivalent (up to technical hypothesis) to low-degree hardness, while in the latter they prove that FP hardness implies LD hardness.
Thus, they establish a link between geometrical properties of the planted/null distributions and computational hardness.

In Gaussian additive models, the authors prove that FP-hardness implies failure of local MCMC algorithms, in the sense that FP-hardness implies the existence of large free-energy barriers in the landscape explored by MCMC.
These prevent MCMC from fully exploring the configuration space in polynomial time.
Together with the LD-FP equivalence, this implies that low-degree hardness implies failure of local MCMC algorithms.

The authors then focus on the sparse linear regression model, where they provide new low-degree computational lower bounds using a proof technique that crucially relies on the new FP criterion.

**Questions:**

I am curious on how the FP hardness criterion relates to overlap gap properties and their consequences: do the authors have some intuition that could be added to the manuscript?

In the manuscript the authors hint at some relationship between the hypothesis testing setting they study and the (to me) more common estimation (retrieval of the planted signal) setting: is there some more precise relationship between computational and theoretical thresholds of the two settings?


**Limitations:**

The authors clearly assess the limitations of their results in the supplementary materials, where they provide counterexamples that suggest that their results do not point towards a complete equivalence between low-degree hardness and FP hardness.
These limitations do not undermine the validity or significance of their results.

**Strengths And Weaknesses:**

### Originality

The papers contains several novel results: the definition of the FP criterion, the study of the relationship of FP hardness with LD hardness and MCMC hardness in different models and new bounds for sparse linear regression.
Moreover, the proof of the bounds relies on the FP criterion, which has to be considered not only as a conceptual item, but also as a practical tool for proofs.
The authors clearly and convincingly state the relationship of their results with existing literature in many remarks both in the main text and in the supplementary materials.

I am not familiar with the literature enough to assess whether all relevant work has been adequately cited.

### Quality

The authors clearly state the assumptions underlying their results, and provide seemingly correct proofs.
I did not check the details of the proofs.

### Clarity

The manuscript is overall clearly and nicely written, and together with the extensive supplementary material provides enough information for the expert reader to be able to fully assess correctness and to reproduce the proofs.

### Significance

I believe that the results presented in this manuscript related to FP-hardness are significant.
Linking algorithmic hardness with statistical-physics-like properties of computational problems is a long-standing research effort. As far as my knowledge goes, it is still not completely clear how geometric properties of free-energy landscapes, such as replica symmetry breaking, relate to computational hardness, and this seems a promising step.

I am not qualified enough to properly assess the significance of the results on sparse regression, and on the novel proof technique based on the FP-criterion.

---

> ### Author Response · Authors · 2022-08-02
> **Rebuttal for Reviewer sVoz**
>
> We thank the reviewer for the positive comments and encouraging feedback. We are also very excited to make a first step on the "long-standing research effort" of "linking algorithmic hardness with statistical-physics-like properties of computational problems".
>
> "I am curious on how the FP hardness criterion relates to overlap gap properties and their consequences"
>
> That is a very good question. In many inference settings, the so-called overlap gap property (OGP) for inference has been a useful tool in predicting/proving the failure of low-temperature MCMC/local-search methods to estimate a planted signal. Indeed, OGP has played a key role in the low-temperature MCMC lower bounds for sparse regression [GZ22] and sparse PCA [BWZ20].
>
> Comparing now the FP and OGP, notice that the two predictions differ fundamentally in that they refer to two *different tasks*: FP predicts the hardness of detection/hypothesis testing, while OGP is tailored to the hardness of estimation tasks (see also below on that). No version of OGP has yet been considered in the context of detection.
>
> Despite that, one can naturally wonder whether, for the multitude of models where detection and estimation have similar behavior, one can relate the two predictions. This would have been very interesting, but at the moment we do not have a rigorous connection between OGP and FP hardness even under such an assumption. Yet, our intuition is that the two predicted hardness thresholds can significantly differ. To make a case in point, let us consider tensor PCA with a uniform signal on the hypercube [RM14], which is a Gaussian additive model where detection and estimation is believed to have the same behavior. In this model, it is well-known in the low-degree hardness literature that the currently predicted "computational limit" to estimate the signal coincides with the LD-hardness prediction. Hence, from our "FP-LD equivalence" Theorems 4.3-4.4 it also coincides with the FP hardness prediction. Yet, it is known that in tensor PCA zero-temperature local search/gradient descent methods fail at a much different "local" threshold [BGJ20]. While OGP has not yet been analyzed for tensor PCA, judging on the prior work, it is natural to conjecture that it appears exactly at this "local" threshold and not at the "computational threshold". Such an expected result will yield indeed a much different hardness OGP prediction compared with the FP one. It is natural to predict that this phenomenon should be generically true whenever MCMC methods underperform in estimating the planted signal as compared to other methods (see also [CMZ22]). Making the above observations rigorous is a very interesting direction for future work. We plan on adding such a discussion in the next version of our paper.
>
>
> "...the hypothesis testing setting they study and the (to me) more common estimation (retrieval of the planted signal) setting: is there some more precise relationship between computational and theoretical thresholds of the two settings?"
>
> This is also a very good question raised by the reviewer. While in many natural settings the detection/estimation thresholds coincide (or can at least be related), it is unfortunately not true that the two are equivalent. For example, when the prior is sufficiently asymmetric, detection is often strictly easier than estimation. Consider for example, the case of non-negative sparse PCA: the noise is $Z \sim N(0,I_n)$, the signal $x$ is $k$-sparse with coordinates $0$ or $1/\sqrt{k}$ and we observe $Y=\lambda x^{\otimes 2}+Z.$ When $k \gg \sqrt{n}$, polynomial-time/information theoretic (IT) detection is possible for a much wider range of $\lambda$ than polynomial-time/IT estimation.
>
> One can then naturally wonder whether (poly-time/IT) estimation implies (poly-time/IT) detection. Unfortunately, there are pathological examples that make such a claim false in full generality (see Section 3.4 of [BMV+17]). However, in most natural settings this direction holds: using the specifics of the studied model it is usually possible to establish a simple reduction proving that any (poly-time) algorithm that solves estimation can be made into a (poly-time) algorithm that solves detection (see e.g. [Appendix C, CGHK+22]). Characterizing a concise, general set of assumptions under which such a reduction holds is an interesting and fundamental question of future work.
>
>
> **All references are from the citation list in the supplementary material.

---

> > ### Comment · Reviewer_sVoz · 2022-08-05
> > **Response to rebuttal**
> >
> > I thank the authors for taking the time to answer my questions.
> >
> > I agree with the authors that adding a brief discussion about the link between FP/OGP and detection/estimation in their paper (or in the SM) would be beneficial to the reader.
> >
> > I have no further questions for the authors, and I confirm my previous evaluation.

---

### Official Review · Reviewer_AiU6 · 2022-07-08

**Rating:** 7
**Confidence:** 3
**Soundness:** 3 good
**Presentation:** 3 good
**Contribution:** 3 good

**Summary:**

## Summary of paper:

This work deals with proving algorithmic hardness of a class of statistical problems against local MCMC algorithms. To do this, they introduce a new algebraic quantity called the Franz-Parisi criterion (FP) and study it. For many learning problems, there is a phenomenon called the information-computation tradeoff where there is a gap between the information-theoretic sample complexity and the minimum sample complexity needed for polynomial time algorithms, e.g., in Sparse PCA, there is almost a quadratic gap. There have been many works trying to characterize this phenomenon by showing lower bounds against efficient algorithms. Techniques include reductions from conjectured hard problems (such as planted clique from graph theory), showing that the geometry landscape of the solution space have barriers in them (borrowing ideas from statistical physics), or showing hardness against powerful families of algorithms (such as the Sum-of-Squares hierarchy). In this work, they show that for the Gaussian additive noise model, under a strong symmetry assumptions, the so-called low-degree hardness (used before in the literature) implies failure of local MCMC algorithms. Their proof goes via this new FP criterion they introduce. They also show another usecase of this criterion, namely in obtaining tradeoffs for sparse regression. They finally study limitations of the FP criterion and show that it fails to predict the right threshold for simple mixtures of boolean distributions.

## Technical summary:

Consider the problem of hypothesis testing of Gaussian additive models, where null distribution is iid Gaussian and the alternative distribution has a planted spike sampled from a prior, e.g. tensor PCA. In this setting, this work shows that two hardness approaches are related. In particular, they show that a hardness hypothesis called low-degree hardness implies hardness of local MCMC algorithms.
- Low-degree hardness (LD) (definition 2.3) is a notion of hardness against hypothesis testing algorithms that utilize low-degree polynomials. Stemming from the analysis of the Sum of Squares family of algorithms, it is a simple and very effective technique to predict computational tradeoffs, however they do not say much about hardness of the Sum of Squares algorithms themselves. The hypothesis basically says that, under favorable conditions such as symmetry and noise robustness, if the projection of the likelihood ratio test (from the Neyman-Pearson lemma) to low-degree polynomials is bounded, then the problem is hard.
- Borrowing ideas from statistical physics, hardness of local algorithms can be obtained by exhibiting free-energy barriers in the solution landscape by approximating the free energy of a posterior distribution when we restrict the overlap with the ground truth. The relevant quantity they borrow inspiration from here is the Franz-Parisi potential, which when massaged appropriately via the Jensen's inequality, gives rise to a new quantity that's called the annealed potential. This is further modified to a notion this work introduces and terms the "Franz-Parisi criterion" (definition 3.2). This work shows that for additive Gaussian models, to show hardness of local alogorithms, it's enough to bound the FP criterion.

Both these approaches can be thought of as trying to bound the (computationally-bounded) Neyman-Pearson likelihood ratio and use it to predict computational barriers. This intuition is made formal in this work for Gaussian additive models. Main assumptions are:
- The null distribution has iid Gaussian entries
- The prior on the signal has sufficient symmetry, which necessiates among other things, thht every signal has the same L2-norm

Under these assumptions, LD hardness implies FP hardness (theorem 4.4, converse is also true) which in turn implies a free-energy barrier (theorem 5.3) and this in turn implies hardness against local MCMC algorithms (variants known in the literature before). As the authors point out, both assumptions seem crucial and not easy to be relaxed.

As another application of their criterion, the authors obtain computational phase transitions for certain specific parameter regimes of the sparse regression problem. In this setting, they found that their new criterion is easier to analyze than the low-degree ratio. However, it's still quite nontrivial and requires conditioning on a high-probability good event (definition 6.3 in the appendix) amidst tedious calculations.

Limits of this criterion are also explored, the authors show that it fails in several aspects. For example, it doesn't predict sharp thresholds for the estimation/recovery problem (compared to hypothesis testing). Moreover, it fails in boolean settings (as the authors point out, the issue is not booleanness, but something more inherent) and it also fails for the planted subgraph problem, where the LD criterion works instead. Therefore, this criterion should be considered as a means to an end in this research direction.

**Questions:**

A few questions were raised above.

## Other questions/comments:

1. Just to clarify, can the sample size in theorem 6.2, i.e. m = (1 + o(1))Rk log(n/k), be replaced by m <= (1 + o(1))Rk log(n/k)? It seems a bit unusual to set the sample size to be something specific

2. L86: "precisely" -> "precise"

3. L166: the partition function is not defined, I believe the first place it's defined is L271.

4. In the last paragraph of page 7 of the appendix, "says bounded" -> "stays bounded"

**Limitations:**

This is primarily a theoretical paper. Limitations of the results are sufficiently addressed, including the various assumptions and conceptual limitations (section 4 of the appendix). Negative societal impacts are not discussed, but I do not think there is any.

**Strengths And Weaknesses:**

## Summary of evaluation:

I was not able to verify the math in this timeframe. However, the results seem plausible, since the tradeoffs obtained here for various problems are expected to be true. The contributions in this work are conceptually interesting and raise potential directions for further research. However, the actual results obtained seem a bit weak since they require a lot of assumptions. Also, they are not readily applicable to other problems due to the difficulty of the calculations involved. Moreover, the proofs are highly technical and will take quite a bit of time to absorb for even experts in either statistical physics or statistics, let alone the average NeurIPS reader, which leads to concerns regarding the fit.

## Strengths:

1. Lower bounds against computational methods have long been studied from different perspectives, especially for average-case problems where NP-hardness results are not known. Recently, researchers have been trying to build bridges between different perspectives, e.g. Brennan et al. 2021 show that SQ algorithms and low-degree tests are equal power under some assumptions. Likewise, this work attempts to bridge the gap between statistical physics approaches and low degree tests.

2. Although the LD test became useful for its ease-of-use in predicting computational thresholds, the authors state that it's hard to apply it to certain parameter regimes of sparse regression. They overcome this with their new FP criterion.

3. The FP criterion recovers predicted thresholds for the spiked Wigner model of PCA. The example worked out is very helpful to understand it.

4. Formalizing all of the intuition this work brings about is nontrivial. The technical mastery in this work is commendable.

## Weaknesses:

1. The main weakness is that even if the FP criterion is a potentially useful criterion, it still seems quite hard to use it for other problems of interest. For example, even for the application to planted sparse models, the authors have to spend significant effort to bound the FP criterion, such as conditioning on a high probability event (definition 6.3). So, the usefulness of this criterion itself for other problems seems rather limited and technically tedious even for experts.

2. Based on my understanding, the transitive symmetry property seems to break when we consider sparse PCA where the sparsity is *at most* k, instead of *exactly* k. Therefore, the results here do not apply for this simple variant of sparse PCA since transitive symmetry seems to be a rather strong assumption. Could the authors clarify this?

3. Although the results are intriguing, my main concern is regarding the fit of this paper for this conference. Due to its length and non-triviality, It may be better suited for a journal or a more theory-focused conference.

4. For someone more from an ML background and not from a statistical physics background, it can be hard/impossible to follow section 1.3 in the appendix. Despite this, I understand that it is not easy to have a self-contained intro to the statistical physics terms used here. I'd suggest that a good reference should be cited to learn these better for the interested reader.

5. In the end of page 14 of the appendix, it's stated that very few tools are available to prove the success of MCMC methods for inference. If that's the case, then why are hardness against MCMC methods a good proxy for computational hardness?

---

> ### Author Response · Authors · 2022-08-02
> **Rebuttal for Reviewer AiU6**
>
> We thank the reviewer for their thought-provoking comments.
> We address the issues raised one by one.
>
>
>
> "This work deals with proving algorithmic hardness of a class of statistical problems against local MCMC algorithms"
>
> From our perspective, the main point of our work is connecting low-degree hardness and notions of hardness arising from free energy landscapes in statistical physics. Though properties of the free energy landscape can be used to rule out MCMC/gradient-based algorithms, the FP criterion is not equivalent to local MCMC algorithms, and, we believe, carries richer information about problem structure.
>
>
>
> "The main weakness is that even if the FP criterion is a potentially useful criterion, it still seems quite hard to use it for other problems of interest."
>
> We believe the FP criterion is actually a *simpler* way to prove low-degree hardness than the alternative direct approach, and we see this as a major strength rather than a weakness. FP can be applied to a wide range of problems; some will be easy to analyze (e.g. spiked Wigner) and others harder. Sparse regression is a difficult problem, and while the FP approach is complicated, it is still much simpler than the alternative approach would be. For some problems (like sparse regression and subsequent work on group testing [CGHK+22]), FP is the *only* way we know how to prove optimal low-degree lower bounds, and we feel this speaks to the value of the FP approach.
>
>
>
> "Therefore, the results here do not apply for this simple variant of sparse PCA since transitive symmetry seems to be a rather strong assumption"
>
> The reviewer is correct in pointing out that the assumptions of our meta-theorem for "low-degree hardness => MCMC lower bounds" rules out the "at most k" sparsity constraint on the prior. On the other hand, this is typical of meta-theorems; at the expense of making assumptions, we are obtaining a general connection between low-degree bounds and MCMC methods (MCMC is a simple and natural class of algorithms not obviously captured by the low-degree framework). What excites us in our meta-theorem is the fact that it captures, with a single argument, many interesting cases that have been studied individually (e.g. [BWZ20] for sparse PCA). We hope that this will be a stepping-stone towards a better understanding, in a unified fashion, the phenomena of statistical-to-computational gaps, and obtaining less restrictive assumptions is an important direction for future research.
>
>
>
> "Although the results are intriguing, my main concern is regarding the fit of this paper for this conference. Due to its length and non-triviality, It may be better suited for a journal or a more theory-focused conference."
>
> We feel that the core ideas in the paper, such as the definition of the FP criterion, are relatively simple to explain and would be appreciated by the NeurIPS community, where connections between ML/inference and statistical physics is a vibrant topic of interest (see e.g. Maillard-Loureiro-Krzakala-Zdeborova, NeurIPS'20 and many others) and results on MCMC methods in inference are traditionally discussed. We agree that our sparse regression example is fairly technical; we intentionally chose a well-known hard problem in the low-degree literature which has resisted analysis so far (and where non-trivial arguments are probably unavoidable) to illustrate the full power of the FP criterion. We feel that a reader need not follow the full details of this calculation in order to appreciate the main ideas presented in the paper.
>
>
>
> "I'd suggest that a good reference should be cited to learn these better for the interested reader."
>
> Thank you for pointing this out. We will add some references (e.g. the book of Mezard-Montanari "Information, Physics, and Computation").
>
>
>
> "few tools are available... why are hardness against MCMC methods a good proxy for computational hardness?"
>
> Indeed, lower bounds against MCMC methods should perhaps not be thought of as strong evidence for inherent computational hardness. Instead, MCMC is a simple and natural class of algorithms that are widely applicable, easy to implement and popular in practice, and so MCMC in inference has attracted much study (even in situations where the MCMC methods in question perform worse than the best-known poly-time algorithms). We also want to stress again that the FP criterion is not equivalent to hardness for MCMC, and as we show in some cases it is likely stronger (since it is equivalent to LDLR for many models).
>
>
>
> "Just to clarify, can the sample size in theorem 6.2, i.e. m = (1 + o(1))Rk log(n/k), be replaced by m <= (1 + o(1))Rklog(n/k)?"
>
> Adding more samples can only make the problem easier. The theorem includes both an upper bound and a lower bound. For the upper bound (algorithmic result), it would be fine to replace m = ... by m >= ..., and for the lower bound it would be fine to replace m = ... by m <= ....
>
> **All references are from the citation list in the supplement.

---

> > ### Comment · Reviewer_AiU6 · 2022-08-05
> > **Response to rebuttal**
> >
> > I thank the authors for their response. Since they have addressed some of my concerns, I'm willing to raise my score. I believe that as a reader of this work, it's clear that FP is not equivalent to MCMC (please clarify if something in my review seemed to suggest otherwise, happy to edit for the sake of correct exposition). Also, as a reader, I felt that the main contribution was the computational hardness conclusion, thanks for clarifying that it is in fact the connection between LD hardness and statistical physics hardness.

---

> > > ### Author Response · Authors · 2022-08-08
> > > **Thanks**
> > >
> > > Thank you very much for raising your score. (No need to edit the review -- we just wanted to emphasize that the main point is not just MCMC lower bounds.)

---

### Official Review · Reviewer_UuKm · 2022-07-10

**Rating:** 8
**Confidence:** 3
**Soundness:** 4 excellent
**Presentation:** 4 excellent
**Contribution:** 4 excellent

**Summary:**

The author(s) introduce a new "Franz-Parisi criterion" for predicting computational hardness in certain types of hypothesis testing problems, where the goal is to distinguish a mixture model with a random planted signal u from a null reference distribution. The criterion is, on one hand, a restriction of the second moment of the likelihood-ratio statistic to pairs of planted signals u,v having "typical" overlap under their prior. On the other hand, it is way of quantifying the growth/decay behavior of an annealed approximation to the Franz-Parisi potential locally around the overlap delta = 0. The authors posit that if this FP-criterion, when restricted to a range of overlaps (-delta,delta) having probability 1-e^{-D_n} for D_n growing slowly with n, is O(1) or 1+o(1), then this provides evidence for computational hardness of strong detection or weak detection respectively in the testing problem.

To support this claim, a number of interesting results are shown rigorously about this criterion:
(1) In a Gaussian additive model which includes spiked matrix and spiked tensor problems with Gaussian noise, it is shown that the FP criterion may be upper and lower bounded by the low-degree likelihood-ratio, up to the degree D_n modulo logarithmic factors. In certain relevant asymptotic regimes, this implies that FP's indication of hardness is equivalent to the low-degree method (which is believed to characterize the sharp threshold for hardness in many of these problems), and it establishes a new connection between the low-degree method and more geometric notions of overlap.
(2) Under an additional symmetry assumption for the Gaussian additive model, it is shown that FP-hardness formally implies a free energy barrier in the posterior distribution of the planted signal, which in turn implies a hitting time lower bound for MCMC algorithms having sufficiently "local" moves. Coupled with the above result, this provides a new method for showing hardness results for local MCMC algorithms using the low-degree approach.
(3) One direction of the above equivalence, that FP-hardness implies low-degree hardness, is shown also for a different class of planted sparse models satisfying a certain symmetry of the planted distribution. This implication is used to establish low-degree-hardness of detection in a sparse linear regression model, up to an sparsity-undersampling threshold curve that appears to be sharp in the studied regime, by bounding a conditional low-degree second-moment via the corresponding conditional FP-criterion.

Outside of the above settings, a discussion of some insightful counterexamples is also provided, where the FP criterion (or overlap-gap-based criteria in general) does not seem to accurately predict hardness of the testing problem.

**Questions:**

I might suggest to the author(s) to include some more of the discussion in Section 1.3 of the supplement (on the Franz-Parisi potential) into the main text, if this is possible. Currently I find Remark 3.3 a bit hard to understand (i.e. what is the relation between the FP potential and a restriction of the partition function?). Without this, it's hard to grasp the intuition behind the main Definition 3.2, as well as how the local/global properties of Figure 1 should be interpreted in the discussions on page 5.

A few typos:
p2 l86, "its precisely domain"
p3 l121, "computational" is repeated
p5 l189, "local minimum separated from", minimum -> maximum

**Limitations:**

Yes

**Strengths And Weaknesses:**

I find the paper to be very interesting, and am strongly supportive of its publication in NeurIPS. Some of the insights are:

(1) An annealed approximation to the Franz-Parisi free energy admits a second-moment type interpretation, and connects to the classical second-moment method in testing.
(2) A formal connection can be shown between restricting the geometric overlap and restricting the polynomial degree in the second-moment method. This seems to be particularly surprising in the planted sparse model setting of Section 3 where there is no explicit form for the low-degree likelihood ratio in terms of the overlap, and where the implication is shown from a symmetry condition alone.
(3) The connection can be used in both directions, to provide geometric approaches to analyzing the (conditional) low-degree likelihood ratio statistic, and to provide algebraic methods of analyzing the overlap landscape. It's helpful to have these connections elucidated at the level of generality of this current work, even if versions of these implications were used implicitly before in more specific models.

I guess the main weakness seems to be that it's unclear whether this FP-criterion fully captures the "physics intuition" for hardness in these types of problems, as (a) it is defined in terms of an annealed potential, and it's a bit unclear (at least to me) when this should have the same behavior as the replica-symmetric potential or the true quenched potential, and (b) it pertains to the properties of this potential only locally near 0, rather than to the entire overlap landscape. But I think these issues are adequately discussed in Section 4, and in my opinion they provide food for thought and do not detract much from the strengths of the work.

---

> ### Author Response · Authors · 2022-08-02
> **Rebuttal for Reviewer UuKm**
>
> We greatly thank the reviewer for their careful reading and positive and encouraging feedback. We will make sure to add more details to Remark 3.3, borrowed from the discussion in Section 1.3 of the supplement, to make it more intuitive and accessible to the reader.

---

### Meta-Review · Area_Chair_VxRC · 2022-08-25

**Recommendation:** Accept
**Confidence:** Certain

**Metareview:**

The paper establishes new bridges between two fundamental notions for the study of average case hardness of hypothesis testing problems: low-degree functions and the Franz-Parisi free energy, and also connect them to MCMC hardness. The paper has been very well received by the reviewers, who have done a great job. I agree with the reviewers on the quality of the paper. I also think that despite NeurIPS not being the most obvious avenue to submit such a paper, there indeed is an active sub-community of NeurIPS readers very much enthusiastic about such results, connections with statistical physics, etc. All issues raised seems to have been properly answered by the authors, so this is a clear accept for a high-quality paper full of rich notions and new fundamental connections.

**Award:**

Yes

---

### Decision · Program_Chairs · 2022-09-14

Accept